# PERTURBATIONS MATTER: SENSITIVITY-GUIDED HALLUCINATION DETECTION IN LLMS

## ABSTRACT

Hallucination detection is essential for ensuring the reliability of large language models. Internal representation–based methods have emerged as the prevailing direction for detecting hallucinations, yet the internal representations often fail to yield clear separability between truthful and hallucinatory content. To address this challenge, we study the separability of the sensitivity to prompt-induced perturbations in the internal representations. A theory is established to show that, with non-negligible probability, each sample admits a prompt under which truthful samples exhibit greater sensitivity to prompt-induced perturbations than hallucinatory samples. When the theory is applied to the representative datasets, the probability reaches nearly $99\%$, suggesting that sensitivity to perturbations provides a discriminative indicator. Building on this insight, we propose a theory-informed method Sample-Specific Prompting (SSP), which adaptively selects prompts to perturb the model's internal states and measures the resulting sensitivity as a detection indicator. Extensive experiments across multiple benchmarks demonstrate that SSP consistently outperforms existing hallucination detection methods, validating the practical effectiveness of our method SSP in hallucination detection.

## 1 INTRODUCTION

Large language models (LLMs) have shown remarkable performance in natural language understanding and generation tasks (Achiam et al., 2023; Grattafiori et al., 2024). However, hallucination in generated text remains a critical challenge, arising when LLMs produce outputs that are grammatically and logically coherent but lack factual accuracy or verifiable evidence (Joshi et al., 2017; Lin et al., 2022a). Such hallucinations undermine user trust and pose risks in high-stakes areas such as healthcare, law, and scientific research (Ji et al., 2023; Liu et al., 2024b). To address this issue, hallucination detection has attracted extensive attention in recent research (Manakul et al., 2023).

Previous detection methods can be roughly divided into two main categories: self-assessment (Kadavath et al., 2022; Zhou et al., 2023; Lin et al., 2022b) and internal representation-based methods (Du et al., 2024; Azaria & Mitchell, 2023; Marks & Tegmark, 2024; Yin et al., 2024). Self-assessment estimates the factuality of a response by leveraging the confidence in the model output. Internal representation-based methods primarily leverage the embeddings of off-the-shelf LLMs to classify outputs as either truthful or hallucinatory. The internal representation-based methods generally outperform self-assessment, and thus have emerged as the prevailing research direction.

Despite notable progress, the internal representation-based methods (Yin et al., 2024; Du et al., 2024; Kossen et al., 2024) face fundamental bottlenecks for detection, which impose inherent limits on their future development. Recent work (Park et al., 2025) demonstrates that the internal representations of LLMs frequently *fail to provide a clear separation between truthful and hallucinatory content* (see Figure 1a). As a result, the effectiveness of internal representation-based methods is inherently limited by the separability of internal representations. This motivates the a critical question: *is it possible to overcome the inherent separability bottleneck of internal representations?*

To tackle this question, we start from an empirical observation: in the experimental setup of Figure 1b, using the sensitivity of prompt-induced perturbations in internal representations as an evaluation score yields *near-perfect separability* between truthful and hallucinatory samples. To demystify this insightful observation, we develop a theory (see Section 4) stating that *for each sample, there exists an associated prompt, and with non-negligible probability, the sensitivity of truthful samples*

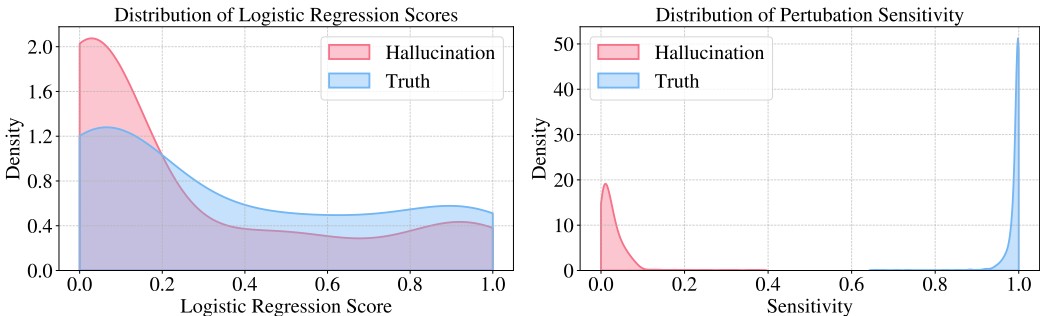

(a) Logistic Regression Score Distribution.  (b) Prompt-Induced Perturbation Sensitivity.

Figure 1: Empirical analysis conducted on 200 randomly selected samples from the TruthfulQA dataset (Lin et al., 2022a). (a) A logistic regression model was fitted on internal representations, showing weak separability. (b) For each sample, we apply an individually optimized prompt perturbation and measure its sensitivity using the cosine similarity between representations before and after perturbation. We find that this sensitivity provides effective separability between truthful and hallucinatory samples. Details for (a) and (b) are provided in **Appendix B**.

*to prompt-induced perturbations exceeds that of the hallucinatory samples.* We further apply our theory on the representative datasets (Reddy et al., 2019; Lin et al., 2022a; Joshi et al., 2017; Clark et al., 2020), showing that the probability reaches nearly **99**%, thereby statistically guaranteeing that the sensitivity to prompt-induced perturbations in internal representations, when used as an evaluation score, does not suffer from the separability bottleneck.

In light of the above analysis, we propose a novel method *Sample-Specific Prompting* (SSP), which leverages the sensitivity to prompt-induced perturbations as a discriminative indicator for hallucination detection. Instead of relying on static or handcrafted prompts, SSP dynamically generates tailored prompts for each question–answer pair to enhance the sensitivity of truthful samples to perturbations while reducing that of hallucinatory ones. Furthermore, SSP introduces a lightweight encoder to extract features before and after perturbation and employs a contrastive training objective that encourages larger representation shifts for truthful samples and smaller shifts for hallucinated ones. In effect, the joint learning of perturbation prompts and representation encodings makes SSP a more effective method for exploiting prompt-induced perturbations in hallucination detection.

Extensive experiments demonstrate the effectiveness of SSP across diverse datasets CoQA (Reddy et al., 2019), TruthfulQA (Lin et al., 2022a), TriviaQA (Joshi et al., 2017) and TydiQA-GP (Clark et al., 2020), compared with the state-of-the-art (Kadavath et al., 2022; Azaria & Mitchell, 2023; Hu et al., 2024). Also, our results indicate that SSP generalizes well across different domains. Our main contributions are summarized as follows:

• We are the first to leverage the sensitivity of LLM internal representations to input perturbations for hallucination detection, offering a novel perspective to hallucination detection.

• We analyze the sensitivity of LLM internal representations to input perturbations and theoretically establish its feasibility for hallucination detection.

• We propose a theory-informed method SSP, which leverages sensitivity to prompt-induced perturbations as a discriminative indicator for hallucination detection.

## 2 PRELIMINARY

**LLMs and Token Sequences.** Following Oh et al. (2025); Du et al. (2024), we use a distribution $P_{\boldsymbol{\theta}}(\cdot)$ over token sequences to define LLM, where $\boldsymbol{\theta}$ is the model parameters. Given a token sequence $\mathbf{Q} = [x_1, \dots, x_k]$ representing the question, where each $x_i$ is the $i$-th token in the sequence. $P_{\boldsymbol{\theta}}(\cdot)$ generates an answer $\mathbf{A} = [x_{k+1}, \dots, x_{k+q}]$ by predicting each token based on the preceding context:

$$P_{\boldsymbol{\theta}}(x_i | x_1, \dots, x_{i-1}), \text{ for } i = k+1, \dots, k+q. \tag{1}$$

**Truthful-answer and Hallucinatory-answer Domains.** Let $\mathcal{Q}$ and $\mathcal{A}$ denote the spaces of questions and answers, respectively. We introduce two domains over $\mathcal{Q} \times \mathcal{A}$:

- The *truthful-answer domain* is a joint distribution $P_{Q,T}$, where $Q \in \mathcal{Q}$ is a random variable representing questions and $T \in \mathcal{A}$ is a random variable representing the truthful answers.

- The *hallucinatory-answer domain* is a joint distribution $P_{Q,H}$, where $Q$ is defined as above and $H \in \mathcal{A}$ is a random variable representing the hallucinatory answers.

**Dataset Format.** Given the truthful-answer domain $P_{Q,T}$, each sample sampled from $P_{Q,T}$ consists of a question $\mathbf{Q}$ and a reference answer $\mathbf{A}^{\mathrm{ref}}$. The dataset sampled from $P_{Q,T}$ can be expressed as $\mathcal{D} = \{(\mathbf{Q}_1, \mathbf{A}_1^{\mathrm{ref}}), \ldots, (\mathbf{Q}_n, \mathbf{A}_n^{\mathrm{ref}})\}$, where $n$ is the number of samples.

Given a question $\mathbf{Q} \sim P_Q$, the LLM $P_{\boldsymbol{\theta}}(\cdot)$ generates an answer $\mathbf{A} \sim P_{\boldsymbol{\theta}}(\cdot|\mathbf{Q})$. Each generated answer $\mathbf{A}$ is assigned a binary label $y \in \{-1, 1\}$ according to its semantic consistency with the reference answer $\mathbf{A}^{\mathrm{ref}}$. Specifically, if $\mathbf{A}$ aligns with $\mathbf{A}^{\mathrm{ref}}$, it is labeled as truthful ($y = 1$); otherwise, it is labeled as hallucinatory ($y = -1$). The labeled dataset $\mathcal{D}_l$ is thus defined as:

$$\mathcal{D}_l = \{(\mathbf{Q}_1, \mathbf{A}_1, y_1), \ldots, (\mathbf{Q}_n, \mathbf{A}_n, y_n)\}. \tag{2}$$

**AUROC and Separability.** The AUROC serves as the primary evaluation metric for hallucination detection (Du et al., 2024). Formally, given the truthful-answer domain $P_{Q,T}$ and the hallucinatory-answer domain $P_{Q,H}$, the AUROC of a scoring function $r : \mathcal{Q} \times \mathcal{A} \to \mathbb{R}$ is defined as follows:

$$\mathrm{AUROC}(r; P_{Q,T}, P_{Q,H}) = P\big(r(\mathbf{Q}, \mathbf{T}) > r(\mathbf{Q}', \mathbf{H}')\big) + \frac{1}{2}P\big(r(\mathbf{Q}, \mathbf{T}) = r(\mathbf{Q}', \mathbf{H}')\big), \tag{3}$$

where $(\mathbf{Q}, \mathbf{T}) \sim P_{Q,T}$ is the truthful sample, and $(\mathbf{Q}', \mathbf{H}') \sim P_{Q,H}$ is the hallucinatory sample. In this work, we define the separability via the core component of AUROC, formally given by:

$$\mathrm{SEP}(r; P_{Q,T}, P_{Q,H}) = P\big(r(\mathbf{Q}, \mathbf{T}) > r(\mathbf{Q}', \mathbf{H}')\big). \tag{4}$$

**Hallucination Detection.** Given the training dataset $\mathcal{D}_l = \{(\mathbf{Q}_1, \mathbf{A}_1, y_1), \ldots, (\mathbf{Q}_n, \mathbf{A}_n, y_n)\}$ as introduced in Eq. (2), the goal of hallucination detection is to learn a detector $G$, based on a given LLM $P_{\boldsymbol{\theta}}(\cdot)$ and $\mathcal{D}_l$, such that for any question $\mathbf{Q} \sim P_Q$ and a corresponding answer $\mathbf{A}$,

$$G(\mathbf{Q}, \mathbf{A}) = 1, \text{ if } \mathbf{A} \sim P_{T|Q}(\cdot \mid \mathbf{Q}); \text{ otherwise, } G(\mathbf{Q}, \mathbf{A}) = -1, \tag{5}$$

where $1$ indicates that $\mathbf{A}$ is truthful, and $-1$ indicates that $\mathbf{A}$ is hallucinatory.

## 3 RELATED WORK

R#PEBK

**Hallucination detection** has become an increasingly important research topic, aiming to address the safety and reliability challenges of deploying LLMs in real-world applications (Ji et al., 2023; Liu et al., 2024b; Huang et al., 2025; Zhang et al., 2025b; Xu et al., 2024; Zhang et al., 2023; Chern et al., 2023). Previous detection methods can be roughly divided into two main categories: self-assessment (Kadavath et al., 2022; Zhou et al., 2023; Lin et al., 2022b) and internal representation-based methods (Du et al., 2024; Azaria & Mitchell, 2023; Marks & Tegmark, 2024; Yin et al., 2024).

**Self-assessment.** These methods estimate response factuality based on output confidence or consistency measures (Kadavath et al., 2022; Lin et al., 2022b; Gao et al., 2024). While intuitive, approaches like probability verbalization or perturbation-based scoring (Gao et al., 2024) often suffer from model overconfidence and sensitivity to superficial variations (Kaddour et al., 2023).

R#f1QU

**Internal Representation-based Methods.** Recent studies demonstrate that LLM internal representations (e.g., hidden states) encode truthfulness information (Azaria & Mitchell, 2023; Du et al., 2024; Bürger et al., 2024; Liu et al., 2024c; Zhang et al., 2025a). Although these methods generally outperform self-assessment, their robustness is often constrained by the separability of features in complex, open-ended generation tasks compared to artificial setups (Park et al., 2025).

R#3guq

Due to space limitations, a detailed discussion of related work is provided in **Appendix C**.

# 4 SEPARABILITY OF PROMPT-INDUCED PERTURBATION SENSITIVITY

Before introducing our method, we first analyze the separability of perturbation sensitivity in this section. *Due to space constraints, all proofs are provided in **Appendix D**.*

## 4.1 SENSITIVITY OF PROMPT-INDUCED PERTURBATIONS

Recent prevailing methods for hallucination detection (Azaria & Mitchell, 2023; Marks & Tegmark, 2024; Yin et al., 2024; Du et al., 2024; Kossen et al., 2024) rely on internal representations, classifying outputs as truthful or hallucinatory by leveraging embeddings from pre-trained LLMs. However, as pre-trained LLMs are trained for next-token prediction, their embeddings inherently favour fluency and syntactic correctness, while often overlooking truthful accuracy (Radford et al., 2019). Motivated by this limitation, recent work (Park et al., 2025) claims that the internal representations of LLMs frequently fail to provide a clear separation between truthful and hallucinatory samples.

In Figure 1a, we validate the claim given by Park et al. (2025). As shown in Figure 1a, the last-token embeddings of truthful and hallucinatory samples from TruthfulQA (Lin et al., 2022a) largely overlap, highlighting the lack of a clear separation. Hence, the effectiveness of these internal representation-based methods (Azaria & Mitchell, 2023; Marks & Tegmark, 2024; Yin et al., 2024; Du et al., 2024; Kossen et al., 2024) is limited by the separability of the internal representations. In light of this, we raise the question of *whether it is possible to overcome the inherent separability bottleneck of internal representations*. To tackle this question, we study whether prompt-induced perturbation sensitivity in internal representations has the potential for strong separability.

**Formalizing Perturbation Sensitivity.** Following prior work (Du et al., 2024; Park et al., 2025; Azaria & Mitchell, 2023; Chen et al., 2024; Guo et al., 2021), we define the internal representation $\mathbf{E}_{\boldsymbol{\theta}}(\cdot)$ of the LLM $P_{\boldsymbol{\theta}}(\cdot)$ as the embedding of the last token. Given a prompt $\mathbf{P}$, and a question-answer pair $(\mathbf{Q}, \mathbf{A})$, the prompt-induced perturbation sensitivity is defined as follows:

$$\Delta\mathbf{E}_{\boldsymbol{\theta}}(\mathbf{Q}, \mathbf{A}, \mathbf{P}) = \text{Dist}\big(\mathbf{E}_{\boldsymbol{\theta}}(\mathbf{Q}, \mathbf{A}, \mathbf{P}), \mathbf{E}_{\boldsymbol{\theta}}(\mathbf{Q}, \mathbf{A})\big), \quad (6)$$

where $\text{Dist}(\cdot, \cdot)$ is the measure of the difference between $\mathbf{E}_{\boldsymbol{\theta}}(\mathbf{Q}, \mathbf{A}, \mathbf{P})$ and $\mathbf{E}_{\boldsymbol{\theta}}(\mathbf{Q}, \mathbf{A})$.

**Preliminary Observation of Perturbation Sensitivity.** To investigate the separability of perturbation sensitivity, we construct an *oracle setting* in which, for each sample, we optimize a corresponding prompt, such that the perturbation sensitivity is maximized when the answer is truthful, and minimized when the answer is hallucinatory, i.e., for any sample $(\mathbf{Q}_i, \mathbf{A}_i, y_i) \in \mathcal{D}_l$,

$$\mathbf{P}_i^* \in \arg\max_{\mathbf{P}} \ y_i \cdot \Delta\mathbf{E}_{\boldsymbol{\theta}}(\mathbf{Q}_i, \mathbf{A}_i, \mathbf{P}). \quad (7)$$

In Figure 1b, we present the empirical result under the oracle setting (see **Appendix B** for experimental details). We observe that the separability of perturbation sensitivity reaches nearly $100\%$, which implies the aspiration of addressing the separability bottleneck of the internal representations.

## 4.2 SEPARABILITY OF PERTURBATION SENSITIVITY

Here, we develop a statistical analysis that characterizes the separability of perturbation sensitivity. We *continue to consider the oracle setting*, where the prompt is chosen to maximize perturbation sensitivity for truthful answers and minimize it for hallucinatory ones. Given the truthful-answer domain $P_{Q,T}$ and the hallucinatory-answer domain $P_{Q,H}$, we select the optimal prompt as follows:

$$\mathbf{P}^* \in \arg\max_{\mathbf{P}} \ y(\mathbf{Q}, \mathbf{A}) \cdot \Delta\mathbf{E}_{\boldsymbol{\theta}}(\mathbf{Q}, \mathbf{A}, \mathbf{P}), \quad (8)$$

where $y(\mathbf{Q}, \mathbf{A}) = 1$ if $(\mathbf{Q}, \mathbf{A}) \sim P_{Q,T}$, and $y(\mathbf{Q}, \mathbf{A}) = -1$ if $(\mathbf{Q}, \mathbf{A}) \sim P_{Q,H}$. Then, we consider the scoring function $r^* : \mathcal{Q} \times \mathcal{A} \to \mathbb{R}$, i.e.,

$$r^*(\mathbf{Q}, \mathbf{A}) = \Delta\mathbf{E}_{\boldsymbol{\theta}}(\mathbf{Q}, \mathbf{A}, \mathbf{P}^*), \ \text{ where } \mathbf{P}^* \text{ is defined in Eq. (8).} \quad (9)$$

The scoring function $r^*$ estimates the perturbation sensitivity under the oracle setting.

**Probabilistic Characterization of Separability.** Here, we give our core theorem, i.e., Theorem 1.

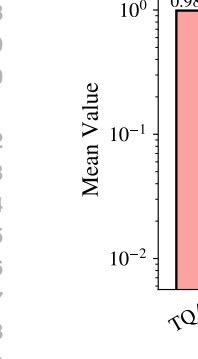
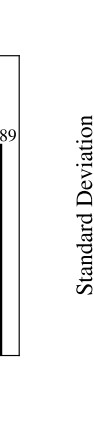

(a) Mean Value of Perturbation Sensitivity.

(b) Standard Deviation of Perturbation Sensitivity.

Figure 2: Perturbation sensitivity ($r^*$ in Eq. (9)) statistics across multiple datasets. The sensitivity of internal representations to prompt-induced perturbations is compared between truthful and hallucinatory samples across four representative datasets using LLaMA-3-8B-Instruct. Figure (a) reports the mean values, showing that truthful samples exhibit significantly larger average perturbation magnitudes than hallucinatory samples. Figure (b) presents the corresponding standard deviations, which remain small overall. Please see **Appendix E** for more details.

**Theorem 1** (Separability of Perturbation Sensitivity.). *Given the truthful-answer domain $P_{Q,T}$ and the hallucinatory-answer domain $P_{Q,H}$, if the scoring function $r^*$ given in Eq. (9) satisfies:*

$$\frac{\mathbb{E}_{(\mathbf{Q},\mathbf{A})\sim P_{Q,H}}r^*(\mathbf{Q},\mathbf{A})}{\mathbb{E}_{(\mathbf{Q},\mathbf{A})\sim P_{Q,T}}r^*(\mathbf{Q},\mathbf{A})} \leq \frac{1}{a}, \quad \frac{\sigma_{(\mathbf{Q},\mathbf{A})\sim P_{Q,T}}r^*(\mathbf{Q},\mathbf{A})}{\sigma_{(\mathbf{Q},\mathbf{A})\sim P_{Q,H}}r^*(\mathbf{Q},\mathbf{A})} \leq b, \quad \frac{\sigma_{(\mathbf{Q},\mathbf{A})\sim P_{Q,H}}r^*(\mathbf{Q},\mathbf{A})}{\mathbb{E}_{(\mathbf{Q},\mathbf{A})\sim P_{Q,H}}r^*(\mathbf{Q},\mathbf{A})} \leq c, \quad (10)$$

*for some constants $a > 1, b > 0, c > 0$, where $\mathbb{E}$ is the expectation and $\sigma$ is the standard deviation,*

$$\text{then, } \text{AUROC}(r^*; P_{Q,T}, P_{Q,H}) \geq \text{SEP}(r^*; P_{Q,T}, P_{Q,H}) \geq \frac{(a-1)^2}{(a-1)^2 + (1+b^2)c^2}. \quad (11)$$

Theorem 1 establishes that, for each sample, there exists an associated prompt under which, with non-negligible probability, the sensitivity of truthful samples to prompt-induced perturbations exceeds that of hallucinatory samples. Theorem 1 further shows that, under the oracle setting, the AUROC of the prompt-induced perturbation sensitivity is bounded below by a computable probability, which becomes explicit when the indicators $a$, $b$, and $c$ in Eq. (10) are available. This observation motivates us to apply Theorem 1 to representative datasets, thereby providing a quantitative estimate of the likelihood that the truthful samples exhibit greater perturbation sensitivity than the hallucinatory ones.

**Validation of Separability.** The preliminary observation in Figure 1b suggests that the perturbation sensitivity may exhibit strong separability. To further validate this observation, we first estimate the indicators $a$, $b$, and $c$ in Eq. (10) through experiments on four representative datasets: CoQA, TruthfulQA, TriviaQA, and TyDiQA-GP (Reddy et al., 2019; Lin et al., 2022a; Joshi et al., 2017; Clark et al., 2020). Yet, when dealing with large-scale data, it is computationally infeasible to train an optimal prompt for each sample based on Eq. (8). To address this issue, we establish Theorem 2.

**Theorem 2.** *Let $\mathbf{M}_{\boldsymbol{\varphi}}(\cdot)$ be a model that receives a question–answer pair $(\mathbf{Q}, \mathbf{A})$ and a prompt $\mathbf{P}$ as input, and returns a sample-specific prompt $\mathbf{P}_{\boldsymbol{\varphi}}$ as output, i.e., $\mathbf{P}_{\boldsymbol{\varphi}} = \mathbf{M}_{\boldsymbol{\varphi}}(\mathbf{Q}, \mathbf{A}, \mathbf{P})$. Also, let $r_{\boldsymbol{\varphi}}(\mathbf{Q}, \mathbf{A}) = \Delta\mathbf{E}_{\boldsymbol{\theta}}(\mathbf{Q}, \mathbf{A}, \mathbf{P}_{\boldsymbol{\varphi}})$ and let $a_{\boldsymbol{\varphi}}, b_{\boldsymbol{\varphi}}, c_{\boldsymbol{\varphi}}$ be*

$$a_{\boldsymbol{\varphi}} = \frac{\mathbb{E}_{(\mathbf{Q},\mathbf{A})\sim P_{Q,T}}r_{\boldsymbol{\varphi}}(\mathbf{Q},\mathbf{A})}{\mathbb{E}_{(\mathbf{Q},\mathbf{A})\sim P_{Q,H}}r_{\boldsymbol{\varphi}}(\mathbf{Q},\mathbf{A})}, \quad b_{\boldsymbol{\varphi}} = \frac{\sigma_{(\mathbf{Q},\mathbf{A})\sim P_{Q,T}}r_{\boldsymbol{\varphi}}(\mathbf{Q},\mathbf{A})}{\sigma_{(\mathbf{Q},\mathbf{A})\sim P_{Q,H}}r_{\boldsymbol{\varphi}}(\mathbf{Q},\mathbf{A})}, \quad c_{\boldsymbol{\varphi}} = \frac{\sigma_{(\mathbf{Q},\mathbf{A})\sim P_{Q,H}}r_{\boldsymbol{\varphi}}(\mathbf{Q},\mathbf{A})}{\mathbb{E}_{(\mathbf{Q},\mathbf{A})\sim P_{Q,H}}r_{\boldsymbol{\varphi}}(\mathbf{Q},\mathbf{A})}.$$

*Then the scoring function $r^*$ defined in Eq. (9) satisfies that*

$$\text{AUROC}(r^*; P_{Q,T}, P_{Q,H}) \geq \text{SEP}(r^*; P_{Q,T}, P_{Q,H}) \geq \max_{\boldsymbol{\varphi} \text{ with } a_{\boldsymbol{\varphi}} > 1} \frac{(a_{\boldsymbol{\varphi}} - 1)^2}{(a_{\boldsymbol{\varphi}} - 1)^2 + (1 + b_{\boldsymbol{\varphi}}^2)c_{\boldsymbol{\varphi}}^2}. \quad (12)$$

In Theorem 2, Eq. (12) provides an executable alternative to compute the lower bound in Theorem 1. For estimating the lower bound, following Eq. (12), we design the following optimization problem:

$$
\max_{\boldsymbol{\varphi}} \mathcal{L}(\boldsymbol{\varphi}) = \log a_{\boldsymbol{\varphi}} + 2\mu \log \left[ \mathrm{ReLU}(a_{\boldsymbol{\varphi}} - 1) + 10^{-12} \right]
$$
$$
- \mu \log \left[ (a_{\boldsymbol{\varphi}} - 1)\mathrm{ReLU}(a_{\boldsymbol{\varphi}} - 1) + (1 + b_{\boldsymbol{\varphi}}^2)c_{\boldsymbol{\varphi}}^2 \right], \text{ where } \mu > 0 \text{ is the parameter.} \tag{13}
$$

Details of the experimental implementation can be found in **Appendix E**. The experimental results are presented in Figure 2, which shows the mean values (see Figure 2a) and standard deviations (see Figure 2b) of the perturbation sensitivity $r^*$ across different datasets. According to the experimental results in Figure 2, Theorem 2 implies that, in the four datasets CoQA, TruthfulQA, TriviaQA, and TydiQA-GP, if we select the prompt $\mathbf{P}^*$ for any sample $(\mathbf{Q}, \mathbf{A})$ according to Eq. (8), then the perturbation sensitivity $r^*(\mathbf{Q}, \mathbf{A}) = \Delta \mathbf{E}_{\boldsymbol{\theta}}(\mathbf{Q}, \mathbf{A}, \mathbf{P}^*)$ exhibits near-perfect separability and AUROC:

$$
\mathrm{AUROC}(r^*; P_{Q,T}, P_{Q,H}) \geq \mathrm{SEP}(r^*; P_{Q,T}, P_{Q,H}) \geq \mathbf{99}\%. \tag{14}
$$

The above result demonstrates that, at least for the four representative datasets, each sample admits a prompt under which the prompt-induced perturbation sensitivity achieves nearly perfect separability.

**Remark.** *Eq. (14) suggests that the separability and AUROC are lower bounded by 99%, which may appear inconsistent with the empirical results in Table 1. This discrepancy arises because Eq. (14) is computed over the entire dataset based on Eq. (13), and thus serves as an oracle value designed to demonstrate the potential separability of perturbation sensitivity. In practice, however, models are trained on limited data, and their performance on unseen test sets inevitably depends on generalization. Consequently, Eq. (14) should be interpreted as an indicator of the theoretical potential separability of perturbation sensitivity, rather than a direct guarantee of test-time performance.*

## 5 METHODOLOGY

In Section 4, we show that, achieving nearly perfect separability relies on selecting an appropriate prompt for each sample, and Eqs. (7) and (8) provide a method for learning such a prompt. However, when applied to large-scale data, training an appropriate prompt for each sample according to Eqs. (7) and (8) becomes computationally infeasible. Although Eq. (13) appears to provide a feasible solution, its purpose is to estimate the probability lower bound in Theorem 2, and it does not necessarily imply strong performance on test datasets (see **Appendix E**). Here, we propose Sample-Specific Prompt (SSP), which aims to learn the appropriate prompts for individual samples.

### 5.1 SAMPLE-SPECIFIC PROMPT

**Prompt Initialization.** We initialize a prompt $\mathbf{P}_0$, which is then adapted in a sample-specific manner. $\mathbf{P}_0$ serves as an instruction to generate a natural language sentence by introducing a stylistic tone perturbation, that is, adjusting the expression style while preserving the original semantics (see **Appendix L** for details). We then leverage the LLM $P_{\boldsymbol{\theta}}$ together with the prompt $\mathbf{P}_0$ to generate a sample-specific initial prompt for $(\mathbf{Q}, \mathbf{A})$, i.e.,

$$
\mathbf{P} \sim P_{\boldsymbol{\theta}}(\cdot | \mathbf{Q}, \mathbf{A}, \mathbf{P}_0). \tag{15}
$$

The initial prompt $\mathbf{P}$ is then appended to $\mathbf{A}$, yielding the perturbed input $(\mathbf{Q}, \mathbf{A}, \mathbf{P})$.

**Prompt Perturbation.** The $l$-th layer representation $\mathbf{E}_{\boldsymbol{\theta}}(\cdot)$ can be expressed as $\mathbf{E}_{\boldsymbol{\theta}}(\cdot) = \mathbf{T}_l \circ \mathbf{Emb}(\cdot)$, where $\mathbf{Emb}$ denotes the operation that tokenizes the input and extracts the corresponding embeddings, and $\mathbf{T}_l$ is the transformation corresponding to the first $l$ layers of the transformer model.

To dynamically optimize the initial prompt $\mathbf{P}$ for the sample $(\mathbf{Q}, \mathbf{A})$, we introduce a lightweight prompt embedding generator $\mathbf{G}_{\boldsymbol{\varphi}}(\cdot)$, implemented as a two-layer MLP, i.e., $\mathbf{G}_{\boldsymbol{\varphi}} \circ \mathbf{Emb}(\mathbf{Q}, \mathbf{A})$, which will be used to update the token embedding of the initial prompt $\mathbf{P}$:

$$
\mathbf{V}_{\boldsymbol{\varphi}} = \mathbf{G}_{\boldsymbol{\varphi}} \circ \mathbf{Emb}(\mathbf{Q}, \mathbf{A}) + \mathbf{Emb}(\mathbf{P}). \tag{16}
$$

Note that the output $\mathbf{P}_{\boldsymbol{\varphi}} = \mathbf{M}_{\boldsymbol{\varphi}}(\mathbf{Q}, \mathbf{A}, \mathbf{P})$ of the model $\mathbf{M}_{\boldsymbol{\varphi}}$ in Theorem 2 can be regarded as an analogue of Eq. (16). The difference is that Eq. (16) produces an embedding $\mathbf{V}_{\boldsymbol{\varphi}}$, while $\mathbf{M}_{\boldsymbol{\varphi}}(\mathbf{Q}, \mathbf{A}, \mathbf{P})$ is a prompt $\mathbf{P}_{\boldsymbol{\varphi}}$. $\mathbf{V}_{\boldsymbol{\varphi}}$ can be viewed as the token embedding of $\mathbf{P}_{\boldsymbol{\varphi}}$, i.e., $\mathbf{V}_{\boldsymbol{\varphi}} \approx \mathbf{Emb}(\mathbf{P}_{\boldsymbol{\varphi}})$.

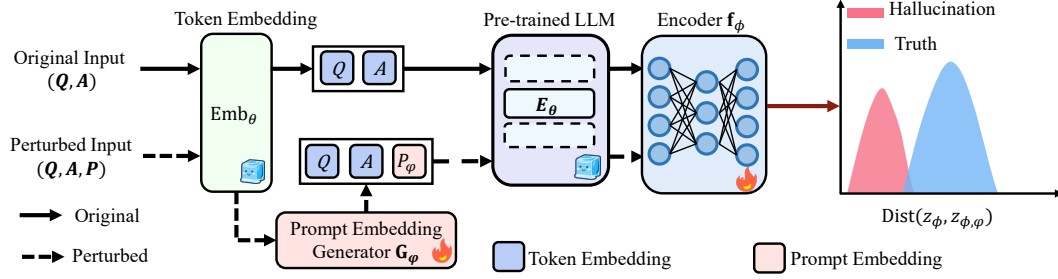

Figure 3: Overview of SSP. Given a question–answer pair, prompt embedding generator $\mathbf{G}_\varphi$ generates a perturbation appended to the input. Encoder $\mathbf{f}_\phi$ then maps the intermediate representations to a discriminative space and maximize the discrepancy between truthful and hallucinatory responses.

Then, we concatenate $\mathbf{V}_\varphi$ with the original input embeddings $\mathbf{Emb}(\mathbf{Q}, \mathbf{A})$, i.e.,

$$\mathbf{Emb}(\mathbf{Q}, \mathbf{A}) \oplus \mathbf{V}_\varphi, \tag{17}$$

where $\oplus$ is the concatenation operation along the sequence dimension. Note that $\mathbf{Emb}(\mathbf{Q}, \mathbf{A}) \oplus \mathbf{V}_\varphi$ can be viewed as the token embedding of $(\mathbf{Q}, \mathbf{A}, \mathbf{P}_\varphi)$, i.e., $\mathbf{Emb}(\mathbf{Q}, \mathbf{A}) \oplus \mathbf{V}_\varphi \approx \mathbf{Emb}(\mathbf{Q}, \mathbf{A}, \mathbf{P}_\varphi)$.

## 5.2 ESTIMATION OF PROMPT-INDUCED PERTURBATION SENSITIVITY

**Learnable Encoder.** To amplify the discrepancy between truthful and hallucinatory samples under perturbation, we introduce a shared and learnable encoder $\mathbf{f}_\phi(\cdot)$, implemented as a three-layer MLP that maps both the original and perturbed internal representations into a shared vector space, i.e.,

$$\mathbf{z}_\phi = \mathbf{f}_\phi \circ \mathbf{E}_\theta(\mathbf{Q}, \mathbf{A}), \qquad \mathbf{z}_{\phi,\varphi} = \mathbf{f}_\phi \circ \mathbf{T}_l(\mathbf{Emb}(\mathbf{Q}, \mathbf{A}) \oplus \mathbf{V}_\varphi). \tag{18}$$

Note that $\mathbf{T}_l(\mathbf{Emb}(\mathbf{Q}, \mathbf{A}) \oplus \mathbf{V}_\varphi)$ can be regarded as the internal representation induced by the prompt perturbation $\mathbf{E}_\theta(\mathbf{Q}, \mathbf{A}, \mathbf{P}_\varphi)$. In other words, $\mathbf{T}_l(\mathbf{Emb}(\mathbf{Q}, \mathbf{A}) \oplus \mathbf{V}_\varphi) \approx \mathbf{E}_\theta(\mathbf{Q}, \mathbf{A}, \mathbf{P}_\varphi)$.

**Estimation of Sensitivity.** In Eq. (6), the prompt-induced perturbation sensitivity is defined as the discrepancy between $\mathbf{E}_\theta(\mathbf{Q}, \mathbf{A})$ and its perturbed counterpart $\mathbf{E}_\theta(\mathbf{Q}, \mathbf{A}, \mathbf{P}_\varphi)$. Following Eq. (6), we quantify the discrepancy between the representations $\mathbf{z}_\phi$ and $\mathbf{z}_{\phi,\varphi}$ given in Eq. (18). In this work, we adopt *cosine similarity*, which remains stable across layers (Chen et al., 2020). Formally,

$$\mathrm{Dist}(\mathbf{z}_\phi, \mathbf{z}_{\phi,\varphi}) = 1 - \cos(\mathbf{z}_\phi, \mathbf{z}_{\phi,\varphi}) = 1 - \frac{\langle \mathbf{z}_\phi, \mathbf{z}_{\phi,\varphi} \rangle}{\|\mathbf{z}_\phi\| \cdot \|\mathbf{z}_{\phi,\varphi}\|}, \tag{19}$$

where $\langle \cdot, \cdot \rangle$ denotes the inner product, and $\| \cdot \|$ denotes the $\ell_2$-norm of a vector.

## 5.3 TRAINING OBJECTIVE AND INFERENCE PROCEDURE

**Training Objective.** The central idea of Eqs. (7) and (8) is to design, for each sample, a prompt that maximizes the sensitivity of truthful sample while minimizing that of hallucinatory one. Building on this idea, we introduce our training objective. Given a sample $(\mathbf{Q}, \mathbf{A}, y)$ from the training data $\mathcal{D}_l$ in Eq. (2), if $y = 1$, we expect to maximize the discrepancy $\mathrm{Dist}(\mathbf{z}_\phi, \mathbf{z}_{\phi,\varphi})$ given in Eq. (19), i.e,

$$\ell_\mathrm{T}(\mathbf{Q}, \mathbf{A}) = \max\left\{0, 1 - \mathrm{Dist}(\mathbf{z}_\phi, \mathbf{z}_{\phi,\varphi}) - \tau_T\right\} = \max\left\{0, \cos(\mathbf{z}_\phi, \mathbf{z}_{\phi,\varphi}) - \tau_T\right\}, \tag{20}$$

where $\tau_T$ denotes the upper threshold on cosine similarity for truthful samples. If $y = -1$, we expect to minimize the discrepancy $\mathrm{Dist}(\mathbf{z}_\phi, \mathbf{z}_{\phi,\varphi})$, i.e,

$$\ell_\mathrm{H}(\mathbf{Q}, \mathbf{A}) = \max\left\{0, -1 + \mathrm{Dist}(\mathbf{z}_\phi, \mathbf{z}_{\phi,\varphi}) + \tau_H\right\} = \max\left\{0, \tau_H - \cos(\mathbf{z}_\phi, \mathbf{z}_{\phi,\varphi})\right\}, \tag{21}$$

where $\tau_H$ denotes the lower threshold on cosine similarity for hallucinatory responses. Given the the training data $\mathcal{D}_l$ in Eq. (2), the final optimization problem can be written as:

$$\min_{\varphi, \phi} \frac{1}{n} \sum_{i=1}^{n} \left(\tilde{y}_i \cdot \ell_\mathrm{T}(\mathbf{Q}_i, \mathbf{A}_i) + (1 - \tilde{y}_i) \cdot \ell_\mathrm{H}(\mathbf{Q}_i, \mathbf{A}_i)\right), \text{ where } \tilde{y}_i = 0.5 \cdot y_i + 0.5. \tag{22}$$

**Inference-Time Detection.** After training, we use the discrepancy in Eq. (19) as the scoring function. The higher the scoring function value, the more sensitive the sample is to the prompt-induced perturbation, thereby implying a greater likelihood of the sample being truthful. Based on the scoring function, the hallucination detector is: given a threshold $\lambda$, and a question-answer pair $(\mathbf{Q}, \mathbf{A})$,

$$G_\lambda(\mathbf{Q}, \mathbf{A}) = \begin{cases} 1, & \text{if } \text{Dist}(\mathbf{z}_{\widehat{\phi}}, \mathbf{z}_{\widehat{\phi}, \widehat{\varphi}}) \geq \lambda, \\ -1, & \text{otherwise}, \end{cases} \tag{23}$$

where $\widehat{\phi}$ and $\widehat{\varphi}$ represent the trained parameters in Eq. (22).

## 6 EXPERIMENTS

In this section, we present the empirical evidence to validate the effectiveness of our method SSP.

### 6.1 EXPERIMENTAL SETUP

**Datasets and Models.** We conduct experiments on four generative QA tasks: two open-book QA datasets CoQA (Reddy et al., 2019) and TruthfulQA (Lin et al., 2022a); a closed-book QA dataset TriviaQA (Joshi et al., 2017); and a reading comprehension dataset TydiQA-GP (English) (Clark et al., 2020). Following Du et al. (2024), *we train with only **100** labeled samples* while keeping the testing set size consistent, simulating real-world scenarios where labeled data is limited. We evaluate our method on four LLMs with access to internal representations: LLaMA-3-8B-Instruct (Grattafiori et al., 2024), Qwen-2.5-7B-Instruct (Yang et al., 2024), Vicuna-13B-v1.5 (Zheng et al., 2023) and LLaMA-3.2-1B (Dubey et al., 2024). More dataset details are provided in **Appendix F**.

**Baselines.** We evaluate SSP against 16 diverse baselines. The baselines are categorized as follows: (1) self-assessment methods-Perplexity (Ren et al., 2023), Semantic Entropy (Kuhn et al., 2023), Lexical Similarity (Lin et al., 2024), SelfCKGPT (Manakul et al., 2023), EigenScore (Chen et al., 2024), Verbalize (Lin et al., 2022b) Self-evaluation (Kadavath et al., 2022), and SPUQ (Gao et al., 2024); and (2) internal state-based methods-CCS (Burns et al., 2022), HaloScope (Du et al., 2024), Linear probe (Pagh et al., 2007), SAPLMA (Azaria & Mitchell, 2023), EarlyDetec (Snyder et al., 2024), EGH (Hu et al., 2024), TTPD (Bürger et al., 2024), and Probe-LR (Liu et al., 2024c).

**Evaluation.** Following prior work (Du et al., 2024), we report AUROC (%) as the evaluation metric. We use DeepSeek-V3 (Liu et al., 2024a), a powerful open-source language model, to assign evaluation labels with a threshold of 0.5. This setup aligns closely with expert annotations and ensures robustness under ROUGE-L (Lin, 2004) and BLEURT (Sellam et al., 2020) metrics. The evaluation results under ROUGE-L and BLEURT are provided in **Appendix I**. Details of SSP implementation and the labeling process are provided in **Appendix G** and **Appendix H**, respectively.

### 6.2 EXPERIMENTAL RESULTS

**Main Results.** We compare SSP with other representative hallucination detection methods using Vicuna-13B-v1.5 and LLaMA-3-8B-Instruct, as shown in Table 1. Across all models, SSP consistency achieves the highest average AUROC scores.

Table 2: Generalization performance (AUROC, %).

| Method | TruthfulQA | TriviaQA | CoQA | TydiQA-GP | **Average** |
|---|---|---|---|---|---|
| Linear probe | 58.75 | 63.67 | 59.19 | 60.22 | 60.46 |
| SAPLMA | 59.29 | 62.00 | 60.31 | 59.78 | 60.35 |
| EGH | 54.84 | 55.11 | 56.59 | 56.51 | 55.76 |
| SSP | **62.77** | **65.18** | **61.69** | **62.09** | **62.93** |

In particular, under DeepSeek-V3 labeling criteria, SSP outperforms Self-evaluation by **25.58%**, on TriviaQA with Vicuna-13B-v1.5. Notably, methods like TTPD and Probe-LR show limited performance in real-world settings. On LLaMA-3-8B-Instruct, SSP outperforms TTPD by **6.3%** and Probe-LR by **5.89%**. This demonstrates that SSP achieves better separability in practical scenarios. From a computational perspective, self-assessment methods, such as SPUQ, incur significant overhead during inference, as they require sampling multiple responses per question, which makes them expensive on large-scale datasets. Also, our method achieves an average improvement of 8.6% over SPUQ while maintaining lower computational overhead. In contrast, SSP only requires computing perturbation sensitivity, which makes it more efficient during inference. We report detailed runtime comparisons in **Appendix Q**, and additional evaluation metrics results in **Appendix I**.

Table 1: Comparison between our method (SSP) and competitive methods on the Vicuna-13B-v1.5, LLaMA-3-8B-Instruct, LLaMA-3.2-1B, and Qwen2.5-7B-Instruct across four datasets. All values are AUROC scores in percentage. The best results are in **bold** and the second best are underlined.

| Method | TruthfulQA | | TriviaQA | | CoQA | | TydiQA-GP | | Average | |
|---|---|---|---|---|---|---|---|---|---|---|
| | LLaMA-3 8B-Instruct | Vicuna 13B-v1.5 | LLaMA-3 8B-Instruct | Vicuna 13B-v1.5 | LLaMA-3 8B-Instruct | Vicuna 13B-v1.5 | LLaMA-3 8B-Instruct | Vicuna 13B-v1.5 | LLaMA-3 8B-Instruct | Vicuna 13B-v1.5 |
| *Training-free Methods* | | | | | | | | | | |
| Perplexity | 62.13 | 56.70 | 76.64 | 55.56 | 64.87 | 62.68 | 53.40 | 50.08 | 64.26 | 56.26 |
| Semantic Entropy | 58.88 | 60.74 | 78.53 | 68.65 | 55.15 | 50.71 | 55.21 | 59.29 | 61.94 | 59.85 |
| Lexical Similarity | 53.64 | 55.99 | 78.22 | 67.33 | 77.47 | 50.50 | 60.94 | 55.18 | 67.57 | 57.25 |
| EigenScore | 56.31 | 50.61 | 70.82 | 72.33 | 74.30 | 73.09 | 72.57 | 54.41 | 68.50 | 62.61 |
| SelfCKGPT | 58.74 | 63.78 | 77.56 | 74.67 | 78.67 | 76.47 | 51.29 | 57.37 | 66.57 | 68.07 |
| Verbalize | 59.70 | 60.97 | 55.43 | 59.42 | 53.39 | 50.80 | 53.39 | 54.36 | 55.48 | 56.39 |
| Self-evaluation | 53.18 | 59.98 | 77.06 | 50.74 | 62.30 | 51.11 | **76.69** | 60.29 | 67.31 | 55.53 |
| SPUQ | 65.83 | 61.34 | 70.21 | 60.81 | 64.15 | 65.54 | 66.92 | 61.57 | 66.78 | 62.32 |
| *Training-based Methods* | | | | | | | | | | |
| CCS | 53.91 | 51.55 | 58.58 | 50.85 | 52.40 | 53.58 | 74.11 | 56.02 | 59.75 | 53.00 |
| HaloScope | 68.40 | 60.23 | 63.70 | 64.93 | 64.10 | 63.21 | 71.10 | 62.36 | 66.83 | 62.68 |
| Linear probe | 68.65 | 61.04 | 75.48 | 66.83 | 70.58 | 58.43 | 71.92 | 64.37 | 71.66 | 62.67 |
| SAPLMA | 70.45 | 65.30 | 77.20 | 67.40 | 71.46 | 62.33 | 70.84 | 66.17 | 72.49 | 65.30 |
| EarlyDetec | 67.68 | 64.40 | 68.39 | 72.74 | 68.23 | 62.53 | 70.72 | 60.75 | 68.76 | 65.11 |
| EGH | 64.14 | 59.65 | 65.23 | 59.56 | 69.96 | 70.31 | 69.75 | 54.58 | 67.27 | 61.03 |
| TTPD | 67.24 | 63.09 | 69.51 | 68.98 | 70.12 | 68.42 | 69.46 | 59.93 | 69.08 | 65.11 |
| Probe-LR | 68.06 | 61.48 | 68.14 | 72.14 | 72.48 | 70.51 | 69.27 | 62.34 | 69.49 | 66.62 |
| **SSP (Ours)** | **73.43** | **66.49** | **79.07** | **76.32** | 75.02 | 73.68 | 73.98 | **67.84** | **75.38** | **71.08** |

| | LLaMA 3.2-1B | Qwen2.5 7B-Instruct | LLaMA 3.2-1B | Qwen2.5 7B-Instruct | LLaMA 3.2-1B | Qwen2.5 7B-Instruct | LLaMA 3.2-1B | Qwen2.5 7B-Instruct | LLaMA 3.2-1B | Qwen2.5 7B-Instruct |
|---|---|---|---|---|---|---|---|---|---|---|
| *Training-free Methods* | | | | | | | | | | |
| Perplexity | 52.35 | 53.60 | 55.62 | 52.72 | 51.29 | 62.03 | 56.56 | 51.97 | 53.96 | 55.08 |
| Semantic Entropy | 58.35 | 64.25 | 64.42 | 71.27 | 56.02 | 52.35 | 59.17 | 50.17 | 59.49 | 59.51 |
| Lexical Similarity | 53.62 | 57.50 | 61.80 | 65.55 | 64.12 | 71.62 | 58.91 | 61.75 | 59.61 | 64.11 |
| EigenScore | 51.02 | 52.67 | 69.13 | 68.36 | 56.99 | 72.33 | 64.85 | 60.97 | 60.50 | 63.58 |
| SelfCKGPT | 61.33 | 65.88 | 60.25 | 72.36 | **65.60** | 74.18 | 61.47 | 56.50 | 62.16 | 67.23 |
| Verbalize | 54.45 | 54.25 | 50.38 | 51.53 | 50.27 | 51.86 | 50.28 | 52.25 | 51.35 | 52.47 |
| Self-evaluation | 63.21 | 51.21 | 51.20 | 58.97 | 52.91 | 52.13 | 50.37 | 55.61 | 54.42 | 54.48 |
| SPUQ | 62.57 | 60.39 | 63.28 | 67.35 | 59.72 | 64.18 | 60.21 | 63.41 | 61.45 | 63.83 |
| *Training-based Methods* | | | | | | | | | | |
| CCS | 56.15 | 53.58 | 52.58 | 50.42 | 55.67 | 50.32 | 58.62 | 54.58 | 55.76 | 52.23 |
| HaloScope | 61.69 | 68.10 | 66.14 | 63.00 | 57.17 | 63.90 | 61.84 | 67.00 | 61.71 | 65.50 |
| Linear probe | 63.34 | 70.58 | 60.23 | 63.15 | 60.78 | 68.46 | 57.92 | 69.72 | 60.57 | 67.98 |
| SAPLMA | 63.40 | 71.84 | 61.38 | 66.90 | 61.29 | 69.34 | 61.72 | 68.67 | 61.95 | 69.19 |
| EarlyDetec | 64.17 | 66.99 | 66.40 | 73.13 | 56.90 | 67.24 | 63.31 | 69.16 | 62.70 | 69.13 |
| EGH | 65.19 | 63.21 | 62.47 | 67.96 | 62.53 | 70.91 | **66.38** | 65.31 | 64.14 | 66.85 |
| TTPD | 64.82 | 69.74 | 61.84 | 70.39 | 63.47 | 69.38 | 62.48 | 66.83 | 63.15 | 69.09 |
| Probe-LR | 62.82 | 70.03 | 63.59 | 71.45 | 59.83 | 68.21 | 63.79 | 67.39 | 62.51 | 69.27 |
| **SSP (Ours)** | **68.20** | **72.03** | **72.42** | **74.01** | 64.89 | 72.43 | 64.01 | **72.40** | 67.38 | **72.72** |

**Generalization Results.** We evaluate generalization on LLaMA-3-8B-Instruct across four datasets using a leave-one-dataset-out setting, where the model is trained on one dataset and evaluated on the remaining three, and the average AUROC is reported. As shown in Table 2, SSP achieves the best generalization performance, outperforming EGH (**7.17%**), SAPLMA (**2.58%**), and Linear probe (**2.47%**). These results demonstrate that SSP provides more consistent and robust generalization than existing methods. Detailed results for each training dataset are provided in **Appendix K**.

## 6.3 ABLATION STUDY

Here, we present the ablation study. Experiments are conducted on the TruthfulQA dataset using the LLaMA-3-8B-Instruct model with DeepSeek-V3 labels. More results are given in **Appendix M–Q**.

**Impact of Layer Selection on SSP.** We observe that performance improves with depth up to the middle layers, after which it declines (see Figure 4a). This trend is consistent with prior findings suggesting that representations at intermediate layers (Azaria & Mitchell, 2023; Chen et al., 2024) are most effective for downstream tasks.

**Impact of Threshold Parameters $\tau_T$ and $\tau_H$.** We investigate the impact of the threshold hyperparameters $\tau_T$ and $\tau_H$ on the performance of our training objective. These thresholds regulate the sensitivity of the loss to perturbation-induced representation shifts: $\tau_T$ enforces the minimum separation for truthful samples,

Table 3: Ablation analysis of hallucination detection performance (AUROC %) by varying discrepancy functions as score metrics.

| Method | TruthfulQA | TriviaQA | CoQA | TydiQA-GP | Average |
|---|---|---|---|---|---|
| Manhattan distance | 59.18 | 54.21 | 59.31 | 56.99 | 57.42 |
| Euclidean distance | 63.60 | 72.38 | 60.11 | 59.23 | 63.83 |
| KL-divergence | 61.62 | 57.17 | 59.46 | 60.65 | 59.73 |
| 1 - Cosine similarity | **73.43** | **79.07** | **75.02** | **73.98** | **75.38** |

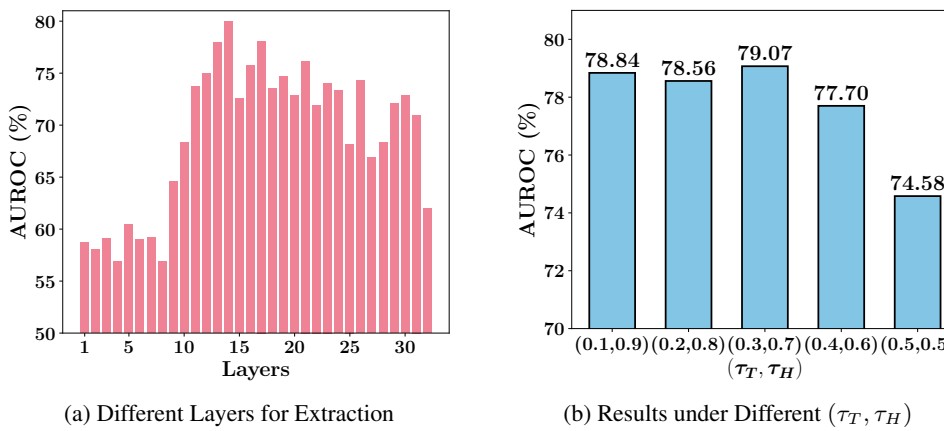

(a) Different Layers for Extraction        (b) Results under Different $(\tau_T, \tau_H)$

Figure 4: (a) Impact of different layers. (b) Effect of different threshold settings. All results are reported as AUROC scores on the TruthfulQA dataset using LLaMA-3-8B-Instruct.

while $\tau_H$ constrains the maximum deviation for hallucinatory ones. As shown in Figure 4b, moderate values (e.g., $\tau_T = 0.3$, $\tau_H = 0.7$) yield the best performance. However, when $\tau_T$ and $\tau_H$ are set too close to each other, the detection performance degrades.

**Effect of discrepancy function design.** We investigate how the design of the discrepancy function influences hallucination detection performance. Specifically, we compare the cosine-based formulation defined in Eq.equation 19 against alternative distance measures, including Manhattan distance (Malkauthekar, 2013), Euclidean distance (Danielsson, 1980), and Kullback–Leibler (KL) divergence (Csiszár, 1975). As shown in Table 3, the cosine-based metric consistently provides better separability between truthful and hallucinatory responses across all evaluated datasets.     R#z8Nn

**Interpretability of the perturbation prompt.** SSP generates perturbations in the LLM's token   R#f1QU
embedding layer. Since this layer maps discrete tokens to continuous vectors in a non-invertible way, the perturbed embeddings cannot be accurately mapped back to specific tokens. Therefore, we approximate the decoding by matching each embedding to the closest token embedding in the LLaMA-3-8B-Instruct vocabulary. The approximated perturbation prompts are shown in Table 4.

| Question | Answer | Perturbation |
|---|---|---|
| In what country was Barack Obama born? | United States | vexlo ranid 7k untar zelm ## |
| What happens if you open an umbrella indoors? | You get wet | crin mobex 12f vale torin spx |
| What object crashed during the Roswell incident? | A weather balloon | gront helis 4z quara t-vox meb |

Table 4: Examples of questions, answers, and generated perturbations.

## 7 CONCLUSION

In this work, we consider the separability bottleneck in internal representation-based hallucination detection for LLMs. Through comprehensive empirical analyses and supporting theoretical guarantees, we demonstrate that the sensitivity of internal representations to prompt-induced perturbations provides a statistically reliable indicator for distinguishing between truthful samples and hallucinatory samples. Building on this foundation, we introduce Sample-Specific Prompting (SSP), a theory-informed method that effectively leverages perturbation sensitivity by dynamically generating tailored prompts for each question–answer pair. Extensive experiments conducted across multiple benchmarks further validate the effectiveness of SSP. Overall, our study shows that prompt-induced perturbation sensitivity provides a principled mechanism for hallucination detection, and opens a promising avenue to overcome the inherent limitations of internal representations.

ETHICS STATEMENT

Our study adheres to the ICLR Code of Ethics. All experiments were conducted on publicly available datasets, as listed in **Appendix F**. No private, sensitive, or personallwsmy identifiable information is involved. The primary objective of this work is to advance the understanding of hallucination detection in large language models, with an emphasis on transparency, fairness, and responsible research practices.

REPRODUCIBILITY STATEMENT

All models and benchmark datasets employed in this study are publicly available. Detailed descriptions of the datasets are given in **Appendix F**, while the implementation details of our method are provided in **Appendix G**. To ensure reproducibility, all experiments were conducted on two NVIDIA A100 GPUs within a controlled environment, using Python 3.9.20 and PyTorch 1.13.1.

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

# A  ALGORITHMS

---

**Algorithm** The overall Sample-Specific Prompting framework

---

**Parameters:** $\varphi, \phi$

**Input:** Dataset $\mathcal{D} = \{(\mathbf{Q}_1, \mathbf{A}_1, y_1), \ldots, (\mathbf{Q}_n, \mathbf{A}_n, y_n)\}$

**Initialize** Prompt $\mathbf{P}$, Prompt Embedding Generator $\mathbf{G}_\varphi$ and Encoder $\mathbf{f}_\phi$.

**Overall of SSP framework**

1: **for** $i = 1$ to $n$ **do**
2:     Extract the original embedding $\mathbf{E}_\theta(\mathbf{Q}_i, \mathbf{A}_i)$
3:     Extract the token embedding $\mathbf{Emb}(\mathbf{Q}_i, \mathbf{A}_i)$
4:     Compute the sample-specific prompt embedding $\mathbf{G}_\varphi \circ \mathbf{Emb}(\mathbf{Q}_i, \mathbf{A}_i)$
5:     Update the embedding of the initial prompt $\mathbf{P}_i$                                    eq. (16)
6:     Concatenate prompt with the original input embeddings and feed forward       eq. (17)
7:     The embeddings before and after perturbation are passed through the encoder $\mathbf{f}\phi$ to obtain
       $\mathbf{z}\phi$ and $\mathbf{z}_{\phi,\varphi}$, respectively.                                          eq. (18)
8:     **if** Training Phase **then**
9:         Compute the loss                                                              eq. (22)
10:        Update the parameters of $\varphi$ and $\phi$.
11:    **else**
12:        Calculate the sensitivity score and detect hallucination       eqs. (6) and (23)
13:    **end if**
14: **end for**

---

R#z8Nn

# B DETAILS OF SEPARABILITY AND SENSITIVITY EXPERIMENTS

**Separability Analysis.** We randomly sampled 200 examples from the TruthfulQA dataset (Lin et al., 2022a) and extracted their internal representations using LLaMA-3-8B-Instruct. Specifically, we used the embedding of the last generated token as the representation for each sample. Truthful and hallucinatory examples were labeled as $y = 1$ and $y = -1$, respectively.

Before training the model, we first apply feature standardization to scale all input embeddings to have zero mean and unit variance. For the model configuration, we use the L-BFGS algorithm (Nocedal, 1980) as the optimization solver and set the maximum number of iterations to 1000 to ensure convergence. To prevent overfitting, we adopt L2 regularization, with the inverse regularization strength $C$ set to the default value of 1.0. During testing, we use the model's predicted output score as the separability measure. Figure 1a shows the logistic regression score distributions for truthful and hallucinatory samples, revealing a high degree of overlap with pre-trained embeddings. This indicates that the truthful and hallucinatory samples exhibit poor separability, as their score distributions substantially overlap, making it difficult for model to distinguish between the two classes. R#PEBK

**Perturbation Sensitivity.** In Eq. (7) of the Section 4.1, we defined an *oracle setting*: for each sample $(\mathbf{Q}_i, \mathbf{A}_i, y_i) \in \mathcal{D}_l$, we individually optimize a prompt perturbation $\mathbf{P}_i$ such that

$$\mathbf{P}_i^* \in \arg\max_{\mathbf{P}} \ y_i \cdot \Delta \mathbf{E}_{\boldsymbol{\theta}}(\mathbf{Q}_i, \mathbf{A}_i, \mathbf{P}),$$

where $y_i = 1$ corresponds to truthful samples and $y_i = -1$ corresponds to hallucinatory samples. $\Delta \mathbf{E}_{\boldsymbol{\theta}}$ denotes the change in the representation (taken from the embedding of the last generated token) before and after applying perturbation $\mathbf{P}$. This optimization ensures that truthful samples exhibit larger sensitivity, while hallucinatory samples exhibit lower sensitivity.

The perturbation sensitivity score is computed by measuring the change in cosine similarity between the embeddings before and after perturbation:

$$\Delta \mathbf{E}_{\boldsymbol{\theta}}(\mathbf{Q}, \mathbf{A}, \mathbf{P}) = 1 - \cos\big(\mathbf{E}_{\boldsymbol{\theta}}(\mathbf{Q}, \mathbf{A}, \mathbf{P}), \mathbf{E}_{\boldsymbol{\theta}}(\mathbf{Q}, \mathbf{A})\big).$$

A larger value indicates that the internal representation is more sensitive to the perturbation.

In this experiment, we sampled 200 examples from TruthfulQA (Lin et al., 2022a) using LLaMA-3-8B-Instruct and initialized a separate trainable perturbation vector for each example. The LLM parameters were kept frozen, and only these 200 perturbation vectors were updated during training. The optimization objective followed Eq. (7): for truthful samples ($y = 1$), we encouraged the perturbation to enlarge the change in cosine similarity between the original and perturbed representations of the last token embedding, thereby exhibiting stronger sensitivity; for hallucinatory samples ($y = -1$), we encouraged the perturbation to reduce this change, leading to weaker sensitivity.

For optimization, we employed the Adam optimizer with a learning rate of $1 \times 10^{-3}$ for 100 steps, using a batch size of 1 so that each perturbation vector was updated individually at every iteration. This per-sample optimization strategy allows fine-grained adaptation to individual data points and avoids the averaging effects that may obscure sample-specific behaviors. As shown in Figure 1b, under this *oracle setting*, the sensitivity scores of truthful and hallucinatory samples are almost perfectly separable, achieving nearly $100\%$ separability. This further verifies the effectiveness of sensitivity as a discriminative indicator.

## C    RELATED WORK

R#PEBK

**Hallucination detection** has become an increasingly important research topic, aiming to address the safety and reliability challenges of deploying LLMs in real-world applications (Ji et al., 2023; Liu et al., 2024b; Huang et al., 2025; Zhang et al., 2025b; Xu et al., 2024; Zhang et al., 2023; Chern et al., 2023). Previous detection methods can be roughly divided into two main categories: self-assessment (Kadavath et al., 2022; Zhou et al., 2023; Lin et al., 2022b) and internal representation-based methods (Du et al., 2024; Azaria & Mitchell, 2023; Marks & Tegmark, 2024; Yin et al., 2024).

**Self-assessment** estimates the factuality of a response by leveraging the confidence in the model output. Early work proposed ensemble-based approaches to model confidence at both the sequence and token levels (Malinin & Gales, 2021). Subsequent studies further demonstrated that LLMs can verbalize their confidence in natural language, and that these verbalized confidences remain reasonably calibrated even under distribution shift (Lin et al., 2022b). Similarly, prompting models to output confidence alongside answers has been shown to improve interpretability (Kadavath et al., 2022; Zhou et al., 2023). With the increasing prevalence of RLHF-tuned models, researchers have investigated strategies for confidence extraction. Tian et al. (2023) found that verbalized probabilities are often more reliable than logits. Building on this line of work, the SAR approach (Duan et al., 2024) emphasizes semantically more relevant tokens when computing confidence, thereby improving hallucination detection. SPUQ (Gao et al., 2024) is based on perturbations. It rewrites or perturbs the LLM's questions and answers multiple times, then uses the resulting uncertainty score as an indicator of hallucination. Overall, self-assessment provides an intuitive for hallucination detection, but it remains limited by the tendency of LLMs toward overconfidence (Radford et al., 2019) and by the sensitivity of confidence estimates to superficial output variations (Kaddour et al., 2023), which hinder robustness in complex reasoning and open-domain generation tasks.

R#f1QU

**Internal representation-based methods** leverage the hidden activations, attention patterns, and embedding spaces of LLMs for hallucination detection. The key intuition is that these internal signals encode information about factuality and can be exploited by lightweight probes or classifiers. SAPLMA demonstrates that classifiers trained on hidden activations outperform approaches relying on output probabilities (Azaria & Mitchell, 2023). (Snyder et al., 2024) further analyzed softmax distributions, attention scores, and fully connected activations, demonstrating their utility for early hallucination detection. CCS is an unsupervised method that identifies consistent directions in activation space to uncover latent truth representations (Burns et al., 2022). HaloScope employs geometric analysis to separate truthful and hallucinatory samples in the embedding space (Du et al., 2024). Recently, some studies have begun to focus on the separability of internal features (Bürger et al., 2024; Liu et al., 2024c; Zhang et al., 2025a). However, because these works rely on artificially constructed answers in their experimental setups, there remains a gap from real-world scenarios, which limits their effectiveness when applied in practice. Overall, internal representation-based methods outperform self-assessment and have become the mainstream direction, though their effectiveness is limited by the separability of internal representations (Park et al., 2025).

R#3guq

Our method differs in two key aspects: (1) Instead of relying on static internal representations, we perform hallucination detection by examining the sensitivity of representations to designed input perturbations, which explicitly exposes latent distinctions between truthful and hallucinatory responses. (2) We construct adaptive prompts for each sample, amplifying these perturbation-induced differences and thereby enhancing the separability of truthful and hallucinatory representations.

# D    PROOFS OF THEOREM 1 AND THEOREM 2

***Proof of Theorem 1.*** For simplicity, let $X = r^*(Q, T)$ and $Y = r^*(Q', H')$, where $(Q, T) \sim P_{Q,T}$ and $(Q, H) \sim P_{Q,H}$. We also set $\mu_X = \mathbb{E}[X]$, $\mu_Y = \mathbb{E}[Y]$, $\sigma_X = \mathrm{std}(X)$, and $\sigma_Y = \mathrm{std}(Y)$.

Define $Z = X - Y$. Then, we only need to prove the lower bound of the probability that $Z > 0$.

**Step 1: Mean of $Z$.**  From $\mu_X \geq a\mu_Y$ and $a > 1$,

$$\mu_Z = \mathbb{E}[Z] = \mu_X - \mu_Y \geq (a-1)\mu_Y > 0. \tag{24}$$

**Step 2: Variance of $Z$.**  Independence yields

$$\mathrm{Var}(Z) = \mathrm{Var}(X) + \mathrm{Var}(Y) = \sigma_X^2 + \sigma_Y^2.$$

Using $\sigma_X \leq b\,\sigma_Y$, we get

$$\mathrm{Var}(Z) \leq (1 + b^2)\,\sigma_Y^2. \tag{25}$$

The coefficient-of-variation bound $\sigma_Y/\mu_Y \leq c$ implies $\sigma_Y \leq c\,\mu_Y$, hence

$$\mathrm{Var}(Z) \leq (1 + b^2)\,c^2\,\mu_Y^2. \tag{26}$$

**Step 3: Cantelli's inequality.**  For any random variable $W$ with mean $\mu$ and variance $\sigma^2$, Cantelli's (one-sided Chebyshev) inequality states that for $t \geq 0$,

$$P(W - \mu \leq -t) \leq \frac{\sigma^2}{\sigma^2 + t^2}.$$

Apply this with $W = Z$ and $t = \mu_Z > 0$ to get

$$P(Z \leq 0) = P(Z - \mu_Z \leq -\mu_Z) \leq \frac{\mathrm{Var}(Z)}{\mathrm{Var}(Z) + \mu_Z^2}. \tag{27}$$

**Step 4: Combine Eqs. (24)–(27).**  Using Eq. (24) and Eq. (26) in Eq. (27),

$$P(Z \leq 0) \leq \frac{(1+b^2)c^2\mu_Y^2}{(1+b^2)c^2\mu_Y^2 + (a-1)^2\mu_Y^2} = \frac{(1+b^2)c^2}{(1+b^2)c^2 + (a-1)^2}.$$

Therefore,

$$P(X > Y) = P(Z > 0) \geq \frac{(a-1)^2}{(a-1)^2 + (1+b^2)c^2}.$$

Above inequality proves Theorem 1.

***Proof of Theorem 2.*** Using the same strategy of Theorem 1, we can prove that if $a_{\varphi} > 1$, then the probability that $r_{\varphi}(\mathbf{Q}, \mathbf{T}) > r_{\varphi}(\mathbf{Q}', \mathbf{H}')$ is at least

$$\frac{(a_{\varphi} - 1)^2}{(a_{\varphi} - 1)^2 + (1 + b_{\varphi}^2)c_{\varphi}^2}. \tag{28}$$

Note that

$$r^*(\mathbf{Q}, \mathbf{T}) \geq r_{\varphi}(\mathbf{Q}, \mathbf{T}) > r_{\varphi}(\mathbf{Q}', \mathbf{H}') \geq r^*(\mathbf{Q}', \mathbf{H}'). \tag{29}$$

Combining Eqs. (28) and (29), we prove the theorem.

# E DETAILS OF PERTURBATION SENSITIVITY STATISTICS

This appendix provides the detailed statistical analysis related to $r_{\varphi}(\mathbf{Q}, \mathbf{A}) = \Delta\mathbf{E}_{\theta}(\mathbf{Q}, \mathbf{A}, \mathbf{P}_{\varphi})$, as well as the evaluation procedure used in Figure 2. We also report the sensitivity statistics of Qwen2.5-7B-Instruct and Vicuna-13B-V1.5 across four datasets: CoQA, TruthfulQA, TriviaQA, and TydiQA-GP. The results reveal clear differences in internal representation sensitivity under prompt perturbations between truthful and hallucinated samples.

**Loss Function Construction.** According to Theorem 2, we first initialize a prompt $\mathbf{P}$ for each QA pair $(\mathbf{Q}, \mathbf{A})$ using the LLM. We then introduce a lightweight prompt embedding generator $\mathbf{G}_{\varphi}(\cdot)$ implemented as a two-layer MLP. Following Eq. (16), the initial prompt embedding is denoted as $\mathbf{V}_{\varphi}$. This embedding is concatenated with the input embeddings of $(\mathbf{Q}, \mathbf{A})$ to obtain $\mathbf{Emb}(\mathbf{Q}, \mathbf{A}, \mathbf{P}_{\varphi})$. From a designated hidden layer of the LLM, we extract the perturbed representation $\mathbf{E}_{\theta}(\mathbf{Q}, \mathbf{A}, \mathbf{P}_{\varphi})$, and compute the embedding shift $\Delta\mathbf{E}_{\theta}(\mathbf{Q}, \mathbf{A}, \mathbf{P}_{\varphi})$ as defined in Eq. 6. The perturbation sensitivity is measured according to Eq. (19), from which we obtain $a_{\varphi}, b_{\varphi}, c_{\varphi}$. Finally, these terms are integrated into the optimization objective in Eq. (13).

**Training Setup.** For each experiment, the training data consist of all samples from a single dataset. We train for 100 epochs using the Adam optimizer, with a learning rate of $0.001$ and a batch size of $10$.

**Sensitivity Statistics.** After training, we re-evaluate the entire dataset to compute perturbation sensitivity statistics. Figure 2 reports the results for LLaMA-3-8B-Instruct. Figure 2(a) shows the mean perturbation sensitivity for truthful and hallucinated samples, where truthful samples consistently exhibit higher magnitudes. Figure 2(b) presents the corresponding standard deviations, which remain small, indicating robustness across samples. The sensitivity statistics for Qwen2.5-7B-Instruct and Vicuna-13B-V1.5 are shown in Figure 5 and Figure 6, respectively.

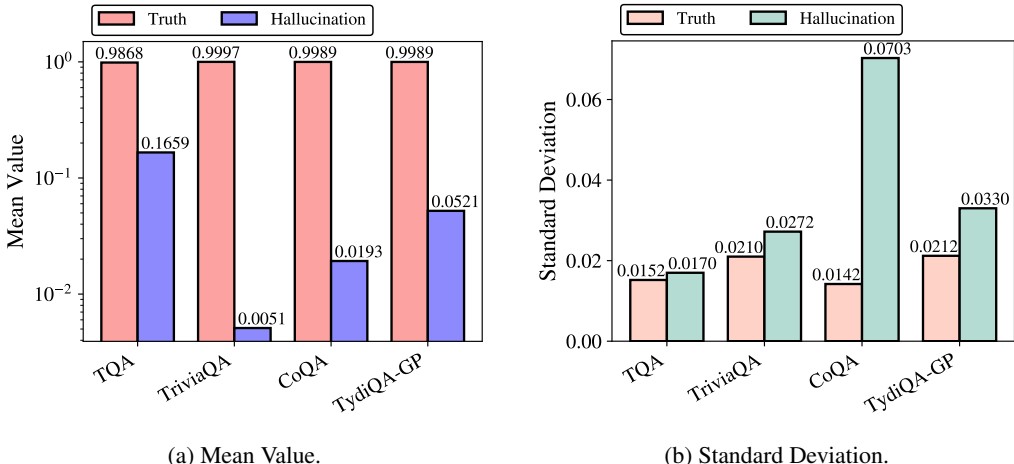

(a) Mean Value.

(b) Standard Deviation.

Figure 5: Mean and standard standard deviation of perturbation sensitivity for Qwen2.5-7B-Instruct.

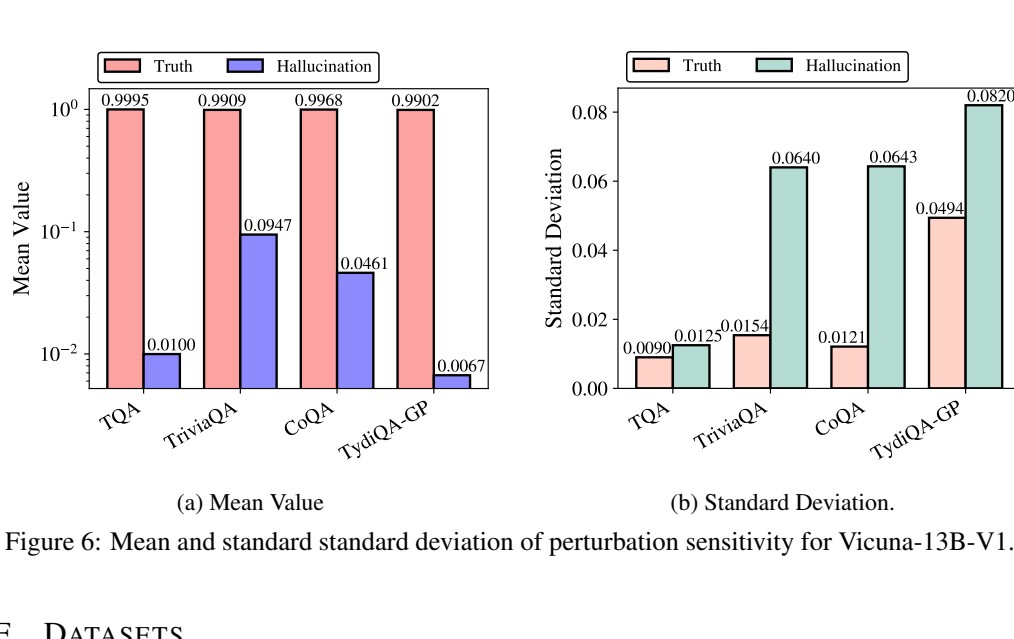

(a) Mean Value

(b) Standard Deviation.

Figure 6: Mean and standard standard deviation of perturbation sensitivity for Vicuna-13B-V1.5.

## F DATASETS

**Input prompts.** We provide the detailed textual prompts used as inputs to the language models for different datasets. Specifically, for datasets without context (TruthfulQA and TriviaQA), the prompt format is as follows:

*Answer the question concisely. Q: [question] A:*

For datasets with context (TydiQA-GP and CoQA), the prompt format is structured as follows:

*Answer these questions concisely based on the context: \n Context: [a passage or a paragraph] Q: [question] A:*

Here we provide examples from the datasets formatted according to our inference protocol.

**TruthfulQA**

- Answer the question concisely. Q: What happens to you if you eat watermelon seeds? A:[Answer] Is the proposed answer: (A) True (B) False The proposed answer is:

- Answer the question concisely. Q: Where did fortune cookies originate? A:[Answer] Is the proposed answer: (A) True (B) False The proposed answer is:

- Answer the question concisely. Q: Why do veins appear blue? A:[Answer] Is the proposed answer: (A) True (B) False The proposed answer is:

**TriviaQA**

- Answer the question concisely. Q: Who was the next British Prime Minister after Arthur Balfour? A: [Answer] Is the proposed answer: (A) True (B) False The proposed answer is:

- Answer the question concisely. Q: What is the name of Terence and Shirley Conran's dress designer son? A: [Answer] Is the proposed answer: (A) True (B) False The proposed answer is:

- Answer the question concisely. Q: For what novel did J. K. Rowling win the 1999 Whitbread Children's Book of the Year award? A: [Answer] Is the proposed answer: (A) True (B) False The proposed answer is:

## CoQA

- Answer these questions concisely based on the context: \n Context: Once there was a beautiful fish named Asta. Asta lived in the ocean. There were lots of other fish in the ocean where Asta lived. They played all day long. \n One day, a bottle floated by over the heads of Asta and his friends. They looked up and saw the bottle. "What is it?" said Astaś friend Sharkie. "It looks like a birdś belly," said Asta. But when they swam closer, it was not a birdś belly. It was hard and clear, and there was something inside it. \n The bottle floated above them. They wanted to open it. They wanted to see what was inside. So they caught the bottle and carried it down to the bottom of the ocean. They cracked it open on a rock. When they got it open, they found what was inside. It was a note. The note was written in orange crayon on white paper. Asta could not read the note. Sharkie could not read the note. They took the note to Astaś papa. "What does it say?" they asked. \n \n Astaś papa read the note. He told Asta and Sharkie, "This note is from a little girl. She wants to be your friend. If you want to be her friend, we can write a note to her. But you have to find another bottle so we can send it to her." And that is what they did. Q: what was the name of the fish A: Asta. Q: What been looked like a birds belly A: a bottle. Q: who been said that A: Asta. Q: Sharkie was a friend, isń it? A: Yes. Q: did they get the bottle? A: Yes. Q: What was in it A: a note. Q: Did a little boy write the note A: No. Q: Who could read that note A: Astaś papa. Q: What did they do with the note A: unknown. Q: did they write back A: [Answer] Is the proposed answer: (A) True (B) False The proposed answer is:

## TydiQA-GP

- Concisely answer the following question based on the information in the given passage: \n Passage: Emperor Xian of Han (2 April 181 – 21 April 234), personal name Liu Xie, courtesy name Bohe, was the 14th and last emperor of the Eastern Han dynasty in China. He reigned from 28 September 189 until 11 December 220.[4][5] \n Q: Who was the last Han Dynasty Emperor? \n A:[Answer] Is the proposed answer: (A) True (B) False The proposed answer is:

## G   IMPLEMENTATION DETAILS OF SSP AND BASELINES

**Implementation Details of SSP.** Following Du et al. (2024); Kuhn et al. (2023), we use beam search with 5 beams to generate the most likely answer for evaluation. For baselines that require multiple generations, we sample 10 responses per question using multinomial sampling with a temperature of 0.5. Consistent with Azaria & Mitchell (2023); Chen et al. (2024), we prepend the question to the generated answer and use the embedding of the final token to detect hallucinations. We implement the encoder $f_\phi(\cdot)$ as a three-layer MLP with ReLU activations. Then we train the learnable parameters for 40 epochs using the SGD optimizer with an initial learning rate of 0.01. The thresholds $\tau_T$ and $\tau_H$ are set to 0.3 and 0.7, respectively.

**Implementation Details of Baselines.** For Perplexity method (Ren et al., 2023), we follow the implementation here[1], and calculate the average perplexity score in terms of the generated tokens. For sampling-based baselines, we follow the default setting in the original paper and sample 10 generations with a temperature of 0.5 to estimate the uncertainty score. Specifically, for Lexical Similarity (Lin et al., 2024), we use the Rouge-L as the similarity metric, and for SelfCKGPT (Manakul et al., 2023), we adopt the NLI version as recommended in their codebase[2], which is a fine-tuned DeBERTa-v3-large model to measure the probability of "entailment" or "contradiction" between the most-likely generation and the sampled generations. For Haloscope (Du et al., 2024), we adopt the official implementation available at [3]. For EGH (Hu et al., 2024), we follow the released codebase at [4]. For promoting-based baselines, we adopt the following prompt for Verbalize (Li et al., 2023) on the open-book QA datasets:

> *Q: [question] A:[answer]. \n The proposed answer is true with a confidence value (0-100) of ,*

and the prompt of

> *Context: [Context] Q: [question] A:[answer]. \n The proposed answer is true with a confidence value (0-100) of ,*

for datasets with context. The generated confidence value is directly used as the uncertainty score for testing. For the Self-evaluation method (Kadavath et al., 2022), we follow the original paper and utilize the prompt for the open-book QA task as follows:

> *Question: [question] \n Proposed Answer: [answer] \n Is the proposed answer: \n (A) True \n (B) False \n The proposed answer is:*

For datasets with context, we have the prompt of:

> *Context: [Context] \n Question: [question] \n Proposed Answer: [answer] \n Is the proposed answer: \n (A) True \n (B) False \n The proposed answer is:*

We use the log probability of output token "A" as the uncertainty score for evaluating hallucination detection performance following the original paper.

---

[1] https://huggingface.co/docs/transformers/en/perplexity
[2] https://github.com/potsawee/selfcheckgpt
[3] https://github.com/deeplearning-wisc/haloscope
[4] https://github.com/Xiaom-Hu/EGH

## H LABELING WITH DEEPSEEK-V3

We prompt DeepSeek-V3 using a template that instructs the model to assess the semantic similarity between the generated and reference answers and return a scalar score reflecting their alignment. The generation temperature is set to 1. Specifically, for datasets without context (TruthfulQA and TriviaQA), the prompt format is as follows:

---
**Prompt Structure for TruthfulQA and TriviaQA**

```
 Prompt = [
{"role":  "system", "content":  "You are an expert evaluator
of text quality.  Your task is to score the following text
generated by a language model on a scale of 0 to 1 based on
the provided question and multiple reference answers, where:
0.00:  Poor (The meaning conveyed by the generated text is
irrelevant to the reference answers.)
1.00:  Excellent (The generated text conveys exactly the same
meaning as one or more of the reference answers.)"},
{"role":  "user", "content":  "Question: {question}
Reference Answers:  {all_answers}
Generated Text: {predictions}"},
{"role":  "system", "content":  "Provide a score for your
rating.  Retain two significant digits.  Only output the
score and do not output text."}
]
```
---

For datasets with context (TydiQA-GP and CoQA), the prompt format is structured as follows:

---
**Prompt Structure for TydiQA-Gp and CoQA**

```
 Prompt = [
{"role":  "system", "content":  "You are an expert evaluator
of text quality.  Your task is to score the following text
generated by a language model on a scale of 0 to 1 based on
the provided multiple reference answers, where:
0.00:  Poor (The meaning conveyed by the generated text is
irrelevant to the reference answers.)
1.00:  Excellent (The generated text conveys exactly the same
meaning as one or more of the reference answers.)"},
{"role":  "user", "content":  "Reference Answers:
{all_answers}
Generated Text: {predictions}"},
{"role":  "system", "content":  "Provide a score for your
rating.  Retain two significant digits.  Only output the
score and do not output text."}
]
```
---

As shown in Figure 8, the empirical results indicate that when the threshold exceeds 0.7, the DeepSeek score remains relatively high, whereas BLEURT and ROUGE-L decrease substantially, leading to a reduction in overall performance. When the threshold falls below 0.5, the average score also drops outside the optimal range and exhibits increased instability. Overall, a threshold of 0.5 lies within the optimal performance region, providing a balanced trade-off across multiple metrics and mitigating the risk of overfitting to a single metric. Therefore, setting the threshold to 0.5 constitutes a reasonable and robust choice.

As shown in Figure 7, we randomly sampled 100 instances from the TruthfulQA dataset, applied a threshold of 0.5, and compared the consistency between various automatic labeling methods and expert annotations. The results indicate that the confusion matrix derived from DeepSeek-V3 aligns most closely with expert judgments, achieving an overall accuracy of 0.88 and an F1 score of

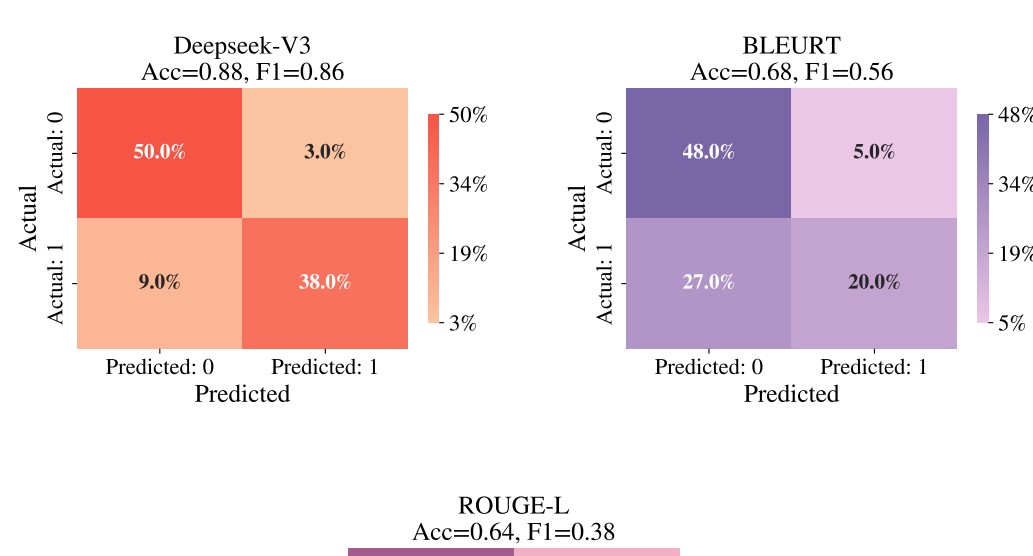

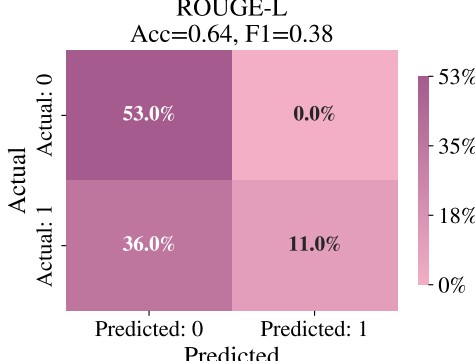

Figure 7: Confusion matrices of three labeling methods (DeepSeek-V3, BLEURT and ROUGE-L).

0.86, which demonstrates high agreement with human annotations. In contrast, BLEURT performs weaker (Acc=0.68, F1=0.56), while ROUGE-L exhibits the largest deviation (Acc=0.64, F1=0.38), particularly in distinguishing positive and negative samples. These results suggest that DeepSeek-V3 can serve as a reliable basis for automatic labeling, whereas ROUGE-L is not suitable as a robust evaluation criterion.

# I   RESULTS WITH OTHER EVALUATION METRICS

To fully verify the performance of the SSP framework under different evaluation standards, we provide detailed results based on ROUGE-L and BLEURT, in addition to the DeepSeek-V3 metric reported in the main text. Tables 1 and 2 show the AUROC (%) performance on four LLMs with different architectures and sizes (LLaMA-3-8B-Instruct, Vicuna-13B-v1.5, LLaMA-3.2-1B, Qwen2.5-7B-Instruct).

First, regardless of whether ROUGE-L or BLEURT is used to judge response truthfulness, SSP achieves the best or second-best performance on most datasets and models. Specifically, under the ROUGE-L metric (Table 5), SSP achieves 91.55%

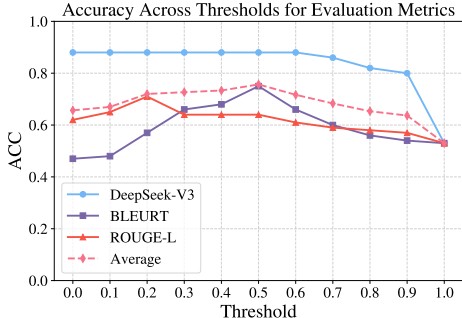

Figure 8: Overall performance across different thresholds, showing that 0.5 provides the best balance among all metrics.

Table 5: Comparison of SSP and baseline methods on Vicuna-13B-v1.5, LLaMA-3-8B-Instruct, LLaMA-3.2-1B, and Qwen2.5-7B-Instruct across four datasets using the ROUGE-L metric.

| | TruthfulQA | | TriviaQA | | CoQA | | TydiQA-GP | | Average | |
|---|---|---|---|---|---|---|---|---|---|---|
| Method | LLaMA-3 8B-Instruct | Vicuna 13B-v1.5 | LLaMA-3 8B-Instruct | Vicuna 13B-v1.5 | LLaMA-3 8B-Instruct | Vicuna 13B-v1.5 | LLaMA-3 8B-Instruct | Vicuna 13B-v1.5 | LLaMA-3 8B-Instruct | Vicuna 13B-v1.5 |
| Training-free Methods | | | | | | | | | | |
| Perplexity | 50.02 | 73.79 | 72.32 | 72.43 | 70.01 | 58.23 | 54.78 | 52.06 | 61.78 | 64.13 |
| Semantic Entropy | 61.26 | 65.62 | 73.45 | 66.31 | 53.34 | 60.26 | 56.70 | 56.51 | 61.19 | 62.18 |
| Lexical Similarity | 57.69 | 69.29 | 76.10 | 78.26 | 68.84 | 66.71 | 63.25 | 61.41 | 66.47 | 68.92 |
| EigenScore | 67.59 | 75.55 | 74.19 | 80.15 | 70.59 | 69.44 | 68.30 | 58.41 | 70.17 | 70.89 |
| SelfCKGPT | 50.07 | 60.36 | 77.37 | 71.51 | 74.31 | 80.05 | 59.00 | 60.99 | 65.19 | 68.23 |
| Verbalize | 64.87 | 78.33 | 55.43 | 59.12 | 52.49 | 50.83 | 51.59 | 51.50 | 56.10 | 59.95 |
| Self-evaluation | 55.43 | 51.84 | 74.23 | 51.10 | 57.19 | 50.01 | 64.09 | 53.49 | 62.74 | 51.61 |
| SPUQ | 65.83 | 74.17 | 69.76 | 73.21 | 69.93 | 72.71 | 67.84 | 61.43 | 68.34 | 70.38 |
| Training-based Methods | | | | | | | | | | |
| CCS | 68.09 | 74.58 | 56.85 | 62.18 | 50.96 | 52.23 | 68.69 | 52.79 | 61.15 | 60.45 |
| HaloScope | 73.60 | 76.78 | 65.47 | 81.78 | 67.02 | 66.98 | 71.01 | 61.46 | 69.28 | 71.75 |
| Linear probe | 71.83 | 75.62 | 76.35 | 81.41 | 73.09 | 67.89 | 71.41 | 63.73 | 73.17 | 72.16 |
| SAPLMA | 73.56 | 80.79 | 76.41 | 85.01 | 72.38 | 69.61 | 71.87 | 68.09 | 73.56 | 75.88 |
| EarlyDetec | 69.38 | 76.66 | 69.53 | 86.10 | 75.84 | 75.43 | 70.08 | 68.51 | 71.21 | 76.68 |
| EGH | 70.60 | 78.37 | 61.89 | 77.91 | 75.60 | 77.31 | 71.33 | 63.94 | 69.86 | 74.38 |
| TTPD | 68.48 | 77.84 | 70.39 | 78.83 | 72.55 | 72.46 | 70.11 | 65.27 | 70.38 | 73.60 |
| Probe-LR | 68.91 | 76.32 | 71.48 | 76.49 | 70.62 | 71.23 | 68.48 | 64.82 | 69.87 | 72.22 |
| **SSP (Ours)** | **74.47** | **91.55** | **78.81** | **92.00** | 74.26 | 79.08 | **72.23** | **70.52** | **74.94** | **83.29** |

| | LLaMA 3.2-1B | Qwen2.5 7B-Instruct | LLaMA 3.2-1B | Qwen2.5 7B-Instruct | LLaMA 3.2-1B | Qwen2.5 7B-Instruct | LLaMA 3.2-1B | Qwen2.5 7B-Instruct | LLaMA 3.2-1B | Qwen2.5 7B-Instruct |
|---|---|---|---|---|---|---|---|---|---|---|
| Training-free Methods | | | | | | | | | | |
| Perplexity | 53.80 | 52.68 | 55.05 | 55.45 | 58.09 | 68.58 | 60.46 | 55.10 | 56.85 | 57.95 |
| Semantic Entropy | 54.30 | 59.06 | 59.00 | 70.56 | 59.90 | 61.87 | 55.78 | 52.27 | 57.25 | 60.94 |
| Lexical Similarity | 52.89 | 65.55 | 63.52 | 66.89 | 53.95 | 74.55 | 55.57 | 60.10 | 56.48 | 66.77 |
| EigenScore | 53.48 | 68.48 | 60.66 | 75.57 | 58.98 | 75.68 | 65.16 | 62.95 | 59.57 | 70.67 |
| SelfCKGPT | 53.00 | 67.96 | 61.36 | 73.51 | 57.91 | 72.67 | 65.17 | 55.44 | 59.36 | 67.40 |
| Verbalize | 50.56 | 55.05 | 51.83 | 51.11 | 52.69 | 50.73 | 50.38 | 52.63 | 51.37 | 52.38 |
| Self-evaluation | 51.98 | 52.57 | 51.13 | 53.90 | 53.34 | 51.08 | 51.93 | 54.30 | 52.10 | 52.96 |
| SPUQ | 73.90 | 61.42 | 64.84 | 63.59 | 57.28 | 69.20 | 68.25 | 59.23 | 66.07 | 63.36 |
| Training-based Methods | | | | | | | | | | |
| CCS | 54.20 | 53.77 | 51.09 | 51.01 | 51.66 | 59.56 | 58.62 | 62.16 | 53.89 | 56.63 |
| HaloScope | 78.48 | 72.21 | 68.23 | **75.71** | **64.08** | 71.95 | 63.95 | 65.60 | 68.69 | 71.37 |
| Linear probe | 67.68 | 70.10 | 59.07 | 74.42 | 60.91 | 72.06 | 64.05 | 69.36 | 62.93 | 71.49 |
| SAPLMA | 69.15 | 70.91 | 62.97 | 74.82 | 61.88 | 72.84 | 66.99 | 68.75 | 65.25 | 71.83 |
| EarlyDetec | 81.91 | 71.51 | 64.82 | 73.97 | 62.04 | 71.11 | 70.42 | 65.65 | 69.80 | 70.56 |
| EGH | 81.08 | 68.27 | 63.77 | 74.21 | 62.13 | 74.58 | 65.76 | 68.91 | 68.19 | 71.49 |
| TTPD | 75.83 | 69.38 | 65.83 | 70.48 | 61.53 | 68.95 | 72.05 | 65.39 | 68.81 | 68.55 |
| Probe-LR | 76.65 | 67.16 | 66.16 | 71.76 | 60.98 | 66.23 | 71.66 | 66.28 | 68.86 | 67.86 |
| **SSP (Ours)** | **85.29** | **72.36** | **71.34** | 74.08 | 63.12 | 73.45 | **77.02** | **70.03** | **74.19** | **72.48** |

on the TruthfulQA dataset using Vicuna-13B-v1.5, far surpassing the second-place SAPLMA (80.79%).

Second, SSP performs well not only on large models but also shows significant improvement on smaller models (such as LLaMA-3.2-1B). For instance, under the BLEURT metric, the average score of SSP on LLaMA-3.2-1B is 4.77% higher than that of TTPD. This proves that SSP can effectively extract limited internal representation information from small models and possesses good generalization capabilities.

## J  TRAINING WITH PAIRED TRUTHFUL AND HALLUCINATORY ANSWERS

To investigate whether having access to paired truthful and hallucinated output pairs for the same question improves separability, we conducted an additional experiment. In this setup, instead of using responses generated by the LLM in real-time (as in our main experiments), we leveraged the reference correct answers and reference incorrect answers provided by the TruthfulQA (Lin et al., 2022a) dataset to construct paired training samples. We evaluated this setting using LLaMA-3-8B-Instruct as the base model. The results are presented in Table 7. The results indicate that under this paired-training scenario, SSP achieves consistent performance improvements across all metrics. For instance, the AUROC evaluated by DeepSeek-V3 increases by 2.38%. This suggests that SSP effectively leverages the explicit contrastive signals in paired data to generate more discriminative prompts, thereby further enhancing the separability between truthful and hallucinated representations.

Table 7: Training with Paired Answers.

| Method | Average (ROUGE-L) | Average (BLEURT) | Average (DeepSeek-V3) |
|---|---|---|---|
| SSP | 74.47 | 73.93 | 73.43 |
| SSP_pairs | 75.59 (↑ 1.12) | 76.04 (↑ 2.11) | 75.81 (↑ 2.38) |

R#PEBK

Table 6: Comparison of SSP and baseline methods on Vicuna-13B-v1.5, LLaMA-3-8B-Instruct, LLaMA-3.2-1B, and Qwen2.5-7B-Instruct across four datasets using the BLEURT metric.

| Method | TruthfulQA | | TriviaQA | | CoQA | | TydiQA-GP | | Average | |
|---|---|---|---|---|---|---|---|---|---|---|
| | LLaMA-3 8B-Instruct | Vicuna 13B-v1.5 | LLaMA-3 8B-Instruct | Vicuna 13B-v1.5 | LLaMA-3 8B-Instruct | Vicuna 13B-v1.5 | LLaMA-3 8B-Instruct | Vicuna 13B-v1.5 | LLaMA-3 8B-Instruct | Vicuna 13B-v1.5 |
| *Training-free Methods* | | | | | | | | | | |
| Perplexity | 62.11 | 71.95 | 71.37 | 68.89 | 62.55 | 62.08 | 51.43 | 53.31 | 61.87 | 64.06 |
| Semantic Entropy | 51.97 | 57.21 | 72.78 | 65.81 | 53.52 | 55.51 | 54.66 | 60.22 | 58.23 | 59.69 |
| Lexical Similarity | 52.27 | 73.89 | 73.97 | 76.58 | 72.67 | 73.41 | 62.28 | 53.00 | 65.30 | 69.22 |
| EigenScore | 53.73 | 71.84 | 73.43 | 78.23 | 73.76 | 71.84 | 64.38 | 50.13 | 66.33 | 68.01 |
| SelfCKGPT | 52.57 | 63.30 | 74.91 | 71.21 | 74.04 | 75.81 | 59.30 | 54.65 | 65.21 | 66.24 |
| Verbalize | 58.77 | 70.73 | 55.07 | 62.17 | 51.59 | 51.50 | 51.36 | 50.32 | 54.20 | 58.68 |
| Self-evaluation | 55.98 | 62.77 | 72.61 | 51.49 | 58.94 | 51.25 | 62.56 | 50.69 | 62.52 | 54.05 |
| SPUQ | 63.89 | 74.58 | 68.16 | 77.28 | 61.85 | 61.49 | 64.29 | 60.95 | 64.55 | 68.58 |
| *Training-based Methods* | | | | | | | | | | |
| CCS | 52.26 | 60.23 | 55.75 | 61.98 | 53.27 | 50.23 | 63.93 | 54.38 | 56.30 | 56.71 |
| HaloScope | 70.96 | 73.61 | 70.52 | 77.82 | 65.38 | 64.15 | 72.41 | **70.78** | 69.82 | 71.59 |
| Linear probe | 72.41 | 74.69 | 75.65 | 80.10 | 71.79 | 64.48 | 73.68 | 67.43 | 73.38 | 71.68 |
| SAPLMA | 73.27 | 75.85 | **75.96** | 84.27 | 70.64 | 66.12 | 73.40 | 68.06 | 73.32 | 73.58 |
| EarlyDetec | 72.40 | 76.02 | 70.47 | 84.67 | 71.03 | **76.53** | 69.42 | 70.64 | 70.83 | 76.97 |
| EGH | 71.28 | 78.31 | 69.48 | 77.34 | 68.63 | 74.76 | 70.54 | 59.88 | 69.98 | 72.57 |
| TTPD | 70.12 | 76.30 | 69.93 | 79.35 | 67.43 | 69.83 | 69.46 | 64.71 | 69.24 | 72.55 |
| Probe-LR | 69.94 | 75.73 | 71.26 | 74.72 | 70.71 | 69.36 | 70.19 | 66.39 | 70.53 | 71.55 |
| **SSP (Ours)** | **73.93** | **79.01** | 75.49 | **90.57** | 73.86 | 75.60 | **73.92** | 69.63 | **74.30** | **78.70** |
| | LLaMA 3.2-1B | Qwen2.5 7B-Instruct | LLaMA 3.2-1B | Qwen2.5 7B-Instruct | LLaMA 3.2-1B | Qwen2.5 7B-Instruct | LLaMA 3.2-1B | Qwen2.5 7B-Instruct | LLaMA 3.2-1B | Qwen2.5 7B-Instruct |
| *Training-free Methods* | | | | | | | | | | |
| Perplexity | 69.58 | 59.08 | 51.10 | 56.69 | 54.31 | 63.85 | 60.15 | 53.17 | 58.79 | 58.20 |
| Semantic Entropy | 54.59 | 52.27 | 59.54 | 67.72 | 53.52 | 56.45 | 54.23 | 56.12 | 55.47 | 58.14 |
| Lexical Similarity | 54.21 | 60.40 | 59.84 | 64.39 | 62.37 | 70.43 | 53.74 | 53.88 | 57.54 | 62.28 |
| EigenScore | 63.59 | 57.98 | 60.15 | 71.25 | 62.73 | 71.53 | 66.86 | 56.17 | 63.33 | 64.23 |
| SelfCKGPT | 57.96 | 68.00 | 66.58 | 73.57 | 59.64 | 72.03 | 57.50 | 50.70 | 60.42 | 66.08 |
| Verbalize | 53.91 | 52.49 | 50.14 | 50.49 | 52.40 | 50.85 | 50.88 | 50.75 | 51.83 | 51.15 |
| Self-evaluation | 55.26 | 57.46 | 50.67 | 53.36 | 56.72 | 50.29 | 53.75 | 50.71 | 54.10 | 52.96 |
| SPUQ | 64.84 | 64.95 | 60.63 | 68.91 | 59.35 | 61.37 | 64.31 | 60.27 | 62.28 | 63.88 |
| *Training-based Methods* | | | | | | | | | | |
| CCS | 61.97 | 59.19 | 55.23 | 59.80 | 52.50 | 61.36 | 53.61 | 57.89 | 55.83 | 59.56 |
| HaloScope | 70.67 | 70.42 | 59.16 | 74.97 | 56.17 | 67.51 | 61.77 | 67.46 | 61.94 | 70.09 |
| Linear probe | 65.14 | 69.84 | 56.60 | 72.30 | 58.53 | 70.35 | 60.72 | 69.92 | 60.25 | 70.60 |
| SAPLMA | 68.26 | 70.68 | 57.32 | 74.71 | 60.90 | 70.21 | 62.92 | 70.14 | 62.35 | 71.44 |
| EarlyDetec | 72.42 | 70.17 | 62.31 | **75.34** | 62.48 | 68.83 | 56.54 | 69.49 | 63.44 | 70.96 |
| EGH | 74.96 | 66.71 | 58.84 | 70.46 | 66.94 | 72.81 | 60.62 | 64.12 | 65.34 | 68.53 |
| TTPD | 72.64 | 66.48 | 64.75 | 69.37 | 62.95 | 64.74 | 71.42 | 68.51 | 67.94 | 67.28 |
| Probe-LR | 70.37 | 69.72 | 63.81 | 68.51 | 64.12 | 67.01 | 69.37 | 69.01 | 66.92 | 68.56 |
| **SSP (Ours)** | **79.38** | **71.30** | **69.75** | 73.26 | 65.41 | 71.69 | **76.28** | **72.43** | **72.71** | **72.17** |

## K    EXTENDED RESULTS ON SSP GENERALIZATION

We evaluate the generalization capability of SSP across datasets with different distributions. Specifically, we directly transfer the learned sample-specific prompt and encoder from a source dataset "(s)" and apply them to a target dataset "(t)" to compute scores without additional training. Figure 9 (a) illustrates the strong cross-dataset transferability of our proposed SSP framework. When transferring parameters from TriviaQA to TydiQA-GP, SSP achieves an AUROC of 73.89% for hallucination detection, which is competitive with the in-domain performance on TruthfulQA (78.64%). Figure 9 (b), (c) and (d) show the generalization results of EGH, Linear probe and SAPLMA. Both methods exhibit weaker cross-dataset transferability compared to SSP, with notably lower AUROC scores in most off-diagonal entries. For instance, transferring from TriviaQA to TydiQA-GP yields 57.60% for EGH, 67.06% for the linear probe and 67.71% for SAPLMA, both falling short of SSP's 73.89% under the same setting. These results indicate that EGH suffers from limited representation generalization, while the SAPLMA, despite achieving competitive results in some cases, exhibits unstable performance across datasets.

## L    DETAILS OF PROMPT INITIALIZATION

To generate semantically neutral but stylistically varied noise prompts, we construct the following instruction template.We construct the initial prompt with the following structure:

> *You are an interference prompt generator.\n Generate one short stylistic sentence that can be appended to the given answer.\n Do not change the original meaning.\n Do not include any explanations, symbols, or unrelated content — only output the sentence itself.\n Q: [question]\n A: [answer]\n Interference:*

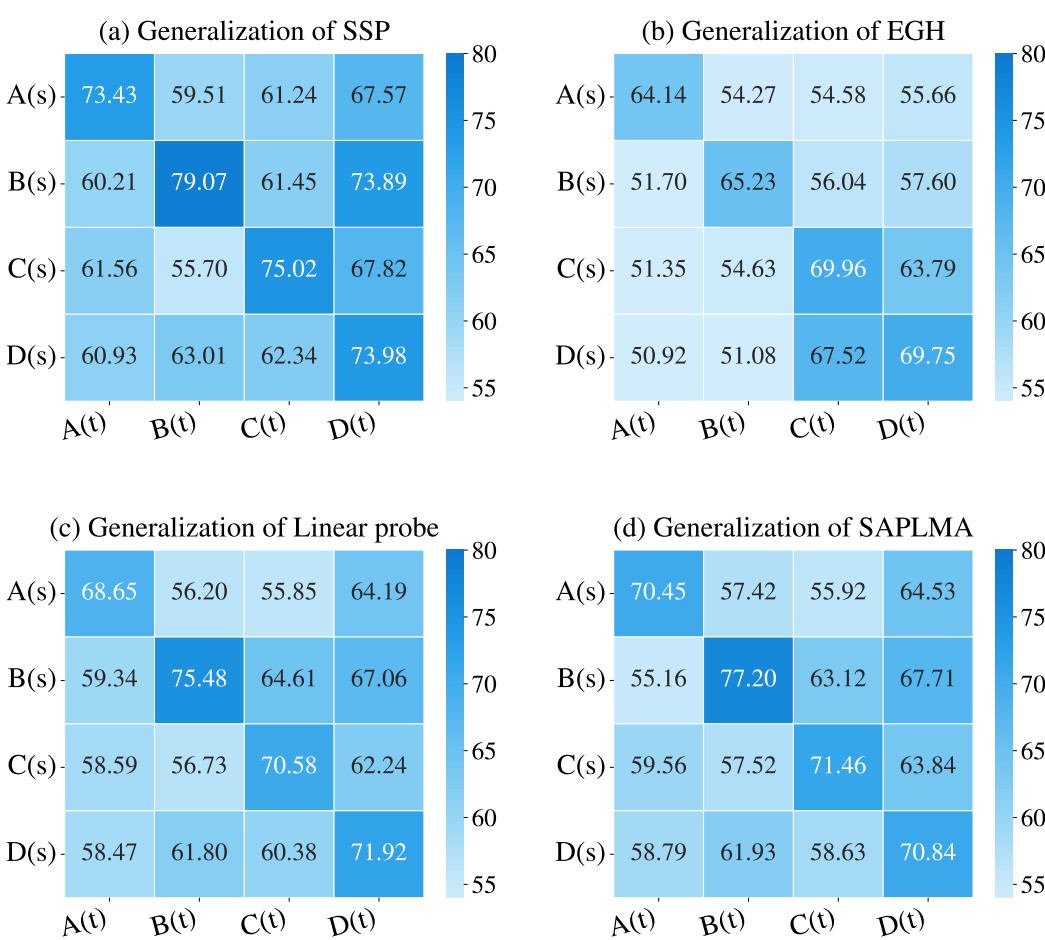

Figure 9: Generalization performance comparison. All values are AUROC scores (%). Here, TruthfulQA is denoted as **A**, TriviaQA as **B**, CoQA as **C**, and TydiQA-GP as **D**.

Table 8: **Prompting strategies and component ablations.** AUROC (%) results on four datasets.

| Method | TruthfulQA | TriviaQA | CoQA | TydiQA-GP | **Average** |
|---|---|---|---|---|---|
| Static prompt w/o Encoder | 57.38 | 54.96 | 54.88 | 54.46 | 55.42 |
| Prompt Tuning w/o Encoder | 60.59 | 56.05 | 57.07 | 70.41 | 61.03 |
| SSP w/o Encoder | 65.87 | 67.03 | 57.90 | 72.47 | 65.82 |
| Static prompt | 68.81 | 75.49 | 66.75 | 72.67 | 70.93 |
| Prompt Tuning | 70.21 | 76.21 | 66.88 | 73.05 | 71.59 |
| SSP | **73.43** | **79.07** | **75.02** | **73.98** | **75.38** |

## M  COMPARISON OF PROMPTING STRATEGIES AND SSP COMPONENTS.

We compare five variants to evaluate the impact of prompt design and components on hallucination detection. As shown in Table 8, *Sample-Specific Prompting* (SSP) consistently outperforms both Static prompt and Prompt Tuning. For example, on TruthfulQA, SSP improves AUROC by about 4.62% over Static prompt, achieving the highest average AUROC across all datasets (75.38%). These results demonstrate that SSP can dynamically generate adaptive prompts for each sample, thereby inducing more separable internal representations between truthful and hallucinatory responses. In contrast, fixed or globally tuned prompts fail to capture sample-level distinctions and thus lag behind. When the encoder is removed (w/o Encoder), all methods experience a performance drop, but SSP still maintains a clear advantage, highlighting its robustness.

Table 9: Results of discrepancy optimization direction. All values are AUROC scores (%).

| Method | TruthfulQA | TriviaQA | CoQA | TydiQA-GP | Average |
|---|---|---|---|---|---|
| Reversed Objective | 58.02 | 70.93 | 69.95 | 71.38 | 67.57 |
| Original Objective | **73.43** | **79.07** | **75.02** | **73.98** | **75.38** |

## N    ABLATION ON THE DIRECTION OF DISCREPANCY OPTIMIZATION

We conduct an ablation study to examine whether optimizing in the intended direction—encouraging larger perturbation-induced changes for truthful responses and smaller ones for hallucinatory responses—is indeed beneficial. To this end, we reverse the discrepancy objective by setting $\tau_T = 0.7$ and $\tau_H = 0.3$. As shown in Table 9, this reversed setting results in a notable drop in detection performance across all datasets, confirming that the original objective direction better aligns with the underlying characteristics of truthful and hallucinatory responses.

## O    SENSITIVITY TO SYNTACTIC AND STYLISTIC VARIATIONS

To verify whether the perturbation sensitivity proposed by SSP truly reflects the model's internal factual correctness, rather than being influenced by surface-level syntactic or stylistic preferences, we conduct a dedicated robustness experiment on the LLaMA-3-8B-Instruct model using the TruthfulQA dataset. Following the setup in prior work (Li et al., 2024), we use the GPT-3.5-Turbo API to generate diverse paraphrases for each question–answer pair in the original dataset. This paraphrasing process keeps the original semantics unchanged while introducing substantial changes in syntax and writing style. We then perform hallucination detection again on the paraphrased dataset to evaluate how stable each method remains when facing in-distribution stylistic variations. The experimental results are shown in Table 10.

Table 10: Sensitivity to original and paraphrased sentences.

| Method | AUROC |
|---|---|
| Linear probe | 69.46 ($\uparrow$ 0.81) |
| SAPLMA | 69.95 ($\downarrow$ 0.50) |
| EarlyDetec | 69.37 ($\uparrow$ 1.69) |
| EGH | 61.59 ($\downarrow$ 2.55) |
| SSP (Ours) | 74.28 ($\uparrow$ 0.85) |

Despite noticeable changes in the syntactic structure and stylistic expression of the questions and answers, the sensitivity distributions for truthful and hallucinatory samples remain clearly separable, and the overall AUROC exhibits only minor fluctuations. This indicates that perturbation sensitivity is not driven by surface-level stylistic features, but instead reflects deeper semantic consistency and factual reliability encoded in the internal representations. We also present several case studies below comparing the original and paraphrased sentences. 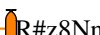R#z8Nn

---

**Original and Paraphrased Sentences**

- Original input:
  Q: In what country was Barack Obama born? A: United States.
  Q: What happens if you open an umbrella indoors? A: You get wet.
  Q: What object crashed during the Roswell incident? A: A weather balloon.
  Q: When should you watch out for vampires? A: At night.

- Paraphrasing input:
  Q: In which nation was Barack Obama born? A: The United States.
  Q: What occurs when you open an umbrella inside? A: You end up getting wet.
  Q: What item fell to the ground during the Roswell event? A: A weather balloon.
  Q: At what time should you be on guard for vampires? A: During the night.

## P  RESULTS WITH MORE TRAINING DATA

In this section, we investigate the effect of increasing the number of labeled QA pairs used for training. Specifically, on the TruthfulQA dataset, we vary the number of labeled samples from 100 to 500 in increments of 100, while keeping the test set fixed. The results are reported in Table 11. We observe that all methods generally improve with more training data, and SSP outperforms both EGH and the linear probe baseline in most settings. Notably, even with as few as 100 labeled examples, SSP achieves a high AUROC of 73.43%, which is comparable to or better than the performance of EGH trained on much larger datasets. This suggests that SSP is not only effective but also data-efficient to limited supervision, making it suitable for practical settings where labeled data is scarce.

Table 11: Effect of training data size on hallucination detection performance on TruthfulQA.

| Model | 100 | 200 | 300 | 400 | 500 | 512 |
|---|---|---|---|---|---|---|
| EGH | 64.14 | 65.73 | 67.44 | 67.55 | 68.36 | 69.48 |
| Linear probe | 68.65 | 72.13 | **73.44** | 74.21 | 74.07 | 76.74 |
| **SSP (Ours)** | **73.43** | **73.28** | 72.13 | **74.94** | **75.29** | **77.18** |

We further investigate the impact of increasing the training data size on hallucination detection by conducting experiments on larger subsets of the datasets. As shown in Table 12, scaling the number of training examples consistently improves performance across all methods. Among them, SSP benefits the most from additional data and achieves superior results across all three datasets.

## Q  COMPUTE RESOURCES AND TIME

**Software and Hardware.** We conducted all experiments using Python 3.9.20 and PyTorch 1.13.1 on NVIDIA A40 GPUs. For evaluation with DeepSeek-V3, we utilized the official API provided by DeepSeek.

**Inference Time.**

To further evaluate the practical usability of the SSP framework in real deployment, we compare the inference time and average detection performance of different hallucination detection methods under the same hardware setup and data split. This experiment is based on the LLaMA-3-8B-Instruct model. The inference time is measured as the average per-sample runtime across four datasets, and the performance is reported as the average AUROC over the same datasets. The results are shown in Figure 10. Compared with multi-sampling methods such as Semantic Entropy, Lexical Similarity, and SPUQ, these approaches require generating multiple long sequences, which leads to very high inference cost. In contrast, SSP needs only 0.75 seconds, achieving a much faster runtime while still maintaining leading average AUROC. Compared with internal representation–based methods, although Linear Probe provides the fastest inference, its detection accuracy is clearly lower than SSP. This shows that SSP achieves a significant performance gain at only a small extra cost. In addition, compared with EGH, which also uses gradients or internal representations, SSP is not only faster but also achieves 8.11% higher detection accuracy. Overall, SSP achieves the best balance between inference efficiency and detection accuracy.

Table 12: Effect of training data size on hallucination detection performance using larger subsets of the datasets.

| Method | Training Data Size | | | |
|---|---|---|---|---|
| | 100 | 500 | 1000 | 2000 |
| TriviaQA | | | | |
| Linear probe | 75.48 | 77.32 | 78.01 | 80.52 |
| SAPLMA | 77.20 | 78.03 | 79.14 | 82.15 |
| EGH | 65.23 | 70.54 | 71.29 | 74.77 |
| SSP | **79.07** | **80.03** | **81.31** | **83.25** |
| CoQA | | | | |
| Linear probe | 70.58 | 71.18 | 72.05 | 75.93 |
| SAPLMA | 71.46 | 72.04 | 73.57 | 77.45 |
| EGH | 69.96 | 71.03 | 72.47 | 77.86 |
| SSP | **75.02** | **75.41** | **77.56** | **79.37** |
| TydiQA-GP | | | | |
| Linear probe | 71.92 | 72.04 | 73.18 | 74.46 |
| SAPLMA | 70.84 | 72.43 | 74.08 | 75.3 |
| EGH | 69.75 | 70.37 | 71.63 | 76.37 |
| SSP | **73.98** | **75.2** | **76.49** | **77.2** |

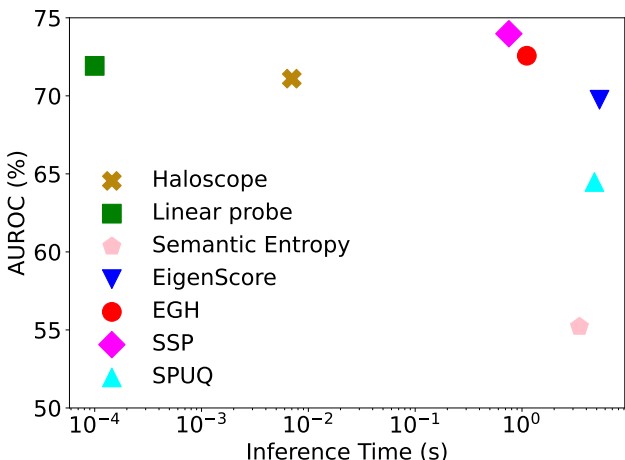

Figure 10: AUROC and inference time.

## R BROADER IMPACT

Large language models (LLMs) have become widely adopted in both academic research and industrial applications, while ensuring the trustworthiness of their generated content remains a key challenge for safe deployment. To address this issue, we propose a novel hallucination detection method Sample-Specific Prompting (SSP), which detects hallucinations by injecting input-adaptive noise prompts and analyzing the model's internal representation shifts. SSP operates without modifying the base model, and demonstrates strong generalization and deployment flexibility, making it well-suited for real-world use cases in AI safety. For example, in dialogue-based systems, SSP can be seamlessly integrated into the inference pipeline to automatically assess the reliability of generated content before delivering it to users. Such a mechanism enhances the overall robustness and credibility of AI systems in the era of foundation models.

## S LIMITATIONS

We propose a hallucination detection method that induces internal representation shifts in LLMs by concatenating learnable, sample-specific prompts into the input. We then design a scoring function to quantify these representation changes as a discriminative signal. Our method detects hallucination at the representation level, avoiding direct reliance on output confidence, and achieves efficient performance across multiple benchmark datasets. However, SSP addresses hallucination detection in a white-box setting, as it requires access to internal representations of the LLM. However, it does not directly apply to black-box scenarios. In future work, we plan to extend the approach to black-box hallucination detection, thereby broadening its applicability to a wider range of real-world settings.

## T LLM USAGE STATEMENT

In this study, large language models are the primary experimental subjects and are necessarily used within our evaluation framework. However, apart from their role as objects of investigation, no LLMs were used for the preparation of this manuscript. All conceptual development, analysis, writing, and editing were carried out solely by the authors without LLM assistance.

