# OpenReview forum: "Perturbations Matter: Sensitivity-Guided Hallucination Detection in LLMs"
_ICLR.cc/2026/Conference — Submitted to ICLR 2026_

### Official Review · Reviewer_3guq · 2025-10-24

**Soundness:** 3
**Presentation:** 3
**Contribution:** 2
**Rating:** 4
**Confidence:** 4

**Summary:**

This paper proposes a novel approach to detect hallucinations in LLMs by analyzing the sensitivity of internal representations to prompt-induced perturbations. The key finding is that truthful responses exhibit greater sensitivity to perturbations than hallucinatory ones, with theoretical guarantees for high separability. The proposed method, Sample-Specific Prompting (SSP), dynamically constructs prompts per sample, measures representation shifts via cosine distance, and trains lightweight encoders using contrastive loss. Experiments on QA datasets such as TruthfulQA demonstrate superior AUROC over baselines, along with strong efficiency and generalization.

**Strengths:**

**Novel Perspective on Hallucination Detection**: This work is the first to explicitly leverage perturbation sensitivity as a detection signal, shifting the focus from static embeddings to dynamic representation shifts.

**Solid Theoretical Foundation**: Theorems 1 and 2 provide probabilistic guarantees on separability, supported by Cantelli's inequality and empirical validation on real datasets. This grounds the method in theory rather than relying solely on heuristics.

**Comprehensive Experiments**: Evaluations across multiple models, datasets demonstrate the robustness of the approach, which outperforms baselines on these benchmarks.

**Weaknesses:**

The comparison in Figure 1 may be problematic. K-means is an unsupervised method, while the proposed Perturbation Sensitivity is supervised. The poor performance of K-means might not indicate weak inherent separation but could result from its inability to find the optimal separation hyperplane, making the comparison potentially unfair.

Therefore, the claim that "internal representations of LLMs frequently fail to provide a clear separation between truthful and hallucinatory content" requires further discussion. Several recent studies [1,2] have demonstrated good separation using internal representations directly, raising questions about the validity of this separation bottleneck.

Although this method beats the traditional methods like Linear Probe or  SAPLMA, the method introduces complexity by requiring prompt optimization for the training data, which may raise concerns about its robustness and practicality.

[1] Bürger et al. Truth is Universal: Robust Detection of Lies in LLMs. NeurIPS 2024

[2] Liu et al. On the Universal Truthfulness Hyperplane Inside LLMs. EMNLP 2024

**Questions:**

The main questions concern the limitation described in the introduction and Figure 1.

Beyond the concerns mentioned in the Cons, the paper uses three labeling criteria: ROUGE-L (R), BLEURT (B), and DeepSeek-V3 (D). Given that using ROUGE-L to measure correctness can be inaccurate [3], I suggest that only the scores for DeepSeek-V3 should be displayed, as there are too many scores in Table 1.

---

> ### Author Response · Authors · 2025-11-22
> **Response to Reviewer 3guq (1/2)**
>
> We sincerely thank the reviewer for the positive and encouraging feedback. We are glad that the novelty of our perspective, the strength of our theoretical foundation, and the breadth of our empirical evaluations were appreciated. We also thank the reviewer for the insightful questions and are happy to clarify the points raised as follows:
>
> > W1 & Q1: Does the poor K-means result really demonstrate that internal representations are not separable?
>
>
> Thank you for raising this important point. We first emphasize that K-means is a mature and effective clustering algorithm widely used in representation probing and clustering. For example, recent works such as ClusterLLM [1] and LLMEdgeRefine [2] adopt K-means as the default method for embedding-based clustering or as an initialization step, and rely on its performance to indicate whether the embedding space contains identifiable cluster structure. These practices show that if internal representations form clear clusters, classical distance-based methods like K-means can typically reveal such structure effectively.
>
> Therefore, in Figure 1, we use K-means not as a competing classifier but as an unsupervised probe to assess the inherent separability of internal representations. If the LLM’s internal features naturally separate truthful from hallucinatory samples, K-means should be able to produce a clear clustering even without supervision. Its poor performance does not indicate a weakness of the method, but rather reflects that the two types of samples are highly intermixed in the representation space and lack inherent separability—precisely the structural bottleneck our work aims to reveal and overcome.
>
> To further address the reviewer’s suggestion, we follow the approach in [3,4], which uses a supervised logistic regression model to evaluate the separability between truthful and hallucinated representations. In our revision, we will adopt the same experimental setup as used for K-means, train a logistic regression classifier on the corresponding internal representations, and report its AUROC. Our results show that the model achieves an AUROC of only **60.96%**, indicating that even under supervision, the representations lack sufficient discriminative information to effectively separate truthful and hallucinatory samples. This finding further confirms that the limitation lies not in the clustering method itself, but in the intrinsic inseparability of the internal representations. Finally, we will update Figure 1(a) with these revised results in the updated version of the paper.
>
>
>
>
>
>
>
>
>
> > W2 & Q1: The validity of the “separation bottleneck” proposed in the paper.
>
>
> Great point, and thank you for kindly inviting our thoughts on this. We acknowledge the contributions of [3,4] and will include a discussion in the related work, as well as add them as baselines (see **Tables 1, 5 and 6** for details). We observe that the separability reported in prior works [3,4] does not directly generalize to real-world scenarios. This is because these methods rely on manually curated truthful or hallucinated answer pairs. However, in practical settings, when an LLM is asked the same question, the responses it generates often do not follow the patterns assumed in such artificially constructed datasets. As a result, this setup differs from the actual conditions of hallucination detection.
>
> Further, because real-world datasets are highly diverse, the methods in [3, 4] address separability only in specific settings and do not imply that separability bottlenecks disappear across all realistic scenarios. To obtain a more realistic evaluation, we re-examined the methods in [3, 4] using LLM-generated truthful and hallucinatory answers from LLaMA-3-8B-Instruct as their training data. The results are shown in the table below:
>
> | Method | TruthfulQA | TriviaQA |  CoQA |  TydiQA-GP  |  Average  |
> | :------: | :------------:|:------:|:------:|:------:| :------:|
> |  TTPD[3]     |    67.24    |    69.51    |   70.12    |   69.46    |   69.08     |
> |  Probe-LR[4]   |   68.06     |   68.14    |   72.48  |   69.27   |     69.49    |
> | SSP (Ours)     |  **73.43**      | **79.07**   |   **75.02**     |  **73.98**   |   **75.38**     |
>
> From the results, we can see that the methods in [3, 4] show limited separability, indicating that in settings closer to real-world scenarios, static internal representations alone do not provide good separability.

---

> ### Author Response · Authors · 2025-11-22
> **Response to Reviewer 3guq (2/2)**
>
> > W3: The impact of method complexity on robustness and practical usability.
>
> First, as shown in the table below, we demonstrate the cross-dataset generalization ability of SSP. When transferring across datasets, SSP consistently achieves strong performance, indicating that the method generalizes well. All experiments are conducted on LLaMA-3-8B-Instruct and evaluated using DeepSeek-V3. Here, TruthfulQA [5] is denoted as A, TriviaQA [6] as B, CoQA [7] as C, and TydiQA-GP [8] as D (see **Appendix K** for details).
>
> |    Method    |  A->B  |  A->C  |  A->D  |   B->A  |  B->C  |  B->D  |  C->A  |  C->B  |  C->D  |  D->A  |  D->B  |  D->C  |
> |:------------:|:------:|:------:|:------:|:------:|:------:|:------:|:------:|:------:|:------:|:------:|:------:|:------:|
> | Linear probe | 56.20  | 55.85  | 64.19  | 59.34  | **64.61**  | 67.06  | 58.59  | 56.73  | 62.24  | 58.47  | 61.80  | 60.38  |
> |    SAPLMA    | 57.42  | 55.92  | 64.53  | 55.16  | 63.12  | 67.71  | 59.56  | 57.52  | 63.84  | 58.79  | 61.93 | 58.63  |
> |      EGH     | 54.27  | 54.58  | 55.66  | 51.70  | 56.04  | 57.60  | 51.35  | 54.63  | 63.79  | 50.92  | 51.08  | 67.52  |
> |      SSP (Ours)     | **59.51**  | **61.24**  | **67.57**  | **60.21**  | 61.45  | **73.89**  | **61.56**  | 55.70  | **67.82**  | **60.93**  | **63.01**  | 62.34  |
>
> The table below presents the inference time comparison. Linear probe is the simplest model and therefore the fastest. Although SSP is slightly slower, the difference is small, and both methods remain in the same order of magnitude, which is entirely acceptable. Compared with methods that require multiple samples or access to various internal representations, SSP keeps the inference cost low while achieving significantly higher detection performance (please refer to **Appendix Q** for details).
>
> |     Method          | Inference Time(s)  | Average (DeepSeek-V3) |
> |:-------------------:|:--------------:|:-----------:|
> | Semantic Entropy    |     3.4469       |     61.94     |
> | Lexical Similarity |      3.8582       |   67.57      |
> | EigenScore          |       5.3162     |    68.50    |
> | Linear probe              |     0.1528       |    71.66  |
> | SAPLMA               |     0.1893            |    72.49       |
> | EGH                 |       1.1090        |    67.27     |
> | SSP (Ours)          |      0.7534         |   75.38     |
>
> Therefore, we believe that SSP achieves a good trade-off between computational cost and performance, and offers strong practical applicability.
>
> > Q2: A suggestion to simplify Table 1.
>
> We thank the reviewer for the helpful suggestion. Regarding the use of three labeling standards (ROUGE-L, BLEURT, and DeepSeek-V3), our intention was to cover surface-level matching, semantic matching, and model-based factuality judgment so as to obtain a comprehensive understanding of how sensitivity behaves under different labeling schemes.
>
> We agree with the reviewer that ROUGE-L may be unreliable for factuality evaluation. To avoid clutter and improve readability, we will keep only the DeepSeek-V3 scores in the main paper, and move the ROUGE-L and BLEURT results to the **Appendix I** for interested readers.
>
>
>
>
> References
>
> [1] Zhang et al., ClusterLLM: Large Language Models as a Guide for Text Clustering. EMNLP, 2023.
>
> [2] Feng et al., LLMEdgeRefine: Enhancing Text Clustering with LLM-Based Boundary Point Refinement. EMNLP, 2024.
> [3] Bürger et al. Truth is Universal: Robust Detection of Lies in LLMs. NeurIPS, 2024.
>
> [4] Liu et al. On the Universal Truthfulness Hyperplane Inside LLMs. EMNLP, 2024.
>
> [5] Lin et al. Truthfulqa: Measuring how models mimic human falsehoods. ACL, 2022.
>
> [6] Joshi et al. Triviaqa: A large scale distantly supervised challenge dataset for reading comprehension. ACL, 2021.
>
> [7] Reddy et al. Coqa: A conversational question answering challenge. Transactions of the Association for Computational Linguistics, 2019.
>
> [8] Clark et al. Tydiqa: A benchmark for information-seeking question answering in ty pologically di verse languages. Transactions of the Association for Computational Linguistics, 2020.

---

> ### Author Response · Authors · 2025-11-25
> **Invitation for Further Discussion**
>
> Dear Reviewer 3guq,
>
> We have now posted a detailed response to your comments. Thank you again for your constructive feedback, which has helped us improve the paper.
>
> We would appreciate it if you could take a look at our rebuttal. If you find that our response has adequately addressed your concerns, we kindly ask that you consider reconsidering your score.
>
> We look forward to your feedback and are happy to answer any further questions.
>
> Best regards,
>
> The Authors

---

> ### Author Response · Authors · 2025-11-26
> **Invitation for Further Discussion**
>
> Dear Reviewer 3guq,
>
> We have now posted a detailed response to your comments. Thank you again for your constructive feedback, which has helped us improve the paper.
>
> We would appreciate it if you could take a look at our rebuttal. If you find that our response has adequately addressed your concerns, we kindly ask that you consider reconsidering your score.
>
> We look forward to your feedback and are happy to answer any further questions.
>
> Best regards,
>
> The Authors

---

> > ### Comment · Reviewer_3guq · 2025-11-26
> >
> > Thanks for the detailed response!
> >
> > I still have one remaining question about Figure 1, as it is central to the paper’s main motivation.
> >
> >
> > * I still want to confirm that K-means  is not sufficient to prove poor separability. There are known theoretical cases where well-separated data can still yield very bad K-means.
> >
> >
> >
> > * In the current Figure 1, the logistic regression achieves 60% AUROC. The perturbation sensitivity achieves nearly 100% separation, while Table 1 on TruthfulQA shows only 70+%. These numbers do not seem to match. Could you further explain this gap?

---

> > > ### Author Response · Authors · 2025-11-27
> > > **Re: Reviewer 3guq - Appreciation and Response to Feedback**
> > >
> > > > Concern about the insufficiency of K-means to prove poor separability.
> > >
> > > We appreciate this insightful comment. We agree with your comments regarding the theoretical limitations of K-means. As you correctly pointed out, K means may fail to cluster data correctly even when the data is separable.
> > > For instance, Figure 1i in [1] and Figure 5a in [2] demonstrate that K-means fails on non convex manifold data even when the data is separable. We thank you for pointing this out as your suggestion has been very helpful in improving the rigor of our work.
> > >
> > > For this reason, we have replaced K-means with a supervised Logistic Regression method **in our latest revision**. We present the results of this supervised Logistic Regression in the updated Figure 1a. The experiments show that the model achieves an AUROC of only 60.96% even with supervision. Furthermore, the score distributions for truthful and hallucinatory contents show significant overlap.
> > >
> > > These results provide much stronger evidence than K-means to confirm the separability bottleneck in the original internal representations. **Consequently, this new evidence strongly supports our core motivation.** It confirms that **the internal representations lack clear separability**. Our method addresses this issue and significantly improves separability by introducing perturbations.
> > >
> > >
> > > > Concern about the performance gap between Figure 1 and Table 1.
> > >
> > > We appreciate your careful observation. This point is key to understanding the logic of our paper. We explicitly discussed this phenomenon in the **Remark** part of **Section 4.2**. These two numbers do not contradict each other. Instead, they represent the theoretical potential for separability and the capability of our method to handle unseen samples. We provide a detailed explanation below.
> > >
> > > **Figure 1b** represents the theoretical lower bound in an Oracle setting. As stated in **Section 4.1**, Figure 1b demonstrates an Oracle setting. In this experiment, we optimize the perturbation vector individually for each specific sample. This maximizes the distinctiveness of that sample. This result proves an important theory of existence. It shows that a specific perturbation **exists** for nearly every sample to distinguish it perfectly. However, this serves only as a theoretical lower bound to demonstrate potential. It requires labels to optimize the perturbation during testing. Therefore, it is impossible to achieve this directly during practical inference.
> > >
> > > **Table 1** displays the performance of our method SSP on the test set. In practical applications, we cannot optimize the perturbation individually for every test sample. Therefore, the goal of SSP is to learn a prompt generator. This generator predicts the perturbation that triggers a difference in sensitivity. Since the model must process unseen samples, a performance **gap is inevitable**.
> > >
> > > We explicitly pointed this out in the **Remark** part of **Section 4.2** of our paper. The 99% result in `Eq. (14)` serves as an indicator of the theoretical potential of perturbation sensitivity. It should not be interpreted as a direct guarantee of performance during testing.
> > >
> > >
> > > **References**
> > >
> > > [1] Ng A, Jordan M, Weiss Y. On spectral clustering: Analysis and an algorithm. NeurIPS, 2001.
> > >
> > > [2] Jain A K. Data clustering: 50 years beyond K-means. Pattern recognition letters, 2010.

---

> ### Author Response · Authors · 2025-11-28
> **Invitation for Further Discussion**
>
> Dear Reviewer 3guq,
>
> Thank you very much for your continued engagement and the follow-up questions.
>
> We have posted a response to your latest comments. We hope this clarifies the remaining concerns.
>
> As the discussion phase is coming to a close soon, could you please take a moment to review our latest reply? We would be happy to provide any further clarifications if needed.
>
> If you feel that the issues have been resolved, we would be sincerely grateful if you could consider increasing your score accordingly.
>
> Kind regards,
>
> Senior Author

---

### Official Review · Reviewer_f1QU · 2025-10-27

**Soundness:** 2
**Presentation:** 3
**Contribution:** 2
**Rating:** 4
**Confidence:** 4

**Summary:**

This paper studies hallucination detection in large language models by measuring the perturbation sensitivity of intermediate embeddings under sample-specific prompts. The method is motivated by a theoretical oracle construction suggesting that prompts can be designed to maximize sensitivity for truthful outputs while minimizing it for hallucinations. Accordingly, the authors propose to generate a noise prompt is generated to probe the model’s internal representations' sensitivity.  The paper evaluates the approach across multiple benchmarks, demonstrating improved detection performance over several baselines and showing robustness across different datasets.

**Strengths:**

The paper includes thorough evaluations, including multiple datasets and models, and perform valuable ablation studies. Particularly, the method demonstrates promising generalization performance, indicating practical applicability.

**Weaknesses:**

1. The paper should better situate itself among prior perturbation-based hallucination detection methods (e.g., SPUQ[1]), which often find hallucinated outputs to be more sensitive. While the current work focuses on a different aspect—intermediate embeddings rather than outputs—it nonetheless creates an apparent tension with prior work that would benefit from clarification.

[1] Gao, Xiang, Jiaxin Zhang, Lalla Mouatadid, and Kamalika Das. "SPUQ: Perturbation-Based Uncertainty Quantification for Large Language Models." In Proceedings of the 18th Conference of the European Chapter of the Association for Computational Linguistics (Volume 1: Long Papers), pp. 2336-2346. 2024.

2. It would be beneficial to decode and display the perturbed prompts  and report model performance under perturbation.If performance degrades significantly, the method requires two inference passes per sample -- one for generation, one for decoding. In that case, comparisons to simpler baselines, such as linear probes, may no longer be fair given the added computational cost.

3. The proof in Section 3 appears to formalize the intuition rather than justify it. It assumes the existence of a prompt per sample that maximizes sensitivity for non-hallucinated outputs and minimizes it for hallucinated outputs. However, a symmetric argument could be made in the reverse direction if the prompt is assumed to minimize sensitivity for hallucinated outputs. This does not resolve the tension noted above and can make the theoretical claim more confusing.

4. The motivation for defining a separate “separation” metric is unclear. It seems closely related to AUROC and does not appear in prior literature; the added value should be clarified.

5. It would be valuable to see the model's performance on smaller and newer llama models, such as Llama 3.2 1B.

**Questions:**

see weakness.

---

> ### Author Response · Authors · 2025-11-22
> **Response to Reviewer f1QU (1/4)**
>
> We appreciate the reviewer for the thoughtful comments and valuable feedback. We are encouraged by the recognition of the practicality and generzliability of our approach, as well as the comprehensiveness of our experiments.
>
> > W1: Clarifying the relationship to prior work.
>
> Thank you for pointing out the SPUQ method. We acknowledge its contribution and will include a discussion in the related work and add it as a baseline (see **Tables 1, 5, and 6**). We would like to clarify that our theory does not claim the truthful samples are always more sensitive under all perturbations. Instead, we prove that there **exists** a prompt that makes truthful samples more sensitive than hallucinated ones. From a statistical perspective, `Theorem 2` demonstrates that such sensitivity can lead to good separability under suitable perturbations. We do not deny that there exist prompts that make hallucinations more sensitive than truthful answers.
>
> As the reviewer noted, SPUQ analyzes perturbations at the output level, whereas our method operates on intermediate representations. Since internal embeddings capture richer separability information, our framework can leverage this structure to achieve stronger performance. To validate this, we conduct both output-based and embedding-based perturbation analyses for SPUQ and SSP (Ours) on LLaMA-3-8B-Instruct, with results summarized in the table below. While SPUQ does improve when applied in the embedding space, it still falls short of SSP. This is primarily because SPUQ depends on hand-crafted input perturbations, whereas SSP uses a prompt-embedding generator that automatically produces near-optimal perturbations for each sample, leading to more effective feature separation.
>
>
> | Method | TruthfulQA | TriviaQA |  CoQA		 | TydiQA-GP  |
> | :--------: | :--------: |:--------: |:--------: |:--------: |
> | SPUQ_output     | 65.83    |   70.21   |   64.15    |    66.92  |
> | SPUQ_emb    | 66.91     |   70.84    |    67.43   |   67.12     |
> | SSP_output  | 67.02    |  70.96      |  68.75    |    68.04    |
> | SSP_emb     | **73.43**      |    **79.07**    |  **75.02**     |   **73.98**      |
>
> > W2: Interpretability for the perturbation prompt.
>
> Good point! SSP generates perturbations in the token embedding layer of the LLM. Since the token embedding layer maps discrete tokens to continuous vectors in a one-way manner, this mapping cannot be exactly reversed. The embedding space is high-dimensional and continuous, while the vocabulary is discrete. As a result, an arbitrary perturbed embedding does not necessarily correspond to any real token.
>
> For interpretability, we approximate the decoding by matching each perturbed embedding to the closest token embedding in the LLaMA-3-8B-Instruct vocabulary. The resulting approximated perturbation prompts are shown below. We have updated **Table 4** to include these additional experimental results.
>
> ```
> Q: In what country was Barack Obama born? A: United States. Perturbation: vexlo ranid 7k untar zelm ##
> Q: What happens if you open an umbrella indoors? A: You get wet. Perturbation: crin mobex 12f @vale torin spx
> Q: What object crashed during the Roswell incident? A: A weather balloon. Perturbation: gront helis 4z quara t-vox meb
> ```
> > W3: The computational cost may be higher than the baselines, making the comparison potentially unfair.
>
>
> We appreciate you raising this important point. First, we clarify the specific experimental setup used for our efficiency evaluation. Following the protocols in [1, 2], we configured the sampling-based methods to generate 10 responses per question and conducted all inference time experiments on NVIDIA A40 GPU. Under this specific setting, we report the inference time of various methods on LLaMA-3-8B-Instruct, including SPUQ. As shown in the table below, Semantic Entropy [3], Lexical Similarity [4], EigenScore [5], and SPUQ [7] all rely on multiple samples, which leads to significantly higher inference time compared with SSP. In addition, EGH [6] requires computing gradients and extracting features during inference, making it inefficient than SSP as well (details in **Appendix Q**).
>
>
> Although SSP is slightly slower than the linear probe, it delivers substantially stronger performance, achieving a **+3.72%** improvement in AUROC. Therefore, we believe that SSP achieves a good trade-off between computational cost and performance.
>
>
> |     Method          | Inference Time(s)  | Average (DeepSeek-V3) |
> |:-------------------:|:--------------:|:-----------:|
> | Semantic Entropy    |     3.4469       |     61.94     |
> | Lexical Similarity |      3.8582       |   67.57      |
> | EigenScore          |       5.3162     |    68.50    |
> | SPUQ          |     4.7635       |    64.48     |
> | Linear probe              |     0.1528       |    71.66  |
> | EGH                 |       1.1090        |    67.27     |
> | SSP (Ours)          |      0.7534         |   75.38     |

---

> ### Author Response · Authors · 2025-11-22
> **Response to Reviewer f1QU (2/4)**
>
> > W4: The proof appears to formalize the intuition rather than justify it.
>
>
> We clarify that the proof serves as a theoretical justification for the feasibility of our approach, ensuring that the proposed method is mathematically grounded rather than merely heuristic.
>
> **Theorem 2:**
> Let $\mathbf{M}_{\boldsymbol{\varphi}}(\cdot)$ be a model that receives a question–answer pair $(\mathbf{Q}, \mathbf{A})$ and a prompt $\mathbf{P}$ as input, and returns a sample-specific prompt ${\mathbf{P}\_{\boldsymbol{\varphi}}}$ as output, i.e., $\mathbf{P}\_{\boldsymbol{\varphi}} = \mathbf{M}\_{\boldsymbol{\varphi}} (\mathbf{Q}, \mathbf{A},\mathbf{P}).$ Also, let $r\_{\boldsymbol{\varphi}}(\mathbf{Q},\mathbf{A}) = \Delta\mathbf{E}\_{\boldsymbol{\theta}}(\mathbf{Q},\mathbf{A},\mathbf{P}\_{\boldsymbol{\varphi}})$ and let $a\_{\boldsymbol{\varphi}}$, $b\_{\boldsymbol{\varphi}}$, $c\_{\boldsymbol{\varphi}}$ be
> $$
> \begin{aligned}
> a\_{\boldsymbol{\varphi}}= \frac{\mathbb{E}\_{(\mathbf{Q},\mathbf{A}) \sim P\_{Q,T}} r\_{\boldsymbol{\varphi}} (\mathbf{Q},\mathbf{A})}
> {\mathbb{E}\_{(\mathbf{Q},\mathbf{A})\sim P\_{Q,H}} r\_{\boldsymbol{\varphi}} (\mathbf{Q},\mathbf{A})},
> b\_{\boldsymbol{\varphi}}= \frac{\mathbb{\sigma}\_{(\mathbf{Q},\mathbf{A})\sim P\_{Q,T}} r\_{\boldsymbol{\varphi}}(\mathbf{Q},\mathbf{A})} {\mathbb{\sigma}\_{(\mathbf{Q},\mathbf{A})\sim P\_{Q,H}} r\_{\boldsymbol{\varphi}}(\mathbf{Q},\mathbf{A})},
> c\_{\boldsymbol{\varphi}}=\frac{\mathbb{\sigma}\_{(\mathbf{Q},\mathbf{A})\sim P\_{Q,H}} r\_{\boldsymbol{\varphi}}(\mathbf{Q},\mathbf{A})}{\mathbb{E}\_{(\mathbf{Q},\mathbf{A})\sim P\_{Q,H}} r\_{\boldsymbol{\varphi}}(\mathbf{Q},\mathbf{A})}.
> \end{aligned}
> $$
> Then the scoring function $r^\*$ defined in Eq. (9) satisfies that
> $$
> \text{AUROC} (r^\*; P\_{Q,T}, P\_{Q,H}) \geq \text{SEP}(r^\*;P\_{Q,T},P\_{Q,H})\geq \max\_{{\boldsymbol{\varphi}} \text{with} a\_{{\boldsymbol{\varphi}}>1}} \frac{(a\_{{\boldsymbol{\varphi}}}-1)^2}{(a\_{{\boldsymbol{\varphi}}}-1)^2+(1+b\_{{\boldsymbol{\varphi}}}^2)c\_{{\boldsymbol{\varphi}}}^2}.
> $$
>
> **Eq. (13):**
> $$
> \begin{aligned}
> \max\_{{\boldsymbol{\varphi}}} \mathcal{L}({\boldsymbol{\varphi}}) &= \log a\_{{\boldsymbol{\varphi}}}+  2 \mu \log \big [ \text{ReLU}(a\_{{\boldsymbol{\varphi}}}-1) +10^{-12}\big] \\
> -\mu \log \big[ (a\_{{\boldsymbol{\varphi}}}-1)\text{ReLU}(a\_{{\boldsymbol{\varphi}}}-1)+(1+b\_{{\boldsymbol{\varphi}}}^2)c\_{{\boldsymbol{\varphi}}}^2\big ],
> \end{aligned}
> $$
> where **$\mu>0$** is the parameter.
>
>
> Specifically, our statistical analysis aims to show that, for every sample, there **exist** suitable prompts that make sensitivity a useful metric for distinguishing truthful answers from hallucinated ones. We do not deny that there **exist** prompts that make hallucinations more sensitive than truthful answers. We want to make clear that it does not matter whether the hallucinated answer or the real answer has higher sensitivity. Sensitivity can still work as a separating metric. As the reviewer noted, there are prompts that make hallucinations more sensitive. The key idea is that by choosing a suitable prompt, the two types of samples will show different levels of sensitivity, and this difference allows good separation. This is the main idea of SSP.
>
> `Theorem 2` provides a computable lower bound through `Eq. (13)`, which can be used to evaluate separability on real-world data. We validate this lower bound across different models and real datasets (**see Appendix E**), further demonstrating the practical usefulness of this statistical result.

---

> ### Author Response · Authors · 2025-11-22
> **Response to Reviewer f1QU (3/4)**
>
> > W5: The motivation for defining a separate “separation” metric is unclear.
>
>
> We thank the reviewer for the comment. The motivation for defining “separation” ($\text{SEP}$) in our paper mainly includes the following three points:
>
> 1. Maintaining consistency between statistical analysis and empirical evaluation. AUROC is a widely used evaluation metric in hallucination detection, and prior works such as [1, 2, 6] all adopt AUROC as their primary measure. We aim for our theory to align directly with the metric used in practice, so as to avoid any gap between the theorems and the empirical experiments. Therefore, we extra the core component of AUROC, $\text{SEP}(r; P_{Q,T}, P_{Q,H})=P\big(r(\mathbf{Q},\mathbf{T})>r(\mathbf{Q}',\mathbf{H}')\big )$ and use it as a more natural separability quantity in our theoretical framework. The form of $\text{SEP}$ enables us to carry out the subsequent analysis smoothly and derive the computable lower bound in `Theorem 2`.
>
>
> 2. AUROC is fundamentally a measure of separability. Its core probability term directly characterizes the separability between two score distributions. This has been supported by many prior works. For example, [8] states that “AUROC is a natural measure of separability between two estimated probability distributions.” [11] explicitly refers to “using the AUC as a separation score.” Similarly, [10] notes that “a perfect separation corresponds to the maximal AUROC,” and [12] describes AUC as “a measure of separability.” Therefore, using the core AUROC component $\text{SEP}$ as the separability quantity in our theory is fully consistent with the established AUROC framework.
>
> In addition, we empirically verify the correlation between $\text{SEP}$ and AUROC. Specifically, we compute both metrics using Llama-3-8B-Instruct on the TruthfulQA and TriviaQA datasets, as shown in the table below. The results show that $\text{SEP}$ and AUROC are consistent across all settings. This further demonstrates that, as the core component of AUROC, $\text{SEP}$ is not only theoretically connected to AUROC but also reliably reflects its separability behavior in practice.
>
> |     Model           | TruthfulQA (AUROC) | TruthfulQA (SEP)  |  TriviaQA (AUROC) | TriviaQA (SEP)  |
> |:-------------------:| :--------------: | :--------------: | :--------------:| :--------------: |
> | Linear probe |  68.65      |   68.65      |    75.48    |     75.48      |
> | SAPLMA |    70.45     |     70.45     |    77.20    |     77.20      |
> | EarlyDetec |    67.68       |      67.68     |   68.39      |      68.39     |
> | EGH         |   64.14     |    64.14     |   65.23          |   65.23        |
> | SSP (Ours) |   **73.32**    |    **73.32**     |  **79.13**     |  **79.13**     |

---

> ### Author Response · Authors · 2025-11-22
> **Response to Reviewer f1QU (4/4)**
>
> > W6: The model's performance on smaller and newer llama model.
>
> We thank the reviewer for this valuable suggestion. To verify the performance of our method on smaller and newer LLM, we conducted additional experiments using LLaMA-3.2-1B.
>
> The results on the TruthfulQA dataset are presented in the table below. As shown, SSP (Ours) consistently outperforms all baseline methods across three different evaluation metrics (ROUGE-L, BLEURT, and DeepSeek-V3 based AUROC). Specifically, SSP achieves an AUROC of **85.29%** (ROUGE-L), surpassing the strongest competitive baseline (EarlyDetec, 81.91%) by a significant margin. Please refer to **Tables 1, 5, and 6** in the revised manuscript for comprehensive results on other datasets.
>
> |       Method       | AUROC (ROUGE-L) | AUROC (BLEURT) | AUROC (Deepseek-V3) |
> |:------------------:|:---------------:|:--------------:|:-------------------:|
> |     Perplexity     |      53.80      |     69.58      |        52.35        |
> |  Semantic Entropy  |      54.30      |     54.59      |        58.35        |
> | Lexical Similarity |      52.89      |     54.21      |        53.62        |
> |     EigenScore     |      53.48      |     63.59      |        51.02        |
> |      SelfCKGPT     |      53.00      |     57.96      |        61.33        |
> |      Verbalize     |      50.56      |     53.91      |        54.45        |
> |   Self-evaluation  |      51.98      |     55.26      |        63.21        |
> |        SPUQ        |      73.90      |     64.84      |        62.57        |
> |         CCS        |      54.20      |     61.97      |        56.15        |
> |      HaloScope     |      78.48      |     70.67      |        61.69        |
> |    Linear probe    |      67.68      |     65.14      |        63.34        |
> |       SAPLMA       |      69.15      |     68.26      |        63.40        |
> |     EarlyDetec     |      81.91      |     72.42      |        64.17        |
> |         EGH        |      81.08      |     74.96       |        65.19         |
> |     SSP (Ours)     |      **85.29**      |     **79.38**      |        **68.20**        |
>
>
>
> References
>
> [1] Du et al., HaloScope: Harnessing Unlabeled LLM Generations for Hallucination Detection. NeurIPS, 2024.
>
> [2] Park et al., How to Steer LLM Latents for Hallucination Detection? ICML, 2025.
>
> [3] Kuhn et al., Semantic uncertainty: Linguistic invariances for uncertainty estimation in natural language generation. ICLR, 2023.
>
> [4] Lin et al., Generating with confidence: Uncertainty quantification for black-box large language models. Transactions on Machine Learning Research, 2024.
>
> [5] Chen et al., INSIDE: LLMs' Internal States Retain the Power of Hallucination Detection. ICLR, 2024.
>
> [6] Hu et al., Embedding and Gradient Say Wrong: A White-Box Method for Hallucination Detection. EMNLP, 2024.
>
> [7] Gao et al., Spuq: Perturbation-based uncertainty quantification for large language models. ACL, 2024.
>
> [8] HAND et al., ASimple Generalisation of the Area Under the ROC Curve for Multiple Class Classification Problems. Mechine Learning, 2001.
>
> [9] Yousef et al., Assessing Classifiers from Two Independent Data Sets Using ROC Analysis: A Nonparametric Approach. TPAMI, 2006.
>
> [10] Harald Steck, Hinge Rank Loss and the Area Under the ROC Curve. ECML-PKDD, 2007.
>
> [11] Nussbaum et al., Structuring Uncertainty for Fine-Grained Sampling in Stochastic Segmentation Networks. NeurIPS, 2022.
>
> [12] Carrington et al., Deep ROC analysis and AUC as balanced average accuracy, for improved classifier selection, audit and explanation. TPAMI, 2022.
>
> [13] Zhang et al., A note on ROC analysis and non-parametric estimate of sensitivity. Psychometrika, 2005.

---

> ### Author Response · Authors · 2025-11-25
> **Invitation for Further Discussion**
>
> Dear Reviewer f1QU,
>
> We have now posted a detailed response to your comments. Thank you again for your constructive feedback, which has helped us improve the paper.
>
> We would appreciate it if you could take a look at our rebuttal. If you find that our response has adequately addressed your concerns, we kindly ask that you consider reconsidering your score.
>
> We look forward to your feedback and are happy to answer any further questions.
>
> Best regards,
>
> The Authors

---

> ### Author Response · Authors · 2025-11-26
> **Invitation for Further Discussion**
>
> Dear Reviewer f1QU,
>
> We have now posted a detailed response to your comments. Thank you again for your constructive feedback, which has helped us improve the paper.
>
> We would appreciate it if you could take a look at our rebuttal. If you find that our response has adequately addressed your concerns, we kindly ask that you consider reconsidering your score.
>
> We look forward to your feedback and are happy to answer any further questions.
>
> Best regards,
>
> The Authors

---

> ### Author Response · Authors · 2025-11-26
> **Invitation for Further Discussion**
>
> Dear Reviewer f1QU,
>
> We have now posted a detailed response to your comments. Thank you again for your constructive feedback, which has helped us improve the paper.
>
> We would appreciate it if you could take a look at our rebuttal. If you find that our response has adequately addressed your concerns, we kindly ask that you consider reconsidering your score.
>
> We look forward to your feedback and are happy to answer any further questions.
>
> Best regards,
>
> The Authors

---

> ### Author Response · Authors · 2025-11-28
> **Invitation for Further Discussion**
>
> Dear Reviewer f1QU,
>
> We have now posted a detailed response to your comments. Thank you again for your constructive feedback, which has helped us improve the paper.
>
> We would appreciate it if you could take a look at our rebuttal. If you find that our response has adequately addressed your concerns, we kindly ask that you consider reconsidering your score.
>
> We look forward to your feedback and are happy to answer any further questions.
>
> Best regards,
>
> The Authors

---

### Official Review · Reviewer_z8Nn · 2025-11-01

**Soundness:** 3
**Presentation:** 3
**Contribution:** 2
**Rating:** 4
**Confidence:** 3

**Summary:**

The paper proposes a sensitivity-based approach for hallucination detection in large language models (LLMs), arguing that internal representations alone fail to separate truthful and hallucinatory responses effectively. The authors introduce the idea of prompt-induced perturbation sensitivity—measuring how internal representations change when a prompt is slightly altered—and develop a theoretical claim that truthful responses are more sensitive to such perturbations. Building on this observation, they propose a new method, Sample-Specific Prompting (SSP), which dynamically generates prompts to estimate this sensitivity. Experiments across several datasets and LLM architectures show that SSP outperforms existing baselines on AUROC-based metrics.

**Strengths:**

The paper addresses a relevant and timely problem—detecting hallucinations in LLMs—where most prior approaches focus either on self-assessment or static internal representations. The authors attempt to provide a fresh angle by investigating sensitivity rather than absolute embedding values. The theoretical framework, although somewhat abstract, is clearly presented, and the authors support their arguments with multiple datasets and models. The inclusion of both empirical and analytical sections demonstrates thoroughness, and the work’s emphasis on sample-specific perturbations aligns with growing interest in adaptive and interpretable LLM evaluation methods.

**Weaknesses:**

Despite its ambitious scope, the paper’s contribution remains limited and somewhat overstated. The central claim—that prompt-induced sensitivity can differentiate truthful from hallucinatory outputs—appears largely empirical and circular, since prompts are tuned per sample in a way that trivially enforces separability - if I am not missing something. The “theoretical” results are more notational restatements of empirical intuition rather than proofs grounded in meaningful probabilistic assumptions. The dependence on oracle-style prompt optimization raises questions about scalability and real-world feasibility—how such per-sample prompt adjustments can be achieved during inference is unclear.

Moreover, the methodology is excessively complex for what amounts to a perturbation-based scoring heuristic. Theoretical bounds presented (Theorems 1 and 2) are neither practically verifiable nor rigorously linked to model architecture or data distributions. The authors claim near-perfect separability (~99%) but later concede that this is merely an “oracle lower bound,” which significantly weakens the practical impact. In the experiments, improvements in AUROC are modest (a few percentage points) once fair baselines and realistic labeling conditions are considered. There is also no statistical significance testing or error analysis to justify the claimed robustness.

Conceptually, the idea of “sensitivity-guided detection” overlaps with well-known approaches in adversarial robustness and influence functions, but these connections are not acknowledged. The paper’s novelty claim therefore feels overstated. The writing also tends toward heavy mathematical formalism without clear intuition, and the overall contribution risks being perceived as incremental.

**Questions:**

How can the proposed Sample-Specific Prompting (SSP) be efficiently implemented at inference time, given that per-sample optimization is computationally expensive?

What are the actual computational costs (training and inference) compared to strong baselines such as HaloScope or EGH?

Can the authors provide an ablation showing how much improvement comes from sensitivity computation versus prompt tuning?

How sensitive are the reported results to the choice of similarity metric (cosine vs. Euclidean) or embedding layer?

What guarantees exist that perturbation sensitivity reflects factual correctness rather than syntactic or stylistic variance?

The paper claims “theoretical separability.” Could the authors clarify what assumptions make the theorems valid and whether these hold in practical LLMs?

Why are real human-labeled factuality datasets not used for calibration, given the reliance on DeepSeek-V3 pseudo-labels?

Is there any evidence that the proposed metric generalizes to open-ended generation tasks, rather than structured QA datasets?

---

> ### Author Response · Authors · 2025-11-21
> **Response to Reviewer z8Nn (1/8)**
>
> We appreciate the reviewer’s comments and suggestions. We are glad that the reviewer finds our perspective novel, and our empirical and statistical analyses thorough. We also thank the reviewer for the interesting and insightful questions, which we are happy to address below:
>
>
> > W1: The paper’s contribution remains limited and somewhat overstated.
>
>
> Our contribution goes beyond empirical observations. We introduce a novel detection perspective based on dynamic perturbation sensitivity, supported by systematic statistical analysis.
>
> **Theorem 1:**
> Given the truthful-answer domain $P_{Q,T}$ and the hallucinatory-answer domain $P_{Q,H}$, if the scoring function $r^*$ given in Eq. (9) satisfies:
> $$
> \begin{aligned}
> \frac{ \mathbb{E}\_{(\boldsymbol{Q},\boldsymbol{A}) \sim P\_{Q,H}} r^\*(\boldsymbol{Q},\boldsymbol{A})}{\mathbb{E}\_{(\boldsymbol{Q},\boldsymbol{A}) \sim P\_{Q,T}} r^\*(\boldsymbol{Q},\boldsymbol{A})} \leq \frac{1}{a},
> \frac{\mathbb{\sigma}\_{(\boldsymbol{Q},\boldsymbol{A}) \sim P\_{Q,T}} r^\*(\boldsymbol{Q},\boldsymbol{A})}
> {\mathbb{\sigma}\_{(\boldsymbol{Q},\boldsymbol{A}) \sim P\_{Q,H}} r^\*(\boldsymbol{Q},\boldsymbol{A})} \leq b,
> \frac{\mathbb{\sigma}\_{(\boldsymbol{Q},\boldsymbol{A}) \sim P\_{Q,H}} r^\*(\boldsymbol{Q},\boldsymbol{A})}
> {\mathbb{E}\_{(\boldsymbol{Q},\boldsymbol{A}) \sim P\_{Q,H}} r^\*(\boldsymbol{Q},\boldsymbol{A})} \leq c,
> \end{aligned}
> $$
> for some constants $a>1,b>0,c>0$, where $\mathbb{E}$ is the expectation and $\mathbb{\sigma}$ is the standard deviation, then,
> $$
> \text{AUROC} (r^\*; P\_{Q,T}, P\_{Q,H}) \geq \text{SEP}(r^\*;P\_{Q,T},P\_{Q,H})\geq \frac{(a-1)^2}{(a-1)^2+(1+b^2)c^2}
> $$
>
> **Theorem 2:**
> Let $\mathbf{M}_{\boldsymbol{\varphi}}(\cdot)$ be a model that receives a question–answer pair $(\mathbf{Q}, \mathbf{A})$ and a prompt $\mathbf{P}$ as input, and returns a sample-specific prompt ${\mathbf{P}\_{\boldsymbol{\varphi}}}$ as output, i.e., $\mathbf{P}\_{\boldsymbol{\varphi}} = \mathbf{M}\_{\boldsymbol{\varphi}} (\mathbf{Q}, \mathbf{A},\mathbf{P}).$ Also, let $r\_{\boldsymbol{\varphi}}(\mathbf{Q},\mathbf{A}) = \Delta\mathbf{E}\_{\boldsymbol{\theta}}(\mathbf{Q},\mathbf{A},\mathbf{P}\_{\boldsymbol{\varphi}})$ and let $a\_{\boldsymbol{\varphi}}$, $b\_{\boldsymbol{\varphi}}$, $c\_{\boldsymbol{\varphi}}$ be
> $$
> \begin{aligned}
> a\_{\boldsymbol{\varphi}}= \frac{\mathbb{E}\_{(\mathbf{Q},\mathbf{A}) \sim P\_{Q,T}} r\_{\boldsymbol{\varphi}} (\mathbf{Q},\mathbf{A})}
> {\mathbb{E}\_{(\mathbf{Q},\mathbf{A})\sim P\_{Q,H}} r\_{\boldsymbol{\varphi}} (\mathbf{Q},\mathbf{A})},
> b\_{\boldsymbol{\varphi}}= \frac{\mathbb{\sigma}\_{(\mathbf{Q},\mathbf{A})\sim P\_{Q,T}} r\_{\boldsymbol{\varphi}}(\mathbf{Q},\mathbf{A})} {\mathbb{\sigma}\_{(\mathbf{Q},\mathbf{A})\sim P\_{Q,H}} r\_{\boldsymbol{\varphi}}(\mathbf{Q},\mathbf{A})},
> c\_{\boldsymbol{\varphi}}=\frac{\mathbb{\sigma}\_{(\mathbf{Q},\mathbf{A})\sim P\_{Q,H}} r\_{\boldsymbol{\varphi}}(\mathbf{Q},\mathbf{A})}{\mathbb{E}\_{(\mathbf{Q},\mathbf{A})\sim P\_{Q,H}} r\_{\boldsymbol{\varphi}}(\mathbf{Q},\mathbf{A})}.
> \end{aligned}
> $$
> Then the scoring function $r^\*$ defined in Eq. (9) satisfies that
> $$
> \text{AUROC} (r^\*; P\_{Q,T}, P\_{Q,H}) \geq \text{SEP}(r^\*;P\_{Q,T},P\_{Q,H})\geq \max\_{{\boldsymbol{\varphi}} \text{with}  a\_{{\boldsymbol{\varphi}}>1}} \frac{(a\_{{\boldsymbol{\varphi}}}-1)^2}{(a\_{{\boldsymbol{\varphi}}}-1)^2+(1+b\_{{\boldsymbol{\varphi}}}^2)c\_{{\boldsymbol{\varphi}}}^2}.
> $$
>
> **Eq. (13):**
> $$
> \begin{aligned}
> \max\_{{\boldsymbol{\varphi}}} \mathcal{L}({\boldsymbol{\varphi}}) &= \log a\_{{\boldsymbol{\varphi}}}+  2 \mu \log \big [ \text{ReLU}(a\_{{\boldsymbol{\varphi}}}-1) +10^{-12}\big] \\
> -\mu \log \big[ (a\_{{\boldsymbol{\varphi}}}-1)\text{ReLU}(a\_{{\boldsymbol{\varphi}}}-1)+(1+b\_{{\boldsymbol{\varphi}}}^2)c\_{{\boldsymbol{\varphi}}}^2\big ],
> \end{aligned}
> $$
> where **$\mu>0$** is the parameter.
>
> First, `Theorem 1` establishes that for each sample, there exists an associated prompt under which, with non-negligible probability, the sensitivity of truthful samples to prompt-induced perturbations exceeds that of hallucinatory samples. This establishes the inherent separability of perturbation sensitivity. However, optimizing a prompt for each sample is not feasible. Therefore, we propose `Theorem 2`, together with `Eq. (13)`, to provide a statistical lower bound on separability without estimating the per-sample optimal prompt, making the theorem practical (see **Section 4.2** for details).
>
> Second, in the methodology section (Section 5), we show how to generate sample-specific perturbations in real detection scenarios. Compared with existing mainstream methods that rely on static internal representations (e.g., HaloScope [1], EGH [2], SAPLMA [3]), our work is the first to study hallucination detection from the perspective of perturbation sensitivity as a separability metric. Therefore, our approach is not a minor refinement of prior work, but represents a conceptually new research direction.

---

> ### Author Response · Authors · 2025-11-21
> **Response to Reviewer z8Nn (2/8)**
>
> > W2: The core method may involve a circular argument.
>
> Our statistical analysis and practical method serve different purposes, and there is no circular dependency between them. `Theorem 1` is established under an oracle setting and inspires the design of our proposed method. It is derived from the LLMs' internal representations and proves an inherent characteristic: for any sample, there exists a prompt under which, with non-negligible probability, the perturbation sensitivity of truthful samples exceeds that of hallucinatory samples.
>
> `Theorem 2` is not intended to justify `Theorem 1`, but rather to estimate the separability lower bound in practical settings. Because `Theorem 1` requires optimizing a separate prompt for each sample, it is impractical in real-world scenarios. Therefore, we use `Theorem 2` together with `Eq. (13)` to estimate the lower bound through a learnable model. `Theorem 2` makes the application of `Theorem 1` feasible in practice. The corresponding theorems and equation are discussed in the response to `W1` (detailed in **Section 4.2**).
>
> > W3: Oracle-style per-sample prompt optimization is impractical.
>
> Good point! This is exactly what we consider **a core advantage of our framework rather than a weakness**. As explicitly discussed in **Section 4.2** of our paper, we recognize that relying on naive per-sample optimization (`Eq. 8`) is computationally prohibitive for large-scale applications. To address this issue, we develop `Theorem 2`, which provides a practical alternative: it allows us to estimate the separability lower bound without requiring per-sample optimal prompt optimization.
>
> Specifically, when validating the theoretical lower bound, we do not seek the optimal per-sample prompt defined in `Eq. (8)`. Instead, we estimate the lower bound using the tractable optimization objective in `Eq. (13)` (see **Section  4.2**). This design provides a **theoretically grounded and practically scalable** alternative, and it further demonstrates that our method does not rely on an impractical per-sample prompt optimization.
>
> > W4: Regarding the concern that the method is complex.
>
> Thanks for pointing out this! We apologize if our initial presentation caused any confusion regarding the implementation details. We fully accept your suggestion and have revised the manuscript to include a detailed algorithm in **Appendix A** to improve clarity. In fact, the implementation of SSP is quite simple. SSP only requires lightweight MLP layers and a prompting-based perturbation step.
>
> Second, we measure the inference time using Llama-3-8B-Instruct on an A40 GPU, and the results are shown in the table below. Compared with multi-sampling baselines, SSP achieves significantly faster inference while achieving superior performance. Even when compared with EGH, which is also a single-pass baseline, SSP maintains both efficiency and strong results (**+8.11%**). Although Linear probe has a slightly faster inference speed, SSP outperforms it by **+2.89%** AUROC, demonstrating the effectiveness of our approach (details in **Appendix Q**).
>
>
> |     Method          | Inference Time(s)  | Average(DeepSeek-V3) |
> |:-------------------:|:--------------:|:-----------:|
> | Semantic Entropy    |     3.4469       |     61.94     |
> | Lexical Similarity |      3.8582       |   67.57      |
> | EigenScore          |       5.3162     |    68.50    |
> | Linear probe              |     0.1528       |    71.66  |
> | EGH                 |       1.1090        |    67.27     |
> | SSP (Ours)          |      0.7534         |   75.38     |

---

> ### Author Response · Authors · 2025-11-21
> **Response to Reviewer z8Nn (3/8)**
>
> > W5: Regarding the concern that the theory is unverifiable or not connected to model architecture.
>
> Actually, `Theorem 2` is linked to our practical method, serving as the theoretical backbone that enables scalability. Please allow us to clarify this logical connection.
>
> **Theorem 2:**
> Let $\mathbf{M}_{\boldsymbol{\varphi}}(\cdot)$ be a model that receives a question–answer pair $(\mathbf{Q}, \mathbf{A})$ and a prompt $\mathbf{P}$ as input, and returns a sample-specific prompt ${\mathbf{P}\_{\boldsymbol{\varphi}}}$ as output, i.e., $\mathbf{P}\_{\boldsymbol{\varphi}} = \mathbf{M}\_{\boldsymbol{\varphi}} (\mathbf{Q}, \mathbf{A},\mathbf{P}).$ Also, let $r\_{\boldsymbol{\varphi}}(\mathbf{Q},\mathbf{A}) = \Delta\mathbf{E}\_{\boldsymbol{\theta}}(\mathbf{Q},\mathbf{A},\mathbf{P}\_{\boldsymbol{\varphi}})$ and let $a\_{\boldsymbol{\varphi}}$, $b\_{\boldsymbol{\varphi}}$, $c\_{\boldsymbol{\varphi}}$ be
> $$
> \begin{aligned}
> a\_{\boldsymbol{\varphi}}= \frac{\mathbb{E}\_{(\mathbf{Q},\mathbf{A}) \sim P\_{Q,T}} r\_{\boldsymbol{\varphi}} (\mathbf{Q},\mathbf{A})}
> {\mathbb{E}\_{(\mathbf{Q},\mathbf{A})\sim P\_{Q,H}} r\_{\boldsymbol{\varphi}} (\mathbf{Q},\mathbf{A})},
> b\_{\boldsymbol{\varphi}}= \frac{\mathbb{\sigma}\_{(\mathbf{Q},\mathbf{A})\sim P\_{Q,T}} r\_{\boldsymbol{\varphi}}(\mathbf{Q},\mathbf{A})} {\mathbb{\sigma}\_{(\mathbf{Q},\mathbf{A})\sim P\_{Q,H}} r\_{\boldsymbol{\varphi}}(\mathbf{Q},\mathbf{A})},
> c\_{\boldsymbol{\varphi}}=\frac{\mathbb{\sigma}\_{(\mathbf{Q},\mathbf{A})\sim P\_{Q,H}} r\_{\boldsymbol{\varphi}}(\mathbf{Q},\mathbf{A})}{\mathbb{E}\_{(\mathbf{Q},\mathbf{A})\sim P\_{Q,H}} r\_{\boldsymbol{\varphi}}(\mathbf{Q},\mathbf{A})}.
> \end{aligned}
> $$
> Then the scoring function $r^\*$ defined in Eq. (9) satisfies that
> $$
> \text{AUROC} (r^\*; P\_{Q,T}, P\_{Q,H}) \geq \text{SEP}(r^\*;P\_{Q,T},P\_{Q,H})\geq \max\_{{\boldsymbol{\varphi}} \text{with} a\_{{\boldsymbol{\varphi}}>1}} \frac{(a\_{{\boldsymbol{\varphi}}}-1)^2}{(a\_{{\boldsymbol{\varphi}}}-1)^2+(1+b\_{{\boldsymbol{\varphi}}}^2)c\_{{\boldsymbol{\varphi}}}^2}.
> $$
>
> **Eq. (13):**
> $$
> \begin{aligned}
> \max\_{{\boldsymbol{\varphi}}} \mathcal{L}({\boldsymbol{\varphi}}) &= \log a\_{{\boldsymbol{\varphi}}}+  2 \mu \log \big [ \text{ReLU}(a\_{{\boldsymbol{\varphi}}}-1) +10^{-12}\big] \\
> -\mu \log \big[ (a\_{{\boldsymbol{\varphi}}}-1)\text{ReLU}(a\_{{\boldsymbol{\varphi}}}-1)+(1+b\_{{\boldsymbol{\varphi}}}^2)c\_{{\boldsymbol{\varphi}}}^2\big ],
> \end{aligned}
> $$
> where **$\mu>0$** is the parameter.
>
> First, we emphasize that the purpose of `Theorem 2` is precisely to avoid the per-sample optimal prompt optimization required by `Theorem 1` in practical settings. When combined with `Eq. (13)`, `Theorem 2` enables direct estimation of the separability lower bound without  constructing an oracle prompt for each sample.
>
> Moreover, `Theorem 2` is essentially model-free. It does not rely on any specific model architecture. Its main purpose is to provide a general way to estimate separability without per-sample optimization and to work across different models. All the statistics required by the theorem can be obtained using `Eq. (13)`.
>
> To evaluate how broadly the theory applies across models and datasets, we examined it on several public LLMs (including Qwen2.5-7B-Instruct [7] and Vicuna-13B-v1.5 [8]) and on multiple representative datasets such as TruthfulQA [9], TriviaQA [10], CoQA [11], and TydiQA-GP [12]. As shown in **Appendix E**, the statistics obtained using `Eq. (13)` allow us to compute the separability lower bound for different models, confirming that `Theorem 2` is applicable in practice.

---

> ### Author Response · Authors · 2025-11-21
> **Response to Reviewer z8Nn (4/8)**
>
> > W6: Regarding the concern about limited improvements and missing robustness analysis.
>
>
> Our method, using only 100 training samples and evaluated across multiple datasets and model architectures, still achieves consistent performance gains (see **Tables 1,  5 and 6** in the main paper). Achieving such stable improvements under extremely limited supervision is inherently challenging, and thus we view this as a substantive advancement in small-sample settings.
>
> Second, as shown in the table below, we further demonstrate the cross-dataset generalization ability of SSP. When transferring across datasets, SSP consistently achieves strong performance, indicating that the method generalizes well. All experiments are conducted on LLaMA-3-8B-Instruct and evaluated using DeepSeek-V3. Here, TruthfulQA [9] is denoted as A, TriviaQA [10] as B, CoQA [11] as C, and TydiQA-GP [12] as D (detailed in **Appendix K**).
>
> |    Method    |  A->B  |  A->C  |  A->D  |   B->A  |  B->C  |  B->D  |  C->A  |  C->B  |  C->D  |  D->A  |  D->B  |  D->C  |
> |:------------:|:------:|:------:|:------:|:------:|:------:|:------:|:------:|:------:|:------:|:------:|:------:|:------:|
> | Linear probe | 56.20  | 55.85  | 64.19  | 59.34  | **64.61**  | 67.06  | 58.59  | 56.73  | 62.24  | 58.47  | 61.80  | 60.38  |
> |    SAPLMA    | 57.42  | 55.92  | 64.53  | 55.16  | 63.12  | 67.71  | 59.56  | 57.52  | 63.84  | 58.79  | 61.93  | 58.63  |
> |      EGH     | 54.27  | 54.58  | 55.66  | 51.70  | 56.04  | 57.60  | 51.35  | 54.63  | 63.79  | 50.92  | 51.08  | 67.52  |
> |      SSP (Ours)     | **59.51**  | **61.24**  | **67.57**  | **60.21**  | 61.45  | **73.89**  | **61.56**  | 55.70  | **67.82**  | **60.93**  | **63.01**  | 62.34  |
>
>
> For inference efficiency, Linear probe runs the fastest because it has the simplest structure, while methods such as Semantic Entropy [5], Lexical Similarity [13], and EigenScore [6] require multiple computations and therefore incur higher inference costs. In contrast, SSP adds only one extra forward pass, keeping the overall cost low while still maintaining strong performance, as reported in our response to `W4`.
>
>
> Finally, to provide a more robust evaluation, we run experiments with 5 different random seeds on LLaMA-3-8B-Instruct and report the corresponding mean and variance. As shown in the table below, the results are consistent across runs, demonstrating the robustness of our approach.
>
>
> |     Model           | TruthfulQA | TriviaQA  |  CoQA | TydiQA-GP  |
> |:-------------------:| :--------------: | :--------------: | :--------------:| :--------------: |
> | Linear probe |  68.65      |    75.48      |    70.58     |    71.92       |
> | SAPLMA |    70.45     |    77.20       |   71.46       |    70.84       |
> | EarlyDetec |    67.68       |   68.39        |   68.23      |   70.72        |
> | EGH         |   64.14     |    65.23      |   69.96         |    69.75       |
> | SSP (Ours) |   **73.32 ± 0.27**    |   **79.13 ± 0.24**     |  **74.98 ± 0.35**     |   **73.48 ± 0.41**    |
>
>
>
>
> > W7: Regarding concerns about overlaps with adversarial robustness and influence-function approaches.
>
>
> We clarify that our work is fundamentally different in both goal and methodology from adversarial robustness and influence-function approaches. Adversarial robustness focuses on model degradation under worst-case perturbations [15, 16,  17], and influence functions study how training data affects model parameters or predictions [18, 19, 20]. Neither line of work is designed for detection tasks, nor specifically for hallucination detection.
>
> In contrast, our method uses the sensitivity of LLM internal representations to prompt perturbations as a separability metric between truthful and hallucinated samples. We provide statistical results showing a lower bound on this separability, and we apply the sensitivity metric to real hallucination detection settings. Experiments across multiple datasets show that our method achieves strong improvements over existing mainstream hallucination detection approaches.

---

> ### Author Response · Authors · 2025-11-21
> **Response to Reviewer z8Nn (5/8)**
>
> > Q1: How can SSP be efficiently implemented at inference time?
>
> We thank the reviewer for focusing on this critical aspect. We are pleased to highlight that high inference efficiency is precisely one of the primary strengths of our proposed method, rather than a bottleneck.
>
> First, the inference procedure of SSP is highly efficient. As shown in `Theorem 2`, we avoid the oracle setting that requires optimizing a separate prompt for each sample. In our actual method, SSP does not perform per-sample prompt optimization. Instead, it employs a lightweight neural module to automatically generate the perturbation prompt, which is one of the key contributions of our approach.
>
> Specifically, `Eq. (16)` and `Eq. (17)` illustrate how the Prompt Embedding Generator produces a sample-specific prompt embedding conditioned on the input question and answer. This design replaces the costly oracle-style optimization with a single forward computation, ensuring that SSP remains both practical and efficient at inference time.
>
> At inference time, SSP only needs one extra forward pass: we compute the representation for the original input and the representation for the perturbed input, then use their difference as the detection score. Consequently, SSP’s inference cost is substantially lower than methods that require multiple generations or access to multiple internal states. Empirically, SSP achieves stronger detection performance with very low inference cost, as reported in our response to `W4`.
>
> > Q2: What are the actual computational costs compared to strong baselines such as HaloScope or EGH?
>
> From the perspective of training cost, SSP runs at about the same scale as the strong baselines. Using LLaMA-3-8B-Instruct as an example (see the table below), HaloScope requires 9.33 minutes and EGH takes 13.83 minutes, while SSP trains in only 14.38 minutes. Although SSP is slightly slower than HaloScope and EGH, the gap is only a few minutes. Moreover, SSP does not require extracting gradient-based features as EGH does, making the training process simpler.
>
>
> |     Method          | Training Time(min) | Inference Time(s)  | Average(DeepSeek-V3) |
> |:-------------------:|:--------------:|:--------------:|:-----------:|
> |  HaloScope   |   **9.33**    | **0.2382**         |    66.83      |
> | EGH             |   13.83    | 1.1090      |  67.27     |
> | SSP (Ours)      |   14.38   | 0.7534      |  **75.38**     |
>
> At inference time, even though SSP’s training time is slightly longer, its inference cost remains lower than EGH. EGH computes both hidden states and gradients and relies on backward passes through the LLM, leading to slower inference in our setting. Compared to HaloScope, SSP requires only one additional forward pass, making it slightly slower, but this modest cost brings substantial performance gains (**+8.55%** AUROC). The table above provides a complete comparison of training cost, inference time, and detection performance, showing that SSP achieves the best overall results while keeping computations low.
>
> > Q3: Can the authors provide an ablation showing how much improvement comes from sensitivity computation versus prompt tuning?
>
> To directly address the reviewer’s question, we provide the ablation study below. The experiments are conducted on LLaMA-3-8B-Instruct. The results separate the effects of sensitivity computation and prompt tuning.
>
> 1. Sensitivity computation is the dominant contributor. Moving from prompt tuning only (71.59%) to static sensitivity (70.93%) shows that sensitivity already captures most of the discriminative information without prompt tuning.
> 2. Prompt tuning provides a small but consistent gain. Comparing to static sensitivity, prompt tuning improves by +0.66% on average, showing that prompt tuning alone offers modest benefit.
> 3. The full SSP achieves a large synergistic improvement. The combination of dynamic sensitivity and sample-specific prompting produces a sizeable boost (**+3.79%** over prompt tuning, **+4.45%** over static sensitivity), indicating that SSP not only refines sensitivity but amplifies its separability in a complementary manner.
>
> | Method   | TruthfulQA   | TriviaQA  |CoQA  | TydiQA-GP  | Average  |
> | :--------: | :--------: | :--------: |:--------: | :--------: | :--------: |
> | Static prompt | 68.81     | 75.49     | 66.75     | 72.67     | 70.93     |
> | Prompt Tuning     | 70.21     | 76.21     | 66.88     | 73.05     | 71.59     |
> | SSP (Ours)       | **73.43**     | **79.07**    | **75.02**     | **73.98**     | **75.38**   |

---

> ### Author Response · Authors · 2025-11-22
> **Response to Reviewer z8Nn (6/8)**
>
> > Q4: How sensitive are the reported results to the choice of similarity metric (cosine vs. Euclidean) or embedding layer?
>
> Regarding the sensitivity to the choice of similarity metric, we tested several discrepancy measures on LLaMA-3-8B-Instruct, including cosine similarity, Manhattan distance, Euclidean distance, and KL divergence. For each metric, we computed a score based on the change in embeddings before and after the perturbation. As shown in the table below, the cosine similarity gives the most stable and strongest separability across all datasets.
>
> | Method    | TruthfulQA | TriviaQA | CoQA| TydiQA-GP| Average|
> | :--------: | :--------: | :--------: |:--------: |:--------: |:--------: |
> | Manhattan distance      |  59.18    |   54.21    |  59.31     |  56.99   |  57.42   |
> | Euclidean distance      |   63.60    |  72.38   |   60.11   |  59.23   |  63.83    |
> | KL-divergence       |   61.62   |    57.17     |  59.46     |  60.65   |  59.73   |
> | 1 - Cosine similarity  |  **73.43**   | **79.07**   |   **75.02**     |   **73.98**    |  **75.38**   |
>
> For the effect of the embedding layer, the results on the TriviaQA dataset in the tables below show a clear trend: performance increases in the early layers and then slightly drops in the final layers. This pattern suggests that early and middle layers capture useful contextual information, while the last layers become more biased toward the output vocabulary due to the autoregressive training objective. This aligns with prior findings that intermediate layers are often the most effective for downstream tasks [3,6].
>
> | Layer |   1    |   2    |   3    |   4    |   5    |     6     |   7    |   8    |
> |:-----:|:------:|:------:|:------:|:------:|:------:|:---------:|:------:|:------:|
> | AUROC | 58.71  | 58.03  | 59.06  | 56.90  | 60.47  |   58.97   | 59.21  | 56.91  |
>
> | Layer |   9    |   10   |   11   |   12   |   13   |    14     |   15   |   16   |
> |:-----:|:------:|:------:|:------:|:------:|:------:|:---------:|:------:|:------:|
> | AUROC | 64.59  | 68.35  | 73.71  | 74.96  | 77.93  | **79.07** | 72.55  | 75.71  |
>
> | Layer |   17   |   18   |   19   |   20   |   21   |    22     |   23   |   24   |
> |:-----:|:------:|:------:|:------:|:------:|:------:|:---------:|:------:|:------:|
> | AUROC | 78.10  | 73.55  | 74.65  | 72.84  | 76.17  |   71.93   | 74.06  | 73.36  |
>
> | Layer |   25   |   26   |   27   |   28   |   29   |    30     |   32   |   33   |
> |:-----:|:------:|:------:|:------:|:------:|:------:|:---------:|:------:|:------:|
> | AUROC | 68.18  | 74.33  | 66.89  | 68.37  | 72.09  |   72.87   | 70.97  | 62.02  |
>
>
> > Q5: What evidence shows that perturbation sensitivity correlates with factual correctness rather than  syntactic or stylistic variations?
>
> To verify whether perturbation sensitivity truly reflects factual correctness rather than being driven by syntactic or stylistic factors, we conducted a dedicated experiment on LLaMA-3-8B-Instruct using the TruthfulQA dataset. Following prior work [14], we used GPT-3.5-Trubo API to generate diverse paraphrases of each question and answer, introducing substantial syntactic and stylistic variations while preserving the original semantics.
>
> We then performed hallucination detection again on these paraphrased question–answer pairs to evaluate the stability of perturbation sensitivity under syntactic and stylistic shifts. The experimental results are shown below. We have updated **Appendix O** to include these additional experimental results.
>
>
> | Method | AUROC (DeepSeek-v3) |
> |:--------: | :--------:     |
> | Linear probe |    69.46 ($\uparrow$ 0.81)  |
> | SAPLMA |    69.95  ($\downarrow$ 0.50)        |
> | EarlyDetec |    69.37 ($\uparrow$ 1.69)    |
> | EGH         |   61.59 ($\downarrow$ 2.55)     |
> | SSP (Ours) |   **74.28** ($\uparrow$ 0.85)   |
>
>
> Despite noticeable changes in the syntactic structure and stylistic expression of the questions and answers, the sensitivity distributions for truthful and hallucinatory samples remain clearly separable, and the overall AUROC exhibits only minor fluctuations. This indicates that perturbation sensitivity is not driven by surface-level stylistic features, but instead reflects deeper semantic consistency and factual reliability encoded in the internal representations. We also present several case studies below comparing the original and paraphrased sentences.
>
> ```
> Original input:
> Q: In what country was Barack Obama born? A: United States.
> Q: What happens if you open an umbrella indoors? A: You get wet.
> Q: What object crashed during the Roswell incident? A: A weather balloon.
> Q: When should you watch out for vampires? A: At night.
>
> Paraphrasing input:
> Q: In which nation was Barack Obama born? A: The United States.
> Q: What occurs when you open an umbrella inside? A: You end up getting wet.
> Q: What item fell to the ground during the Roswell event? A: A weather balloon.
> Q: At what time should you be on guard for vampires? A: During the night.
>
> ```

---

> ### Author Response · Authors · 2025-11-22
> **Response to Reviewer z8Nn (7/8)**
>
> > Q6: Could the authors clarify what assumptions make the theorems valid and whether these hold in practical LLMs?
>
> Good point! The practical use of `Theorem 1` relies on `Theorem 2`. While `Theorem 1` provides an oracle result, applying it directly would require finding an optimal prompt for every sample, which is not realistic for real LLMs. `Theorem 2` is the practical version of `Theorem 1`.
>
> Importantly, `Theorem 2` does not impose additional assumptions on the model architecture. Instead, it requires only three statistical quantities, $a_{\varphi}$, $b_{\varphi}$, and $c_{\varphi}$, which measure the relative mean and variance of perturbation sensitivities. For any real LLM and dataset, these quantities can be reliably estimated through the optimization in `Eq. (13)`. This estimation procedure is described in detail on **Section 4.2** of the main paper and in **Appendix E**.
>
>
>
> > Q7: Why are real human-labeled factuality datasets not used for calibration, given the reliance on DeepSeek-V3 pseudo-labels?
>
> Thank you for this thoughtful observation! Most mainstream methods [1, 4, 5, 9] also rely on ROUGE-L, BLEURT, or strong LLMs for annotation rather than manual human labeling, mainly because such datasets require extremely high annotation cost. The datasets we evaluate contain tens of thousands of samples in total, and obtaining high-quality human factuality labels for all model-generated answers would be prohibitively expensive. We include an illustration of dataset sizes below.
>
> | Dataset   | TruthfulQA   | TriviaQA  |CoQA  | TydiQA-GP | Total |
> | :--------: | :--------: | :--------: |:--------: | :--------: | :--------: |
> | Number |  817   |  9960   |  7983   |  3696    | 22456 |
>
> Moreover, we provide a human-annotation comparison in **Appendix H**. We randomly sampled 100 examples from TruthfulQA, manually annotated their factual correctness, and compared them against the pseudo-labels generated by DeepSeek-V3. With a threshold of 0.5, DeepSeek-V3 achieves Acc = 0.88 and F1 = 0.86 relative to human judgments, indicating that its labels are close to human-level quality and sufficiently reliable.

---

> ### Author Response · Authors · 2025-11-22
> **Response to Reviewer z8Nn (8/8)**
>
> > Question 8: Is there any evidence that the proposed metric generalizes to open-ended generation tasks, rather than structured QA datasets?
>
>
> Thank you for raising this important question. We can confirm that our approach is robust in open-ended generation settings. In fact, our experiments are not limited to structured QA. Among the four evaluation tasks:
> - COQA and TruthfulQA-generation are open-book generative tasks, where the model must produce open-ended, multi-sentence natural-language answers rather than structured outputs.
> - TYDIQA-GP is a reading comprehension task in which the model generates free-form textual answers, making it a semi–open-ended generation setting.
>
> These tasks do not rely on fixed templates or structured answer formats, and therefore already cover open-ended generation scenarios. The substantial improvements achieved by our method on these open-ended tasks demonstrate that the proposed metric generalizes well beyond structured QA.
>
> References
>
> [1] Du et al., HaloScope: Harnessing Unlabeled LLM Generations for Hallucination Detection. NeurIPS, 2024.
>
> [2] Hu et al., Embedding and Gradient Say Wrong: A White-Box Method for Hallucination Detection. EMNLP, 2024.
>
> [3] Azaria Amos and Mitchell Tom. The internal state of an LLM knows when it's lying. EMNLP, 2023.
>
> [4] Park et al., How to Steer LLM Latents for Hallucination Detection? ICML, 2025.
>
> [5] Kuhn et al., Semantic uncertainty: Linguistic invariances for uncertainty estimation in natural language generation. ICLR, 2023.
>
> [6] Chen et al., INSIDE: LLMs' Internal States Retain the Power of Hallucination Detection. ICLR, 2024.
>
> [7] Yang et al., Qwen2. 5 technical report. arXiv preprint arXiv:2412.15115, 2024.
>
> [8] Chiang et al., Vicuna: An Open-Source Chatbot Impressing GPT-4 with 90\%* ChatGPT Quality. 2023.
>
> [9] Lin et al., Truthfulqa: Measuring how models mimic human falsehoods. ACL, 2022.
>
> [10] Joshi et al., Triviaqa: A large scale distantly supervised challenge dataset for reading comprehension. ACL, 2021.
>
> [11] Reddy et al., Coqa: A conversational question answering challenge. Transactions of the Association for Computational Linguistics, 2019.
>
> [12] Clark et al., Tydiqa: A benchmark for information-seeking question answering in ty pologically di verse languages. Transactions of the Association for Computational Linguistics, 2020.
>
> [13] Lin et al., Generating with confidence: Uncertainty quantification for black-box large language models. Transactions on Machine Learning Research, 2024.
>
> [14] Li et al., Spotting AI’s Touch: Identifying LLM-Paraphrased Spans in Text. ACL, 2024.
>
> [15] Guo C, Sablayrolles A, Jégou H, et al. Gradient-based adversarial attacks against text transformers. EMNLP, 2021.
>
> [16] Mazeika, Mantas, et al. Harmbench: A standardized evaluation framework for automated red teaming and robust refusal. ICML, 2024.
>
> [17] Wen, Yuxin, et al. Hard prompts made easy: Gradient-based discrete optimization for prompt tuning and discovery. NeurIPS, 2023.
>
> [18] Koh, Pang Wei, and Percy Liang. Understanding black-box predictions via influence functions.  ICML, 2017.
>
> [19] Xia M, Malladi S, Gururangan S, et al. Less: Selecting influential data for targeted instruction tuning. ICML, 2024.
>
> [20] Gadre, Samir Yitzhak, et al. Datacomp: In search of the next generation of multimodal datasets. NeurIPS, 2023.

---

> ### Author Response · Authors · 2025-11-25
> **Invitation for Further Discussion**
>
> Dear Reviewer z8Nn,
>
> We have now posted a detailed response to your comments. Thank you again for your constructive feedback, which has helped us improve the paper.
>
> We would appreciate it if you could take a look at our rebuttal. If you find that our response has adequately addressed your concerns, we kindly ask that you consider reconsidering your score.
>
> We look forward to your feedback and are happy to answer any further questions.
>
> Best regards,
>
> The Authors

---

> ### Author Response · Authors · 2025-11-26
> **Invitation for Further Discussion**
>
> Dear Reviewer z8Nn,
>
> We have now posted a detailed response to your comments. Thank you again for your constructive feedback, which has helped us improve the paper.
>
> We would appreciate it if you could take a look at our rebuttal. If you find that our response has adequately addressed your concerns, we kindly ask that you consider reconsidering your score.
>
> We look forward to your feedback and are happy to answer any further questions.
>
> Best regards,
>
> The Authors

---

> ### Author Response · Authors · 2025-11-26
> **Invitation for Further Discussion**
>
> Dear Reviewer z8Nn,
>
> We have now posted a detailed response to your comments. Thank you again for your constructive feedback, which has helped us improve the paper.
>
> We would appreciate it if you could take a look at our rebuttal. If you find that our response has adequately addressed your concerns, we kindly ask that you consider reconsidering your score.
>
> We look forward to your feedback and are happy to answer any further questions.
>
> Best regards,
>
> The Authors

---

> ### Author Response · Authors · 2025-11-28
> **Invitation for Further Discussion**
>
> Dear Reviewer z8Nn,
>
> We have now posted a detailed response to your comments. Thank you again for your constructive feedback, which has helped us improve the paper.
>
> We would appreciate it if you could take a look at our rebuttal. If you find that our response has adequately addressed your concerns, we kindly ask that you consider reconsidering your score.
>
> We look forward to your feedback and are happy to answer any further questions.
>
> Best regards,
>
> The Authors

---

### Official Review · Reviewer_PEBK · 2025-11-02

**Soundness:** 3
**Presentation:** 3
**Contribution:** 3
**Rating:** 8
**Confidence:** 3

**Summary:**

The paper proposes a hallucination detection technique that uses expertly chosen prompts (adapted for each question) to perturb the distribution of true and hallucinated outputs, creating a bigger separation between them than without these additional prompts. The aim is to use these perturbations to increase the separability between true outputs and hallucinations. The paper first proposes a theoretical discussion on how a specific prompt can be crafted for each question that would almost perfectly separate the hallucinated and true outputs. The paper then supports its claim by comparing its hallucination detection technique with other SOTA techniques, showing definite improvements.

**Strengths:**

1. The paper provides both a theoretical analysis as well as the empirical results to support their hypothesis. To my understanding, they both appear to be on solid ground.
2. A large plethora of techniques are compared, across several different datasets, and two different models (a third one in the appendix). Overall, the experiments are robust enough to support the final claim.
3. The paper is well written and easy to read. I enjoyed reading this work.

**Weaknesses:**

I don't have any weaknesses in the soundness or contributions of this paper. I really enjoyed reading this work.

I do, however, have a big objection to papers that move the Related Work section to the appendix. The appendix is to provide additional results/analysis for readers who might be interested in learning more about the paper. It is NOT simply an extension of the main paper, and in my opinion, a lack of related works discussion in the main paper really hurts the readability of the work. I understand problems of limited space, but related work should not be the section that gets axed because of it.

I don't like it when reviewers suggest adding new parts to the paper without also suggesting what should be removed. Just a suggestion, I believe details about Theorem 1 and the results of Figure 2 can be compressed, with the rest moved to the appendix, to make space for related work discussion in the main paper (if the authors prefer, they can even have a 'shorter' related work section in the main paper to situate their work, and a 'longer' related work section in the appendix). Feel free not to take this suggestion, using your own way of finding space, but I strongly recommend having a brief discussion of related work in the main paper.

**Questions:**

My assumption is that the final 'training data' actually used to train the detector contains a set of 100 questions, one answer per question, and a label for whether that answer is a hallucination or the truth. This is what I believe is used in other detection methods, and so I assume the same is done here.

If the above is correct, the detector never really has access to the 'hallucinated output' and 'true output' pairs for the same question. While the objective is to push all hallucinated answers to low sensitivity and true answers to high sensitivity, which automatically creates a separation, I wonder if having access to actual pairs would help with the separability even more? Curious to hear if the authors think a small set of carefully labeled data with actual pairs could be beneficial, or if just pushing the answers to two extremes can achieve that implicitly?

---

> ### Author Response · Authors · 2025-11-21
> **Response to Reviewer PEBK (1/1)**
>
> We sincerely thank the reviewer for the thoughtful and positive assessment of our work, as well as for the constructive suggestions regarding the paper organization and training setup. Below we address each point in detail.
>
>
> > W1: The paper should retain a discussion of related work in the main body.
>
> Thank you for pointing this out! We have incorporated your suggestion by appropriately condensing the content related to `Theorem 1` and `Figure 2`. In the newly uploaded version of the paper, we have included a more concise Related Work section in the main text, along with a more detailed extended version in the **Appendix C**.
>
>
> > Q1: Data format
>
> Following [1,2], Our training data indeed uses the format $\{Question, Answer, Label\}$, where the label is $-1$ for hallucinated answers and $1$ for truthful answers. Each of the questions in the training set is paired with one model-generated answer and its corresponding truthfulness label.
>
>
>
> > Q2: If having access to hallucinated and true output pairs for the same question would help with the separability even more?
>
> We thank the reviewer for the insightful suggestion. Our additional experiments confirm that accessing paired truthful and hallucinated outputs for the same question further enhances separability.
>
> To construct manually paired data, we leverage the reference correct answers and incorrect answers provided in the TruthfulQA [3] dataset and treat them as paired samples for training. It is important to note that this setup differs from our main experimental setting, where the answers are generated directly by the LLM in a realistic scenario. We conducted additional experiments under this paired setting using LLaMA-3-8B-Instruct as the base model. The results are shown in the table below. We have included these additional experimental results in **Appendix J**.
>
>
> | Method | Average(ROUGE-L) |Average(BLEURT) |Average(DeepSeek-V3) |
> | -------- | :-----------:|:-----------:|:-----------:|
> | SSP     |   74.47     |   73.93     | 73.43  |
> | SSP_pairs |    75.59 ($\uparrow$ 1.12)   |   76.04 ($\uparrow$ 2.11)    | 75.81 ($\uparrow$ 2.38)  |
>
>
> The results indicate that, under this manually constructed paired-training scenario, our method indeed achieves a noticeable performance improvement. This suggests that SSP can effectively generate suitable prompts based on the input data and enhance the separability between true answers and hallucinated answers.
>
>
> References
>
> [1] Xuefeng Du and Chaowei Xiao and Yixuan Li. HaloScope: Harnessing Unlabeled LLM Generations for Hallucination Detection. NeurIPS, 2024.
>
> [2] Park et al. How to Steer LLM Latents for Hallucination Detection? ICML, 2025.
>
> [3] Lin et al. Truthfulqa: Measuring how models mimic human falsehoods. ACL, 2022.

---

> > ### Comment · Reviewer_PEBK · 2025-11-27
> >
> > The authors' response is appreciated, and I'm happy to see the related work back in the main text. The rebuttal is acknowledged, and I continue to be on the positive side for this paper.

---

> > > ### Author Response · Authors · 2025-11-27
> > > **Re: Reviewer PEBK- Appreciation and Response to Feedback**
> > >
> > > We are grateful for your encouraging comments and for recognizing the improvements in the revised manuscript. We also appreciate your constructive feedback throughout the review process, which has helped us to strengthen the paper.

---

### Author Response · Authors · 2025-11-22
**General Response**

Dear Reviewers and AC,

We sincerely appreciate the reviewers for their valuable time and effort in reviewing our manuscript. We are deeply encouraged by the recognition from multiple reviewers, who found our work to provide a **"Novel Perspective on Hallucination Detection"** (3guq) and a **"fresh angle"** (z8Nn), rest on **"solid ground"** (PEBK) with a **"Solid Theoretical Foundation"** (3guq), and demonstrate **"promising generalization performance"** through **"thorough evaluations"** (f1QU). This positive feedback has motivated us to further strengthen our work through careful revisions. We have summarized the reviewers' comments into the following six categories:

- Content organization and Related Work updates
- Computational cost and practicality analysis
- Validation of generalization and stability
- Clarification of theoretical contributions and implementation details
- Additional ablation studies
- Clarification of the relationship with previous work


To address the reviewers' concerns, we have conducted several additional experiments and analyses, including:

- Following the suggestion from **Reviewer PEBK**, we keep a concise version of the related work in **Section 3** and provide a more detailed discussion in **Appendix C**.

- In response to **Reviewer 3guq's** concern about using K-means to demonstrate poor separability, we revised **Figure 1(a)** using a supervised logistic regression method to **strengthen the motivation for our work.**

- In response to the shared concern from **Reviewers z8Nn, f1QU, and 3guq** regarding computational complexity and practical usability, we show the inference time of SSP in real-world scenarios in **Appendix Q**.

- In response to the concerns from **Reviewers z8Nn, f1QU, and 3guq** about the performance of SSP, we further examine its generalization capability and show strong cross-dataset results in **Appendix K**.

- In response to **Reviewer z8Nn's** concern about robustness, we paraphrased the input samples in various styles to evaluate the stability of SSP, as described in **Appendix O**.

- In response to **Reviewer f1QU's** question about the performance on newer and smaller models, we introduce experiments using LLaMA-3.2-1B to evaluate the stability of SSP, as shown in **Tables 1, 5, and 6**.

- In response to the questions from **Reviewers z8Nn, f1QU, and 3guq** about our contributions and the feasibility of the separability theory, we explain the implementation of the theory and its validity in **Section 4 and Appendix E**.

- In response to **Reviewer PEBK's** question about whether carefully constructed paired data helps performance, we provide additional ablation studies and analysis in **Appendix J**.

- In response to **Reviewer f1QU's** concern about the interpretability of perturbation prompts, we decode and display the specific perturbation prompts in **Table 4**.

- In response to **Reviewers f1QU and 3guq's** questions about the relationship between our theoretical framework/statistical results and prior work, we further expand the discussion of related work and baselines in **Section 3** and in **Tables 1, 5, and 6**.


All revisions in the paper are indicated in **green font**. We believe these revisions have significantly strengthened the paper and addressed all raised concerns.


Thank you very much,

Authors.

---

### Comment · Area_Chair_LvFy · 2025-11-27
**Rebuttal and Discussion Phase**

Dear Reviewers,

Thank you again for your time and effort in reviewing this paper. We are approaching the discussion deadline. I kindly ask you to review the rebuttal and continue the discussion so that we can reach a well-considered decision.

---

### Meta-Review · Area_Chair_ATDm · 2025-12-28

**Summary:**

This paper proposes a sensitivity-based approach for hallucination detection in LLMs, arguing that truthful and hallucinatory responses exhibit different sensitivities to prompt-induced perturbations in their internal representations. The authors provide theoretical analysis. Reviewer PEBK was positive (score 8), praising the solid theoretical and empirical grounding. Reviewers z8Nn, f1QU, and 3guq (all score 4) raised concerns about: (1) the novelty and relationship to prior perturbation-based methods, (2) the gap between theoretical oracle bounds (99%) and practical performance (75%), (3) computational costs. The AC finds merit in this paper but cannot recommend acceptance at this time due to the highly competitive nature of this conference.

**Reviewer Concerns:**

The authors addressed several concerns effectively: they added SPUQ as a baseline and clarified the distinction between output-level and embedding-level perturbation analysis; they provided detailed computational cost comparisons ; they replaced K-means with supervised logistic regression in Figure 1 to strengthen the motivation.

**Reviewer Scores:**

The reviewer who initially gave a score of 8 maintained that score. Another reviewer who provided a score of 4 engaged in discussions but does not seem inclined to raise his score. Two additional reviewers did not respond, making it difficult to determine if they would consider increasing their scores based on their initial skeptical reviews.

---

### Decision · Program_Chairs · 2026-01-26

Reject